# Characterization of *PIK3CA* and *PIK3R1* somatic mutations in Chinese breast cancer patients

Li Chen[1,2], Liu Yang[1,2], Ling Yao[1], Xia-Ying Kuang[3], Wen-Jia Zuo[1,2], Shan Li[1], Feng Qiao[1], Yi-Rong Liu[1,2], Zhi-Gang Cao[1], Shu-Ling Zhou[2,4], Xiao-Yan Zhou[2,4], Wen-Tao Yang[2,4], Jin-Xiu Shi[5], Wei Huang[5], Xin Hu[1,2] & Zhi-Ming Shao[1,2]

Deregulation of the phosphoinositide 3-kinase (PI3K) pathway contributes to the development and progression of tumors. Here, we determine that somatic mutations in *PIK3CA* (44%), *PIK3R1* (17%), *AKT3* (15%), and *PTEN* (12%) are prevalent and diverse in Chinese breast cancer patients, with 60 novel mutations identified. A high proportion of tumors harbors multiple mutations, especially *PIK3CA* plus *PIK3R1* mutations (9.0%). Next, we develop a recombination-based mutation barcoding (ReMB) library for impactful mutations conferring clonal advantage in proliferation and drug responses. The highest-ranking *PIK3CA* and *PIK3R1* mutations include previously reported deleterious mutations, as well as mutations with unknown significance. These *PIK3CA* and *PIK3R1* impactful mutations exhibit a mutually exclusive pattern, leading to oncogenesis and hyperactivity of PI3K pathway. The *PIK3CA* impactful mutations are tightly associated with hormone receptor positivity. Collectively, these findings advance our understanding of PI3K impactful mutations in breast cancer and have important implications for PI3K-targeted therapy in precision oncology.

[1] Key Laboratory of Breast Cancer in Shanghai, Department of Breast Surgery, Fudan University Shanghai Cancer Center, Shanghai 200032, China. [2] Department of Oncology, Shanghai Medical College, Fudan University, Shanghai 200032, China. [3] Department of Breast Surgery, The First Affiliated Hospital, Sun Yat-Sen University, Guangzhou 510080, China. [4] Department of Pathology, Fudan University Shanghai Cancer Center, Shanghai 200032, China. [5] Department of Genetics, Shanghai-MOST Key Laboratory of Health and Disease Genomics, Chinese National Human Genome Center and Shanghai Industrial Technology Institute (SITI), Shanghai 201206, China. These authors contributed equally: Li Chen, Liu Yang, Ling Yao. Correspondence and requests for materials should be addressed to X.H. (email: xinhu@fudan.edu.cn) or to Z.-M.S. (email: zhimingshao@yahoo.com)

Breast cancer is the most common cancer among women worldwide[1]. Recent comprehensive cancer genomic analyses have revealed that components of the phosphoinositide-3 kinase (PI3K) pathway are frequently altered in human cancers[2]. The somatic mutations in the components of this pathway include mutations in *PIK3CA*, which encodes the PI3K catalytic subunit p110α, *PIK3R1*, which encodes the PI3K regulatory subunit p85α, and other genes encoding the PI3K-associated modulators. Many of these mutations could determine pathway activation[3,4]. Signaling through the PI3K pathway is essential for the occurrence of both normal and malignant cellular processes, including proliferation, apoptosis, and metabolism[5,6].

The *PIK3CA* gene is mutated in ~36% of breast cancers reported in The Cancer Genome Atlas (TCGA) and the Catalogue of Somatic Mutations in Cancer (COSMIC) databases. A total of 160 *PIK3CA* somatic mutations, including 125 missense mutations, 8 silent mutations, 1 nonsense mutation, 19 deletion-frameshift mutations, and 7 insertion-frameshift mutations, have been reported in breast cancer in the TCGA[7] and COSMIC[8] databases. These mutations most frequently occur in the helical domain (hotspots E545K and E542K) or the kinase domain (hotspot H1047R) of the *PIK3CA*-encoded p110α[2]. These tumor-associated *PIK3CA* mutations lead to constitutive p110α activation and oncogenic transformation in multiple cancers[9–12].

*PIK3R1* encodes the p85α regulatory subunit which inhibits the kinase catalytic activity of p110α. p85α also binds to PTEN, preventing PTEN ubiquitination and increasing its stability[13]. Functional mutants in p85α lack the ability to bind p110α and PTEN, leading to upregulation of PI3K signaling[14]. Recently, Cheung et al. uncovered an unexpected neomorphic role for the *PIK3R1* R348* and L370fs truncations by showing that these truncated proteins were localized to the nucleus, where they facilitated nuclear JNK pathway activation[15].

Major advances in sequencing technology have led to a dramatic increase in the rate of discovery of non-hotspot novel mutations[16]. In Western populations of breast cancer patients, somatic mutations in the PI3K pathway have been systematically analyzed[2,10,17]. However, there has not yet been a systematic study of somatic alterations in the PI3K pathway in Chinese breast cancer patients. Furthermore, our understanding of the functions of somatic mutations in this pathway has not kept pace with the discovery of individual mutations. Therefore, the functional characteristics and clinical significance of rare mutations in the PI3K pathway remain to be elucidated. Although in silico analysis of the evolutionary conservation of the amino acids affected can predict the impact of the mutation on protein folding, recent large-scale functional assays are expected to provide more direct evidence regarding the biological effects of somatic mutations[18,19].

Here, we survey the landscape of somatic mutations affecting the major components of the PI3K pathway in Chinese breast cancer patients using amplicon sequencing. We also perform a high-throughput mutation-phenotype screen for impactful mutations that contribute to cancer development and drug resistance. We identify a series of mutations in *PIK3CA* and *PIK3R1* that lead to oncogenesis and hyperactivity of the PI3K pathway. We find a pattern of mutual exclusivity for driver mutations in *PIK3CA* and *PIK3R1* that encode two proteins whose activity coordinately regulate PI3K activation. Furthermore, the *PIK3CA* impactful mutations display a tight correlation with hormone receptor positivity in TCGA data set and our cohort. Together, the present study seeks to uncover a rational basis for PI3K impactful mutations in breast cancer, in hopes that it may be used as a basis for clinical intervention in patients with PI3K pathway hyperactivity.

## Results

### Somatic mutations of PI3K pathway in Chinese breast cancer.
Here, we sought to provide a landscape of non-synonymous somatic mutations of PI3K pathway in Chinese breast cancer patients. Our cohort included 149 patients ranging from 25 to 72 years of age who were pathologically confirmed at Fudan University Shanghai Cancer Center (FUSCC). The patients' clinicopathological parameters are listed in Supplementary Table 1. In all, 32 (21.5%) of the patients were Luminal A, 75 (50.3%) were Luminal B, 17 (11.4%) were HER2-enriched, and 25 (16.8%) were TNBC (Fig. 1a). We performed amplicon exon sequencing of the following genes: *PIK3CA*, *PIK3R1*, *AKT1*, *AKT2*, *AKT3*, *PTEN*, and *PDK1* at ~1000× coverage. Sanger sequencing of paired blood DNA was conducted in all cases to exclude germline mutations. In total, we identified 75 somatic missense mutation events (Supplementary Table 2). Of the patients, 89 (60%) possessed one or more mutations in genes involved in the PI3K pathway (Fig. 1a).

*PIK3CA* was the most frequently mutated gene (65 samples, 43.6%), which was consistent with its high frequency (36%) in the TCGA database[2] (Fig. 1b). The *PIK3CA* mutation frequency differed for the four subtypes, with 50.0% (16/32) of Luminal A, 46.7% (35/75) of Luminal B, 41.2% (7/17) of HER2-enriched, and 28% (7/25) of TNBC patients (Fig. 1c). The *PIK3CA*-encoded p110α contains an N-terminal adapter-binding domain (ABD), a Ras-binding domain (RBD), a C2 domain, a helical domain, and a catalytic domain. The *PIK3CA* hotspot mutations (E542K, E545K, and H1047R) within helical and kinase domains in previous literatures[20] also had high frequencies in our cohort (Supplementary Table 3). Thirteen novel mutations were also detected; three (M30V, E39K, and K51N) of these were in ABD domain, three (N202D, I211S, and Q219R) were in RBD domain, two (R516G and N521S) were in the helical domain, one (K1041E) was in the kinase domain, and four (W498R, S514G, E710G, and Q760L) were in the linker regions (Fig. 1d).

*PIK3R1* represents one of frequently mutated genes in tumors, including endometrial carcinoma (33.8%), metastatic prostate adenocarcinoma (11.5%), and glioblastoma (11%)[14]. Non-synonymous *PIK3R1* mutations were reported at a moderate frequency (2.8%) in TCGA breast cancer database[2]. *PIK3R1* mutations were prevalent (25 tumors, 17%) in our cohort (Fig. 1a). The luminal B subtype had a high frequency of *PIK3R1* (19/75, 25.3%) mutations (Fig. 1e). The *PIK3R1*-encoded p85α regulatory subunit has an N-terminal SH3 domain, a domain homologous to the Rho GTPase-activating protein domain (Rho-GAP), and three SH2 domains (nSH2, iSH2, and cSH2). The somatic mutations in *PIK3R1* were located in Rho-GAP, iSH2, cSH2, and linker region (Fig. 1f and Supplementary Table 4). A series of novel *PIK3R1* mutations were found in our cohort; these included P84L, E160D, D178N, V213D, Q329L, Y504D, Q552K, I559N, N564D, N595K, N600H, N632I, N673T, and K674R.

*AKT1* (10.1%), *AKT2* (10.1%), *AKT3* (14.8%), and *PDK1* (4.7%) mutations also presented a higher frequency in the FUSCC cohort than that in the TCGA data set. Although the E17K mutation in the *AKT1* is common in Chinese patients, we also observed eight novel *AKT1* mutations, five novel *AKT2* mutations, six novel *AKT3* mutations, and seven novel *PDK1* mutations in our cohort (Supplementary Fig. 1). We detected seven novel mutations in *PTEN* which included the recurrent mutation P283A. In this study, a total of 60 novel somatic mutations are identified in the FUSCC cohort.

Recently, Pereira et al. found that *PIK3CA* or *PIK3R1* aberrations frequently co-occurred with *PTEN* mutations in breast cancer[20]. Consistent with this phenomenon, we also demonstrated that a high proportion of patients from FUSCC harbored multiple somatic mutations in the PI3K pathway;

notably, *PIK3CA* plus *PIK3R1* mutations were identified in 9% of our cohort (Fig. 1g).

**ReMB library screening design and its applications.** High-throughput pooled screening provides an efficient approach to the investigation of mutation phenotypes. Here, we employed a recombination-based retroviral expression system in which 30-bp unique oligonucleotide pairs were inserted into the retroviral backbone, resulting in a set of vectors that contained various barcodes (Fig. 2a). The human full-length *PIK3CA* and *PIK3R1* coding sequences (CDSs) were cloned into the donor vector. Based on the mutation profiles of the TCGA, COSMIC, and FUSCC cohorts (Supplementary Fig. 2), a set of *PIK3CA* and

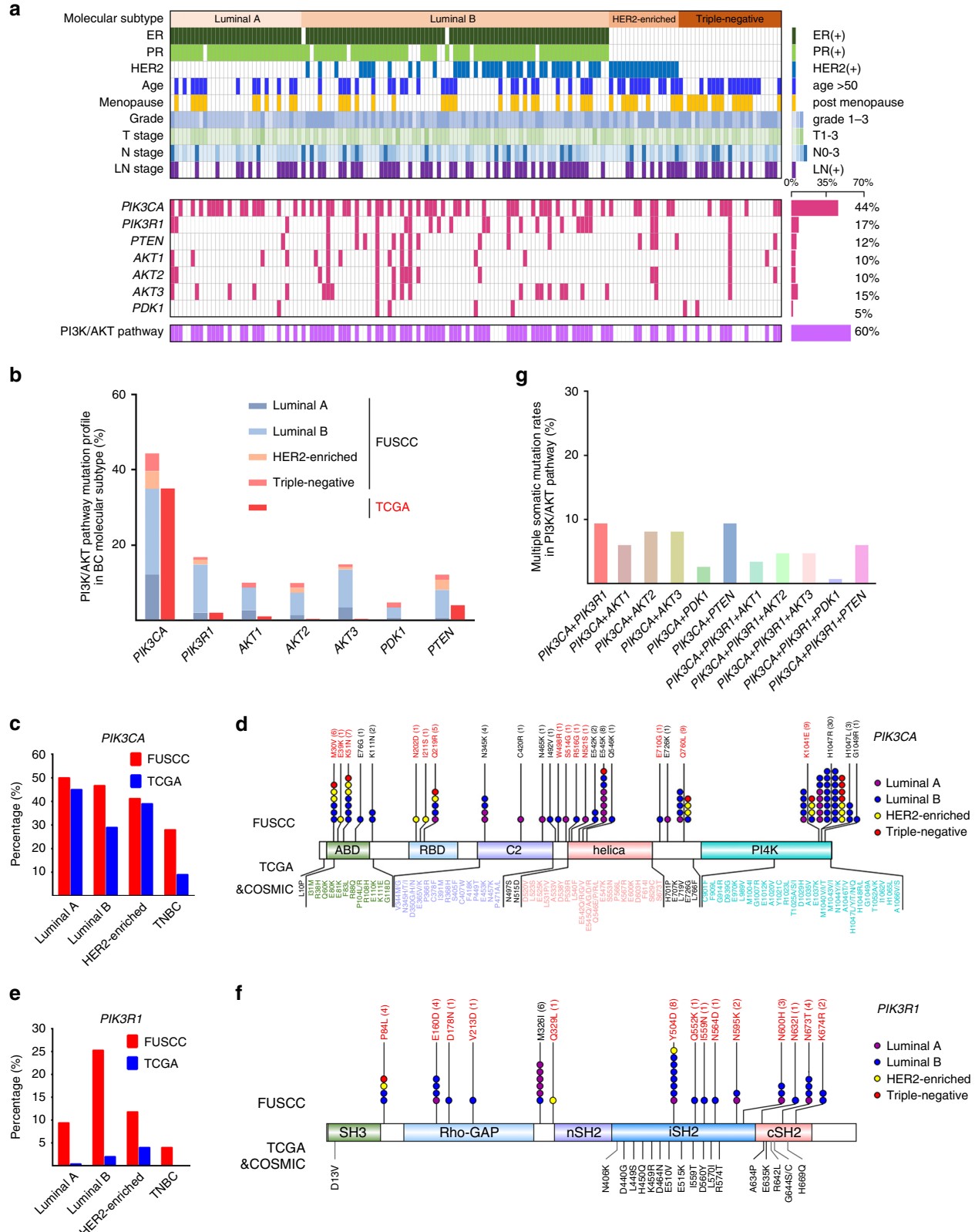

*PIK3R1* mutants were generated. The recombination reaction was performed using a retrovirus-based mutation-barcode plasmid, as previously described[21]. All of the *PIK3CA* or *PIK3R1* clones were mixed into a pool with equal amount for packaging of the retroviral *PIK3CA*- or *PIK3R1*-ReMB library (Fig. 2a). Next, normal mammary cells (MCF-10A and HMEC) were transduced with the *PIK3CA*- or *PIK3R1*-ReMB virus at a low multiplicity of infection (MOI = 0.3). After selection with puromycin for 5 days, only cells with an exogenous *PIK3CA* or *PIK3R1* mutation sequence were preserved, and each individual mutant was represented by at least 2000 cells. Subsequently, the pooled ReMB-transduced cells were used in proliferation and drug resistance assays.

The ReMB library enables us to rapidly screen for driver mutations associated with proliferation ability and drug (doxorubicin and pan-Class I PI3K inhibitor BKM120) responses. First, $IC_{50}$ assays were performed to determine the doxorubicin and BKM120 concentrations that were effective in inhibiting the growth of transduced mammary cells (Supplementary Fig. 3). As shown in Fig. 2b, exposure to doxorubicin (1.5 nM) or BKM120 (1.2 µM) resulted in growth inhibition of transduced mammary cells. To identify mutations that confer resistance to doxorubicin or BKM120, genomic DNA was extracted from the cells that survived to Day 7 or 14 in the cell proliferation and drug response screens as well as from the original cells after puromycin selection (Day 0). The genomic DNA was subjected to PCR amplification of the barcoded regions and analyzed using next-generation sequencing (NGS). The barcoded regions can be specifically amplified without interference from the endogenous host DNA (Fig. 2c). Sanger sequencing indicated that the barcoded region presented as mixed peaks (Fig. 2d).

**ReMB screenings for impactful *PIK3CA* and *PIK3R1* mutations.** NGS analyses indicated that all of the barcodes corresponding to mutations could be aligned to the *PIK3CA*- and *PIK3R1*-ReMB libraries. The cumulative distribution frequency curves demonstrated that there was no significant change in the total number of mutants in the screens (Fig. 3a, b). The read counts for most of the barcodes were consistent in two separate replicates of the proliferation and drug resistance assays, respectively (Fig. 3c, d).

We ranked the enrichment scores (ESs) of the mutations identified in the proliferation and drug resistance assays. A small number of mutations were enriched in these screens, whereas most of the mutations showed few changes (Fig. 3e–h and Supplementary Fig. 4). The average enrichment score (Avg ES) for impactful mutations in *PIK3CA* and *PIK3R1* were illustrated in Supplementary Table 5–10. The hierarchical clustering maps based on the ES signatures showed that the enriched mutations assembled consistently in the replicates, indicating that ReMB

screening provides a reliable method of identifying impactful mutations.

**Phenotypic characterization of impactful *PIK3CA* mutations.** Overall, we identified 11 non-synonymous impactful mutations in *PIK3CA*; these included eight proliferation-driving mutations, nine doxorubicin-resistant mutations, and eight BKM120-resistant mutations (Fig. 4a). The fold change information is illustrated in Supplementary Fig. 5. As expected, our highest-ranking proliferation-driving mutations included the *PIK3CA* "hotspot" mutations N345K, E542K, E545K, and H1047R/L[18], as well as rare mutations with fewer previous reports, including N345I, E453K, M1043V, H1047T, and G1049R. Most mammary cells that harbor proliferation-driver mutations also exhibited a relatively higher tolerance to doxorubicin and BKM120 treatment (Fig. 4a). Mutations N345I, E453K, E542K, H1047R/L, and G1049R were related to both doxorubicin and BKM120 resistance. Interestingly, the impactful *PIK3CA* amino acid substitutions are distributed not only within the helical and PI4K domains, but also within the ABD and C2 domains (Supplementary Table 11).

Figure 4b presents a schematic structure of the *PIK3CA* protein showing the amino acid changes associated with cell proliferation and drug response. The three-dimensional positions of mutated residues are shown in Fig. 4c. Most impactful mutations in *PIK3CA* and *PIK3R1* occurred at residues lying at the interfaces between p110α and p85α, or between the functional domains within p110α[22]. (1) E39K in ABD domain: the ABD domain not only binds to the iSH2 domain of p85α, but also interacts with the PI4K kinase domain in p110α[9]. Arg38 and Arg88 are located at the interface between the ABD and the kinase domains, within hydrogen-bonding distance of Gln738, Asp743, and Asp746 of kinase domain. Mutations of Arg38 and Arg88 are likely to disrupt the interactions and resulted in conformational change of the kinase domain of p110α[22]. The E39K mutation identified in our study have not been reported elsewhere. We hypothesize that the E39K mutation may either affect the interactions with the kinase domain or function by changing construct of Arg38. (2) N345K/I and E453K in C2 domain: the Asn345 mutation may disrupt the interaction of the C2 domain with iSH2, thus altering the regulatory effect of p85 on p110α[22,23]. (3) E542K and E545K in helical domain: the helical domain mediates interaction with the nSH2 domain of p85α, which is responsible for the p85α-induced inhibition of p110α. E542K and E545K mutations in the helical domain could interfere with this p85α–p110α interaction and disrupt signal regulation[22,24,25]. (4) M1043V, H1047R/T/L, and G1049R in kinase domain: Ignacia Echeverria et al. reported that H1047R hotspot mutation in the kinase domain changed the orientation of His1047, which resulted in easier access to membrane-bound phosphoinositol-4,5-bisphosphate substrate of

---

**Fig. 1** Mutational profiling of the PI3K pathway and the clinical features of 149 breast cancers. **a** Tumor samples grouped by subtype: luminal A ($n = 32$), luminal B ($n = 75$), HER2-enriched ($n = 17$), and basal-like ($n = 25$). The total number of mutations in each case and in each gene is illustrated. The percentage of mutations in each gene is shown on the right. **b** Comparison of mutation frequencies in our profiling with those in the TCGA data set. Higher frequencies of mutations in *PIK3R1*, *AKT1*, *AKT2*, and *AKT3* were observed in the FUSCC cohort. The distributions of the mutations with respect to cancer subtype are indicated by color. **c** Comparison of *PIK3CA* mutation frequency in the FUSCC and TCGA data sets according to breast cancer molecular subtype. **d** The *PIK3CA* protein possesses five domains: an adapter-binding domain (ABD), a Ras-binding domain (RBD), the C2 domain, a helical domain, and the PI4K kinase catalytic domain. The substitutions shown at the top are the non-synonymous mutations that were identified in the current study; the substitutions shown on the bottom are the non-synonymous mutations described in the TCGA and COSMIC data sets. The novel identified mutations are highlighted in red. Each filled circle represents an individual mutated tumor sample, whereas different colors indicate the tumor subtypes: luminal A (purple), luminal B (blue), triple-negative (red), and HER2-enriched (yellow). **e** Comparison of *PIK3R1* mutation frequencies among different tumor subtypes in FUSCC and TCGA. **f** Schematic structure of *PIK3R1*. **g** Multiple somatic mutations in the major components in the PI3K pathway were found in the FUSCC cohort

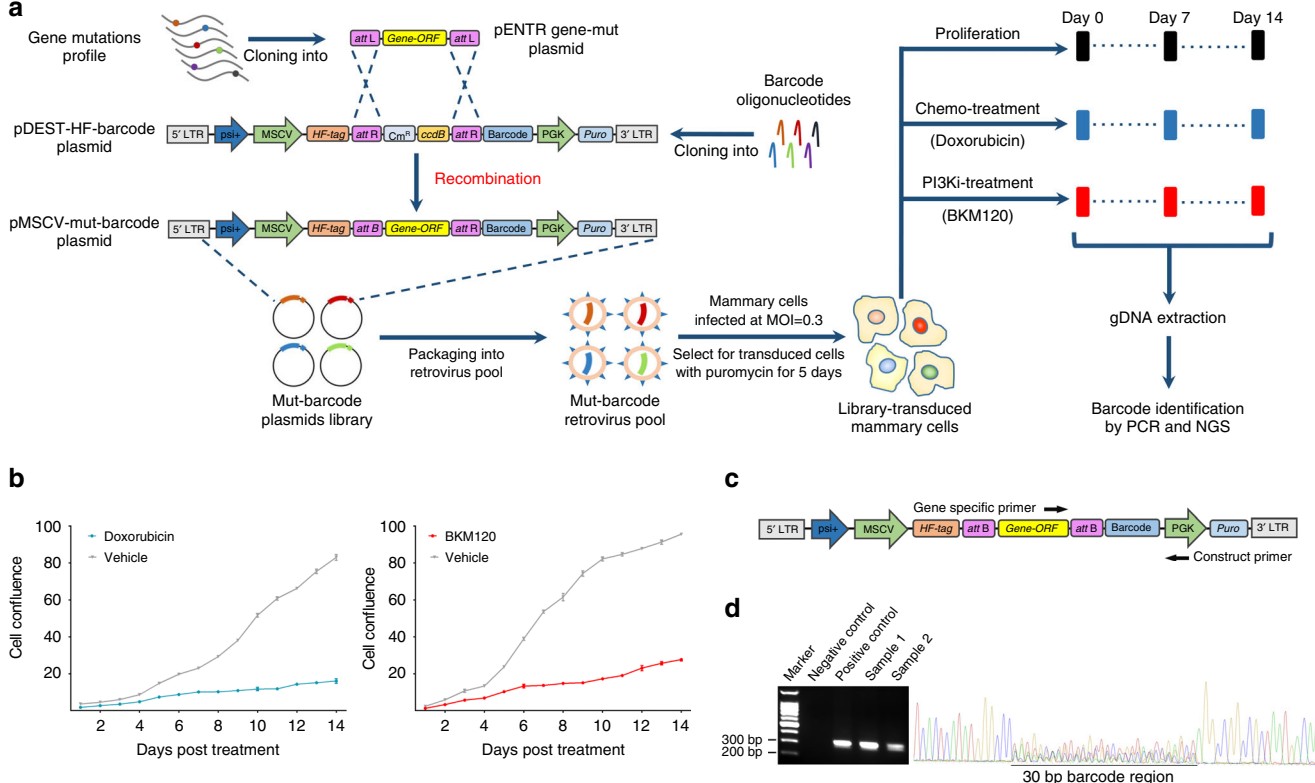

**Fig. 2** Design of the recombination-based mutation barcoding (ReMB) library and its application to functional screening. **a** The design of the ReMB library and its screening schema are illustrated. We constructed a series of donor clones containing the gene-coding sequences of wild-type or various mutations according to the mutation profiles of the TCGA, COSMIC, and FUSCC cohorts. A retroviral pDEST-HA-Flag vector was used to construct the barcode plasmids. Random 30-bp annealed oligonucleotide pairs were inserted into this retroviral backbone. We then constructed a set of vectors containing various barcodes with one barcode in each vector. The recombination reaction was performed for a retroviral mutation plasmid that contained a unique barcode sequence. All of the clones were mixed into a pool to package a gene mutations retrovirus library that included mutants, wild-type and negative controls. The gene mutation library was delivered into MCF-10A or HEMC cells by retroviral infection at an MOI of 0.3. Cells stably expressing the mutated genes were obtained by puromycin selection for 5 days. Functional screens were conducted for 2 weeks to examine proliferation, doxorubicin, and BKM120 responses, followed by PCR amplification of the barcode sequences that were integrated into the chromosomes. The purified PCR products were subjected to barcode deconvolution analysis using NGS. **b** Growth curves were obtained for transduced MCF-10A cells that were exposed to concentrations of doxorubicin (1.5 nM) or BKM120 (1.2 μM) for 14 days. Five wells were measured per condition. The error bars indicate mean ± s.d. derived from three independent experiments. **c** Schematic showing the primer design. **d** Electrophoresis revealed a DNA fragment containing 30-bp barcodes after PCR; Sanger sequencing was used to validate the mixed barcode sequences

$PIK3CA$[26]. Paraskevi Gkeka et al. reported that the hydrogen bonds between His1047 and Met1043, and between Gly1049 and Asn1044 contributed in the stabilization of kinase domain of wild-type $PIK3CA$[27]. The hydrogen bonds were disrupted in H1047R mutant and consequently facilitated ATP hydrolysis. We speculate that mutations at residue Met1043 and Gly1049 could promote $PIK3CA$ activity in a similar way. Together, these impactful mutations were considered to be structural damaging alteration as disease-causing drivers.

To validate these top-ranking mutations, we transduced retrovirus expressing the wild-type or mutant $PIK3CA$ sequences into MCF-10A cells to generate stable cell lines. Growth-factor-independent proliferation is considered to be one of the hallmarks of cancer[10]. The exogenous expression of oncogenic mutations such as $PIK3CA$ H1047R or E545K is sufficient to induce proliferation in MCF-10A cells[28,29]. As expected, the cells expressing proliferation-driving $PIK3CA$ mutants were able to proliferate in the absence of epidermal growth factor (EGF), whereas the proliferation rates of the wild-type and control MCF-10A cells were lower (Fig. 4d). On the contrary, the cells expressing non-impactful $PIK3CA$ mutants (K51N, D538Y, and E710G) showed no pro-proliferation effect (Supplementary

Fig. 6a). When plated on basement membrane gels, mammary epithelial cells form acinar-like structures consisting of a hollow lumen surrounded by cells[30]. As shown in Fig. 4e, MCF-10A cells expressing proliferation-driving $PIK3CA$ mutants (E39K, E542K, and H1047R) formed large, highly proliferative, abnormal structures in the matrix; in contrast, cells expressing wild-type $PIK3CA$ formed acini similar to those formed by control cells.

The status of the major components of the PI3K pathway was also monitored. When growth factors were withdrawn, phosphorylation of AKT (p-AKT) at Ser473 residue in MCF-10A cells expressing wild-type $PIK3CA$ and controls were significantly lower than those in cells maintained in normal growth medium. In contrast, MCF-10A cells with impactful mutants continued to exhibit high p-AKT levels in the absence of growth factors (Fig. 4f).

In cells expressing impactful $PIK3CA$ mutations, the basal level of p-AKT was near maximal among increasing EGF doses; whereas in cells expressing wild-type $PIK3CA$, an EGF concentration of 0.05 ng ml$^{-1}$ or higher was needed to achieve a similar level of relative activation (Fig. 4g). In addition, in the cells expressing impactful $PIK3CA$ mutations, the basal levels of p-AKT(Ser473) were higher than the maximal p-AKT(Ser473)

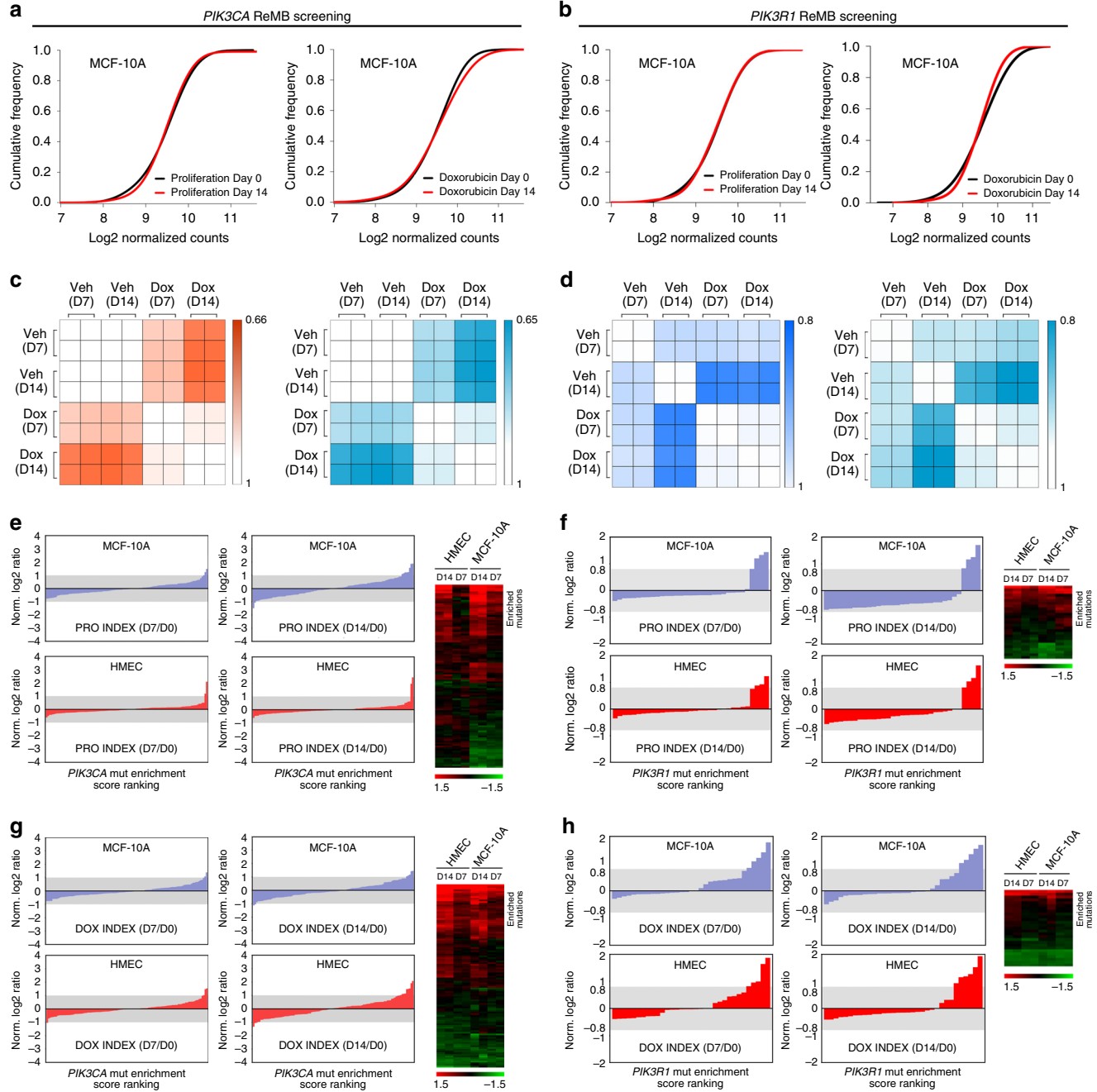

**Fig. 3** ReMB library screening for proliferation- and doxorubicin resistance-associated mutations of *PIK3CA* and *PIK3R1* in MCF-10A and HEMC cells. **a**, **b** Cumulative frequency of *PIK3CA* and *PIK3R1* mutations in the proliferation and doxorubicin response assays on Day 0 and Day 14 after transduction. **c**, **d** Rank correlations of normalized read counts between biological replicates and treatment conditions in MCF-10A and HMEC cells (Veh vehicle, Dox doxorubicin). Pearson's correlation coefficient was used for comparisons. **e–h** Waterfall chart showing the enrichment of mutations in the indicated assays normalized to Day 0 read counts in MCF-10A and HMEC cells. For each cell line, the mutations were ranked on the basis of the mean normalized log$_2$ (Day 14/Day 0) ratios and log$_2$ (Day 7/Day 0) ratios of the read counts. The shaded rectangle indicates a log$_2$ ratio range between −1 and 1. The clustering of the two cell lines reveals the enriched mutations that were consistent in the proliferation and drug response screens. The color scale represents the mean normalized log$_2$ (Day 14/Day 0) ratios and log$_2$ (Day 7/Day 0) ratios of the read counts

levels in the control cells. Our results show that these impactful *PIK3CA* mutations activate the PI3K pathway in mammary epithelial cells in a growth-factor-independent manner. We further examined whether cells expressing impactful *PIK3CA* mutants displayed enhanced resistance to BKM120. In cells expressing *PIK3CA* mutations, p-AKT levels did not decrease until high concentration of BKM120 was added; whereas in cells expressing control or wild-type *PIK3CA*, p-AKT levels decreased with a lower concentration of BKM120 (Fig. 4h). As expected,

these impactful mutants of *PIK3CA* implicated in doxorubicin- and BKM120-resistance promote cells proliferation in growth medium containing doxorubicin (1 nM) or BKM120 (1 μM). In contrast, the growth of cells carrying *PIK3CA* non-impactful mutations (E710G and K51N) were repressed (Fig. 4i, j).

**Phenotypic characterization of impactful *PIK3R1* mutations.** We also identified six non-synonymous impactful mutations in

*PIK3R1*, including five proliferation-driving mutations, six doxorubicin-resistant mutations, and five BKM120-resistant mutations (Fig. 5a). The fold change information is illustrated in Supplementary Fig. 7. Of these impactful mutations, D560Y and R574T were reported only once (1/3554) in COSMIC breast samples, respectively. Four *PIK3R1* variants (E160D, Q239L,

N564D, and K674R) are novel mutations identified in our cohort. It is noteworthy that *PIK3R1* E160D mutation represents one of the medium-frequency mutations in our cohort (4/149, 2.7%). The *PIK3R1* mutations exhibiting higher clonal advantages in proliferation and drug resistance to doxorubicin includes E160D in Rho-GAP domain, D560Y, N564D, and R574T in iSH2

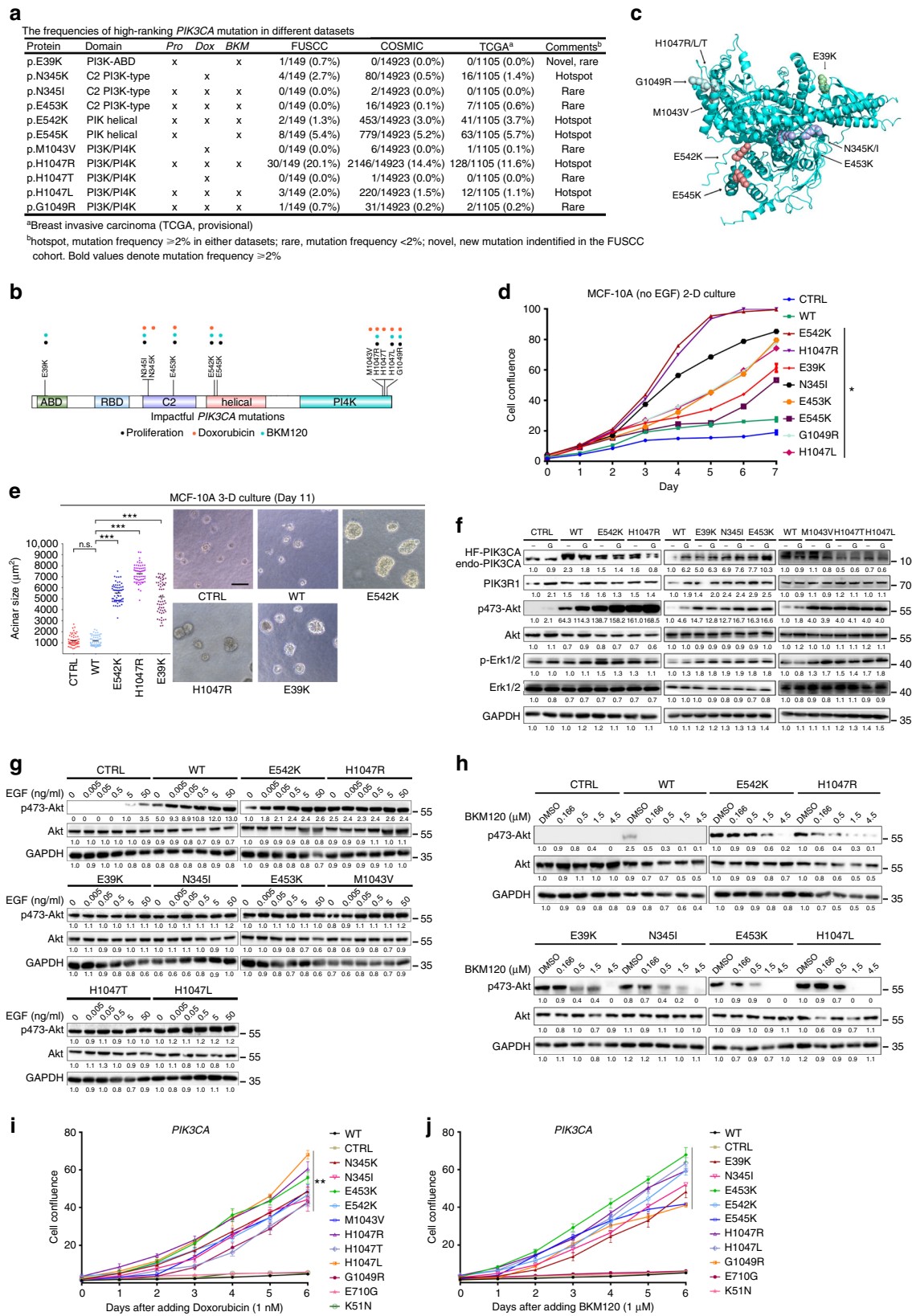

domain, and Q392L and K674R in linker domain (Supplementary Table 12). These mutations also exhibited resistance to BKM120, with the exception of K674R. Similarly, these drug-resistant mutations mostly overlap with driver mutations identified in the proliferation assay. These impactful *PIK3R1* mutations are illustrated in Fig. 5b. The three-dimensional positions of individual mutated residues are shown in Fig. 5c. The impactful *PIK3R1* mutations were located in various domains, suggesting they may also have different mechanisms. (1) E160D in Rho-GAP domain: the SH3 and Rho-GAP domain is responsible for PTEN binding[13]. Cheung et al. demonstrated that the *PIK3R1* E160* mutation disrupted the interaction between p85α and PTEN, resulting in destabilization of PTEN, increased PI3K signaling and transformation[31]. The ReMB screen showed that E160D had clonal advantages in cell proliferation and drug response, suggesting that novel E160D mutation might be a functional mutation in the Rho-GAP domain. (2) D560Y, N564D, and R574T in SH2 domains: the nSH2 and iSH2 domains are required for interaction with the ABD, C2, and helical domains of p110α. Recent studies have indicated that multiple functional mutations (K379E, R503W, KS549delN, D560Y, KS549delN, N564D, and QYL579-delL) in the nSH2 and iSH2 domains of *PIK3R1* activate the canonical PI3K pathway by disrupting the inhibitory contact of the C2 domain in p110α[32]. We speculate that N564D and R574T mutations may function in a similar way. (3) Q329L and K674R in linker regions: we identified Q329L (at the nSH2 domain boundary), and K674R (at the cSH2 domain boundary) as mutations that affect cell proliferation and drug response. The above findings suggest that more detailed classification of functional mutations in *PIK3R1* and additional research on the mechanisms involved will be needed.

We further generated stable cell lines carrying the wild-type or mutant *PIK3R1*. The novel proliferation-driving *PIK3R1* mutants E160D, Q329L, N564D, and K674R promoted cell proliferation in the absence of EGF (Fig. 5d, up). Although the oncogenic mutations in the *PIK3R1*-encoded p85α regulatory subunit exhibited relatively weak ability to promote growth-factor-independent proliferation, their proliferation rates were markedly upregulated in the presence of EGF (Fig. 5d, down). While the non-impactful *PIK3R1* mutants (I559N, D440G, and G644C) could not promote proliferation in comparison to positive controls (K674R and N5564D) (Supplementary Fig. 6b, c). Additionally, proliferation-driving *PIK3R1* mutations can cause cells to form abnormal acinar structure, likely reflecting the fact that *PIK3R1* deficiency is sufficient to initiate oncogenesis and pathway activation (Fig. 5e).

We found that the levels of p-AKT (Ser473) were slightly higher when cells with exogenous *PIK3R1* mutations were grown in the absence of EGF, whereas cells expressing wild-type *PIK3R1* or control cells exhibited less AKT activation signaling (Fig. 5f). In contrast, these driver mutations significantly upregulated AKT signaling upon treatment of EGF (Fig. 5f). Cells with the *PIK3R1* K674R or E160D mutations showed higher resistance to activation of AKT signaling in response to BKM120 than cells expressing wild-type *PIK3R1* or control cells (Fig. 5g). The *PIK3R1* impactful mutations could render cells proliferation at low concentrations of doxorubicin or BKM120. On the contrary, the growth of cells carrying non-impactful mutation (D440G) were repressed (Fig. 5h, i).

**Mutual relevance of *PIK3CA* and *PIK3R1* impactful mutations.** In our cohort, 44% (65/149) harbored *PIK3CA* mutations and 17% (25/149) harbored *PIK3R1* mutations. Notably, a high proportion of the patients (9%) had multiple *PIK3CA* and *PIK3R1* mutations, and the proportion of patients harboring either *PIK3CA* or *PIK3R1* mutations was 42% (Fig. 6a). The proliferation-driving mutations and neutral mutations were classified by the ReMB screens. A total of 50 patients carrying proliferation-driving *PIK3CA* (43/149, 29%) or *PIK3R1* (8/149, 5%) mutations were identified in our cohort. It is worth noting that only one patient in this cohort (1/149, 0.7%) harbored double-driver mutations in both the *PIK3CA* and *PIK3R1* genes (Fig. 6a). In contrast, 49 patients (33%) had single-driver mutations in either *PIK3CA* or *PIK3R1*. The finding that the distributions of driver *PIK3CA* and *PIK3R1* mutations exhibit a mutually exclusive pattern suggests that a single-driver mutation is likely to be sufficient to activate the pathway. A detailed spectrum of driver mutations in *PIK3CA* and *PIK3R1* is shown in Fig. 6b.

The combination of *PIK3CA* H1047R and *PIK3R1* K674R-driver mutations could promote cell proliferation significantly in comparison with a single impactful mutation (Fig. 6c). Western blot analysis showed that p-AKT(Ser473) levels reached the maximum in the *PIK3CA* H1047R and *PIK3R1* K674R combination; meanwhile, the *PIK3CA* H1047R and *PIK3R1* WT

---

**Fig. 4** Validation of impactful *PIK3CA* mutation-induced phenotypes related to proliferation and drug response. **a** The frequencies of impactful *PIK3CA* mutations (proliferation, doxorubicin, and BKM120 responses) in the FUSCC, COSMIC, and TCGA data sets. **b** Schematic representation of the locations of impactful amino acid substitutions in the PI3K/PI4K, C2 PI3K-type, PIK helical, and PI3K-ABD domains of *PIK3CA* (proliferation, black; doxorubicin, red; BKM120, blue). **c** The locations of the impactful mutations in the dimensional structure of the *PIK3CA*-encoded p110α protein are illustrated. Colors denote mutant residues within the corresponding domains in **b**. **d** Growth curves of MCF-10A cells expressing *PIK3CA* wild type and mutants were determined in the absence of EGF using an incucyte cell imaging system. Five wells were measured per condition. The error bars shown are mean ± s.d. of 5 technical replicates from a representative of three independent experiments. (*$P < 0.05$, for Student's *t* test). **e** The effects of *PIK3CA* mutations on acinar morphogenesis by mammary cells cultured on a bed of Matrigel were assessed. Representative images of acini were obtained on Day 11. The dot plot shows the size distribution and the mean (horizontal line) for each cell line ($n \geq 3$, >60 acini per experiment; Mann–Whitney test: ***$P < 0.0001$; n.s. not significant). Scale bar, 100 μm. **f** Activation status of PI3K pathway components in the indicated cells in response to growth factors (EGF and insulin). MCF-10A cells expressing wild-type or mutant *PIK3CA* were seeded in growth medium for 24 h and then cultured in starvation medium (−) or in growth medium (G) containing 20 ng ml$^{-1}$ EGF and 10 μg ml$^{-1}$ insulin for an additional 24 h. The cells were immunoblotted with the indicated antibodies. Western blots were representative of three independent experiments. **g** Activation status of p-AKT (Ser473) after the addition of increasing concentrations of EGF. The cell lines were deprived of EGF and insulin for 24 h and then stimulated for 10 min with increasing concentrations of EGF. Western blots were representative of three independent experiments. **h** Activation status of p-AKT (Ser473) after increasing concentrations of BKM120 were added to cultures of the indicated cell lines. The cell lines were cultured in starvation medium for 24 h and then treated with increasing concentrations of BKM120 for 1 h. The lysates were subjected to western blotting. Western blots were representative of three independent experiments. **i, j** Growth curves of MCF-10A cells expressing *PIK3CA* wild-type and mutants with doxorubicin or BKM120. Five wells were measured per condition. Error bars represent mean ± s.d. derived from three independent experiments. (**$P < 0.01$, for Student's *t* test)

combination resulted in a near-maximal level (Fig. 6d). In one case, we observed that while a single proliferation-driving mutation *PIK3CA* H1047L could promote cell proliferation and upregulate p-AKT signaling, when combined with neutral mutation *PIK3R1* Y504D, its effect on cell proliferation and AKT signaling activation was significantly decreased (Fig. 6e, f). In another case, proliferation-driving mutation *PIK3R1* Q329L at the nSH2 domain boundary could upregulate cell proliferation

and p-AKT(Ser473) level; while the *PIK3CA* Q760L mutation located in linker region between helical and kinase domains exhibited pro-proliferation and AKT signaling effects. However, a combination of *PIK3R1* Q329L and *PIK3CA* Q760L did not increase proliferation and AKT signaling compared to *PIK3R1* Q329L mutation and *PIK3CA* WT (Fig. 6g, h), indicating that the mechanisms for combined mutations in clinical cases will be complicated and challenging.

**a** The frequencies of high-ranking *PIK3R1* mutations in different datasets

| Protein | Domain | *Pro* | *Dox* | *BKM* | FUSCC | COSMIC | TCGA[a] | | Comments[b] |
|---------|--------|-------|-------|-------|-------|--------|------|---|-------------|
| p.E160D | Rho-GAP | x | x | x | 4/149 (2.7%) | 0/3554 (0.0%) | 0/1105 | (0.0%) | Novel, hotspot |
| p.Q329L | NONE | x | x | x | 1/149 (0.7%) | 0/3554 (0.0%) | 0/1105 | (0.0%) | Novel, rare |
| p.D560Y | iSH2 | x | x | x | 0/149 (0.0%) | 1/3554 (0.0%) | 0/1105 | (0.0%) | Rare |
| p.N564D | iSH2 | x | x | x | 1/149 (0.7%) | 0/3554 (0.0%) | 0/1105 | (0.0%) | Novel, rare |
| p.R574T | iSH2 | | x | x | 0/149 (0.0%) | 1/3554 (0.0%) | 0/1105 | (0.0%) | Rare |
| p.K674R | NONE | x | x | | 2/149 (1.3%) | 0/3554 (0.0%) | 0/1105 | (0.0%) | Novel, rare |

[a]Breast invasive carcinoma (TCGA, provisional)
[b]hotspot, mutation frequency ≥2 % in either dataset; rare, mutation frequency <2%; novel, new mutation indentified in the FUSCC cohort. Bold values denote mutation frequency ≥2%

**c**
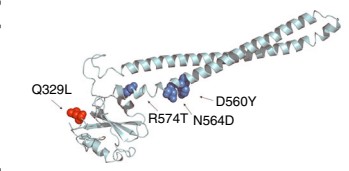

**b** Impactful *PIK3R1* mutations

Proliferation ● Doxorubicin ● BKM120

**d**

EGF(−)

CTRL, WT, K674R, E160D, N564D, Q329L

EGF(+)

CTRL, WT, K674R, E160D, N564D, Q329L

**e** MCF-10A 3-D culture (Day 11)

CTRL, WT, K674R, K674R, N564D

**f**

| | | no EGF | | | | | EGF | | | | |
|---|---|---|---|---|---|---|---|---|---|---|---|
| | CTRL | WT | K674R | E160D | N564D | Q329L | CTRL | WT | K674R | E160D | N564D | Q329L |

Flag-PIK3R1 — 70
0  1.0  1.2  0.8  1.2  0.6    1.0  0.4  1.0  0.6  0.4

p473-Akt — 55
0  0.1  1.0  2.0  2.2  8.0    7.1  8.0  22.5  25.9  39.5  40.1

Akt — 55
1.0  0.7  0.7  1.1  1.0  1.1    0.7  0.7  0.9  0.7  0.9  0.9

GAPDH — 35
1.0  0.9  1.2  1.1  1.2  1.2    1.2  1.1  1.0  1.1  1.0  0.9

**g**

| | CTRL | | | WT | | | K674R | | | E160D | | |
|---|------|---|---|----|---|---|-------|---|---|-------|---|---|
| BKM120 (μM) | DMSO | 0.5 | 4.5 | DMSO | 0.5 | 4.5 | DMSO | 0.5 | 4.5 | DMSO | 0.5 | 4.5 |

p473-Akt — 55
0.2  0.1  0    1  0.2  0    1.0  0.2  0.4    0.9  0.2  0.1

Akt — 55
1.0  1.0  1.0    1.0  1.0  0.9    1.0  1.2  1.2    1.0  1.0  0.9

GAPDH — 35
1.0  1.1  1.1    1.1  1.0  1.1    1.0  1.1  1.1    1.0  1.0  0.9

**h** *PIK3R1*

WT, CRTL, E160D, Q329L, D560Y, N564D, R574T, K674R, D440G

Days after adding doxorubicin (1 nM)

**i** *PIK3R1*

WT, CRTL, E160D, Q329L, D560Y, N564D, R574T, D440G

Days after adding BKM120 (1 μM)

To summarize, PI3K pathway activation can be induced by impactful *PIK3CA* or *PIK3R1* mutations in breast cancer. Most of these mutations occurred in *PIK3CA* and led to gain of function and PI3K activation. A significant proportion of cases had impactful mutations in *PIK3R1* that disturbed the regulatory function of p85α and resulted in pathway activation. Rarer cases with multiple impactful mutations in *PIK3CA* and *PIK3R1* exhibit hyperactivity of PI3K signaling (Fig. 6i).

**Clinical significance of *PIK3CA* impactful mutations**. We analyzed the relationship between mutation status and clinical-pathological characteristics in the TCGA data set. The clinical-pathological characteristics of patients with mutations are as summarized in Table 1. In total, 317 of 977 patients (32.4%) harbored *PIK3CA* missense mutation. Patients diagnosed after the age of 50 years did show a higher rate of *PIK3CA* mutation than younger patients ($P = 0.032$). Breast tumors with mutated *PIK3CA* were more likely to be ER positive ($P < 0.001$) and PR positive ($P < 0.001$). Furthermore, we classified all patients into three categories (impactful mutation, non-impactful mutation, and no mutation). A total of 262 patients (26.8%) had *PIK3CA* impactful mutations, and 55 patients (5.6%) had *PIK3CA* non-impactful mutation. *PIK3CA* impactful mutations were closely associated with hormone receptors status (ER, $P < 0.001$; PR, $P < 0.001$), meanwhile non-impactful mutations were not correlated. As there were insufficient cases in the FUSCC cohort, we observed that *PIK3CA* mutations displayed a trending correlation with ER status ($P = 0.060$, Supplementary Table 13). These findings suggest that the high frequency of *PIK3CA* impactful mutations may contribute to the hyper-activated PI3K signaling in the luminal subtype. Due to the limited number of genetic alteration events in *PIK3R1*, clinical significance analysis was not conducted.

Since impactful mutations reside mainly in the C2, helical and PI4K kinase domains, we analyzed the relationships between the mutated domains and clinical-pathological characteristics (Table 2). When compared against patients who harbored wild-type or non-impactful mutations, patients with helical domain mutation were more likely to be diagnosed after the age of 50 years ($P = 0.025$). Notably, breast cancers with impactful mutations in C2 domain (21 cases, 2.1%), helical domain (105 cases, 10.7%), or PI4K domain (138 cases, 14.1%) were more likely to be ER- and PR-positive ($P = 0.032$, $P < 0.001$, $P < 0.001$, for ER; $P = 0.030$, $P < 0.001$, $P < 0.001$, for PR; respectively). No other significant association was found with mutated *PIK3CA* domains. The sample size in the FUSCC cohort was too small to meet statistical significance (Supplementary Table 14).

We also investigated the relationship between mutation status and survivals in the TCGA data set. Patients carrying *PIK3CA* mutation had similar clinical outcomes to patients without *PIK3CA* mutations ($P > 0.05$, Supplementary Fig. 8a, b). There were no significant differences in disease-free survival (DFS) and overall survival (OS) between the *PIK3CA* wild-type, impactful mutation, and non-impactful mutation carriers (Supplementary Fig. 8c, d). In different molecular subtypes, no significant difference of DFS or OS was observed between *PIK3CA* wild-type and mutated group, or between impactful and non-impactful *PIK3CA* group (Supplementary Fig. 9a–i). Similarly, we observed no statistically significant differences between impactful mutation carriers and other subgroups in the FUSCC cohort (Supplementary Fig. 8e–h).

We further analyzed the correlations between the *PI3KCA* impactful mutations and doxorubicin benefit in the FUSCC cohort. All therapeutic regimen decisions were based on the Chinese Anti-Cancer Association guidelines for the treatment of breast cancer. Doxorubicin or doxorubicin-based treatment was one of the recommended adjuvant chemotherapies for primary breast cancer during 2007–2009. A total of 137 patients underwent a mastectomy and axillary lymph node dissection or breast conservation surgery followed by doxorubicin-based adjuvant chemotherapy (Supplementary Table 13). *PIK3CA* impactful mutations showed no significant effect upon the DFS or OS of patients undergoing doxorubicin treatment (Supplementary Fig. 10a, b), suggesting that *PIK3CA* impactful mutations were unlikely to be associated with doxorubicin efficacy. Of course, the associations between impactful mutations and other therapeutics require further investigations.

## Discussion

Characterization of mutations in cancer can provide insights into tumorigenesis and reveal candidates for targeted therapeutics. Targeted sequencing focusing on specific regions of genome allows us to sequence targeted genes with a high level of coverage without generating significant quantities of off-target data, providing a highly efficient way to find biologically and pathologically relevant variants. Here, we determined the somatic mutation spectrum of the PI3K pathway in Chinese breast cancer patients. Similar to the results of a previous study[2], the mutation rate was highest in *PIK3CA*, in which E542K, E545K, and H1047R mutations were frequently observed. We found that around 50% of breast cancers of the luminal subtype harbor *PIK3CA* mutations, comparable to those in the TCGA data set (45%) and the BOLERO-2 trial population (47.6%)[2,33]. Although mutations within the helical and kinase domains represent the majority of identified *PIK3CA* mutations, we also identified a considerable

**Fig. 5** Validation of impactful *PIK3R1* mutation-induced phenotypes related to proliferation and drug response. **a** The frequencies of impactful *PIK3R1* mutations (proliferation, doxorubicin, and BKM120 responses) in the FUSCC, COSMIC, and TCGA data sets. **b** The locations of the identified impactful amino acid substitutions in the SH3, Rho-GAP, nSH2, iSH2, and cSH2 domains of *PIK3R1* are illustrated (Proliferation, black; doxorubicin, red; BKM120, blue). **c** The locations of the impactful mutations in the dimensional structure of the *PIK3R1*-encoded p85α protein are illustrated. Colors denote mutant residues within the corresponding domains in **b**. **d** Growth curves were determined in MCF-10A cells expressing *PIK3R1* wild-type and mutants in the absence of EGF (up) or presence of EGF (down). The error bars shown are mean ± s.d. of 5 technical replicates from a representative of three independent experiments. (*$P < 0.05$, for Student's *t*test). **e** Effects of *PIK3R1* mutations on mammary acinar morphogenesis. The dot plot shows the size distribution and the mean (horizontal line) for each cell line. ($n \geq 3$, >60 acini per experiment; Mann–Whitney test: ***$P < 0.0001$; n.s. not significant). Scale bar, 100 μm. **f** Activation status of the PI3K pathway in the indicated cells 24 h after growth factor (EGF and insulin) withdrawal. The cells were lysed and immunoblotted with the indicated antibodies. Western blots were representative of three independent experiments. **g** Activation status of p-AKT (Ser-473) after the addition of increasing concentrations of BKM120 to the indicated cell lines. The cell lines were cultured in starvation medium for 24 h and then treated for 1 h with increasing concentrations of BKM120. Western blots were representative of three independent experiments. **h, i** Growth curves of MCF-10A cells expressing *PIK3R1* wild-type and mutants with doxorubicin or BKM120. Five wells were measured per condition. Error bars represent mean ± s.d. derived from three independent experiments. (**$P < 0.01$, for Student's *t* test)

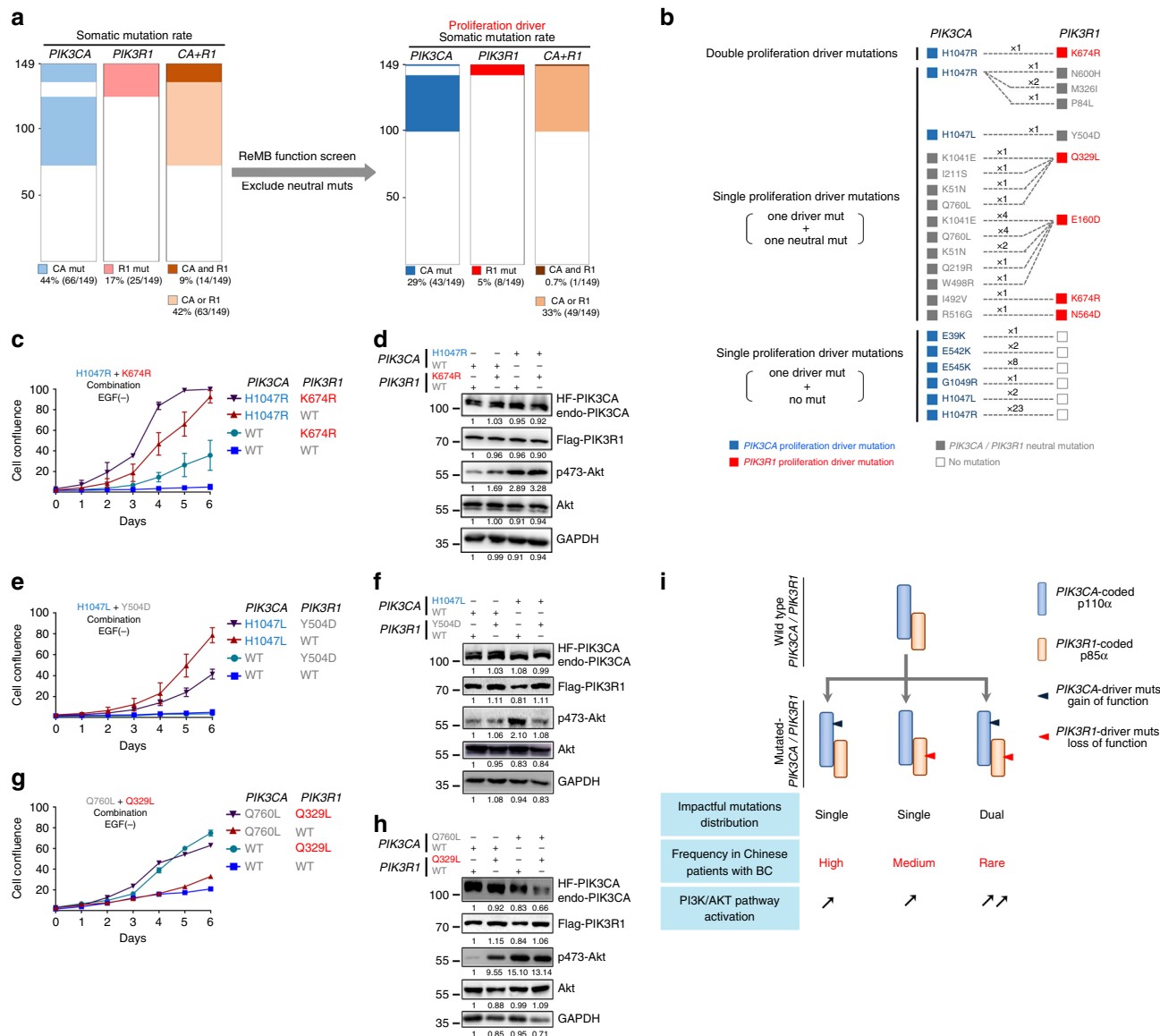

**Fig. 6** Combination spectrum of impactful somatic mutations in *PIK3CA* and *PIK3R1* in breast cancer. **a** A mutually exclusive pattern of proliferation-associated mutations in *PIK3CA* and *PIK3R1* was illustrated in breast cancer by functional screening. Left, somatic mutation frequencies and combined spectrum of *PIK3CA* and *PIK3R1* mutations in the FUSCC cohort; right, proliferation-driver mutation frequencies and the combined spectrum of proliferation-driver mutations after screening. **b** Prevalence of proliferation-driver mutations in individual cancers: *PIK3CA*-driver mut (blue); *PIK3R1*-driver mut (red); *PIK3CA* or *PIK3R1* non-driver mut (gray); no mutation (empty). **c**, **e**, **g** Growth properties of MCF-10A cells expressing a variety of combined *PIK3CA* and *PIK3R1* mutations. Five wells were measured per condition. The error bars indicate ± s.d. derived from three independent experiments. **d**, **f**, **h** The levels of p-AKT at Ser-473 in the indicated cells were analyzed by western blotting. Western blots were representative of three independent experiments. **i** Hypothetical schematic model of dual impactful mutations of *PIK3CA* and *PIK3R1* that regulate the PI3K pathway

number of novel mutations (13 substitutes), some of which resided in other domains. The diverse spectrum of somatic mutational changes in *PIK3CA* reflects some genetic characteristics of Chinese patients. We demonstrated a remarkably higher frequency of *PIK3R1* mutations in the Chinese population than that in the TCGA data set, suggesting that the *PIK3R1* mutation is prevalent and diverse within Chinese patients. We thought there might be three contributors to the difference. First, the sequencing platform differs between the current study and the others. The Ion Torrent PGM sequencing allowed us to sequence target genes with coverage of 1000×, which have been proven to be a reliable and highly efficient targeted sequencing platform[34]. Second, the *PIK3CA* mutations are frequent events in luminal and HER2-positive subtypes, but are less common in TNBCs. There were some differences in the distribution of molecular subtypes by the stratification of race/ethnicity. Chlebowski et al. reported that African-American women had lower incidences of ER-positive (57%) and PR-positive (41%) tumors than white women (ER, 77%; PR, 63%, respectively) in the SEER data set[35]. Our previous study reported that the proportion of hormone-dependent breast cancer was relatively lower in Chinese females (50–60%) compared to females from Western countries[36,37], which may also contribute to the distinctive mutation profile observed in Chinese population. Third, we recognized that a limited number of cases from one center may be insufficient to represent the overall characteristics of mutation pattern in

**Table 1 Clinicopathological variables and *PI3KCA* mutations in TCGA database**

| Variables | Number of patients (%) | PIK3CA mutation | | P value | PIK3CA impactful mutation | | | P1 | P2 |
|---|---|---|---|---|---|---|---|---|---|
| | | Neg (%) | Pos (%) | | No mutation | Neg (%) | Pos (%) | | |
| Total | 977 | 660 (67.6) | 317 (32.4) | | 660 (67.6) | 55 (5.6) | 262 (26.8) | | |
| Age (median 58, range 26–90) | | | | **0.032** | | | | 0.186 | 0.060 |
| ≤50 years | 294 (30.1) | 213 (21.8) | 81 (8.3) | | 213 (21.8) | 13 (1.3) | 68 (7.0) | | |
| >50 years | 683 (69.9) | 447 (45.8) | 236 (24.1) | | 447 (45.8) | 42 (4.3) | 194 (19.8) | | |
| Menopausal status | | | | 0.468 | | | | 0.186 | 0.775 |
| Premenopause | 308 (31.5) | 213 (21.8) | 95 (9.7) | | 213 (21.8) | 13 (1.3) | 82 (8.4) | | |
| Postmenopause | 669 (68.5) | 447 (45.8) | 222 (22.7) | | 447 (45.8) | 42 (4.3) | 180 (18.4) | | |
| Tumor size | | | | 0.622 | | | | 0.246 | 0.191 |
| ≤2 cm | 258 (26.4) | 168 (17.2) | 90 (9.2) | | 168 (17.2) | 13 (1.3) | 77 (7.9) | | |
| >2, ≤5 cm | 564 (57.7) | 386 (39.5) | 178 (18.2) | | 386 (39.5) | 36 (3.7) | 142 (14.5) | | |
| >5 cm | 155 (15.9) | 106 (10.8) | 49 (5.0) | | 106 (10.8) | 6 (0.6) | 43 (4.4) | | |
| LN status | | | | 0.563 | | | | 0.897 | 0.479 |
| Negative | 477 (48.8) | 318 (32.5) | 159 (16.3) | | 318 (32.5) | 26 (2.7) | 133 (13.6) | | |
| Positive | 500 (51.2) | 342 (35.0) | 158 (16.2) | | 342 (35.0) | 29 (3.0) | 129 (13.2) | | |
| ER status | | | | **<0.001** | | | | 0.193 | **<0.001** |
| Negative | 258 (26.4) | 212 (21.7) | 46 (4.7) | | 212 (21.7) | 13 (1.3) | 33 (3.4) | | |
| Positive | 719 (73.6) | 448 (45.9) | 271 (27.7) | | 448 (45.9) | 42 (4.3) | 229 (23.4) | | |
| PR status | | | | **<0.001** | | | | 0.198 | **<0.001** |
| Negative | 357 (36.5) | 287 (29.4) | 70 (7.1) | | 287 (29.4) | 19 (1.9) | 51 (5.2) | | |
| Positive | 620 (63.5) | 373 (38.2) | 247 (25.3) | | 373 (38.2) | 36 (3.7) | 211 (21.6) | | |
| HER-2/neu status | | | | 0.37 | | | | 0.869 | 0.278 |
| Negative | 798 (81.7) | 534 (54.7) | 264 (27.0) | | 534 (54.7) | 44 (4.5) | 220 (22.5) | | |
| Positive | 179 (18.3) | 126 (12.9) | 53 (5.4) | | 126 (12.9) | 11 (1.1) | 42 (4.3) | | |

*Note*: Based on Pearson $\chi^2$ test, for which *P* is based on Fisher's exact test; Bold values denote *P* value <0.05. *P1*, no mutation vs. non-impactful mutation; *P2*, no mutation vs. impactful mutation
*ER* estrogen receptor, *HER-2* human epidermal growth factor receptor 2, *PR* progesterone receptor, *Neg* non-impactful mutation, *Pos* impactful mutation, *LN* lymph node

Chinese patients. Further analysis of somatic mutations in a large cohort will be necessary to achieve a comprehensive mutation profile for Chinese breast cancer patients.

Regulation of mutation-induced PI3K activation has long been thought to involve a single element, either a *PIK3CA* gain-of-function mutation or a *PIK3R1* loss-of-function mutation. However, the picture becomes far more complex when one considers that multiple mutations in *PIK3CA* and *PIK3R1* may cooperatively regulate activation. In this study, we found that the multiple mutation rate for PI3K family members is significantly higher than that reported in data sets obtained from Western populations. Considering that our *PIK3CA* mutation frequency was largely consistent with data from TCGA data set, the 17% *PIK3R1* mutation rate may contribute to the high proportion of multiple mutations. Here, we obtained a more detailed functional classification of *PIK3CA* and *PIK3R1* mutations and explored their mutual relevance. Most cells with hyperactive PI3K signaling had either a *PIK3CA*-driver mutation or a *PIK3R1*-driver mutation; in contrast, multiple-driver mutations were rare. This finding demonstrates that the driver mutations in *PIK3CA* and *PIK3R1* exhibit a mutually exclusive pattern, suggesting that pathway activation is likely to be dominated by single-driver mutations. We investigated the rare case that harbored two driver mutations. The combination of *PIK3CA* H1047R plus *PIK3R1* K674R cooperatively promotes proliferation and PI3K activation, supporting a previous hypothesis that multiple mutations in different components of the PI3K pathway might cooperate to cause efficient transformation[38]. Furthermore, the neutral mutation *PIK3R1* Y504D can partially reverse *PIK3CA* H1047L-induced proliferation and AKT activation, implying that truly neutral mutations may not as common as we once thought. These observations suggest a synergetic model where single or multiple-driver mutations in *PIK3CA* and/or *PIK3R1* are involved in the regulation of PI3K activation in breast cancer (Fig. 6i).

We noticed that the driver mutations that propel growth of cells were also found to be related to resistance to doxorubicin and/or BKM120. We speculated that these activating PI3K mutations may have some general effects upon both proliferation and chemotherapy responses. Doxorubicin, an anthracycline antibiotic, remains the most effective agent for breast cancer patients in clinical practice[39,40]. Our results suggested that doxorubicin might exhibit its cytotoxic effect in a PI3K-pathway-independent manner. BKM120 is a small molecule pan-Class I PI3K inhibitor that inhibits p110α, p110β, p110γ, and p110δ[41,42]. BKM120 decreases the cellular levels of p-Akt in mechanistic models and relevant tumor cell lines and shows a strong anti-proliferative effect. Inhibition of PI3K is a potentially attractive strategy for breast cancer treatment, and BKM120 has already entered clinical trials[43,44]. However, the responses of PI3K-driver mutations to BKM120 treatment remains to be verified. The tumor stabilization and partial tumor responses have been observed in *PIK3CA* mutant breast cancers treated with BKM120; meanwhile, dramatic regressions of *PIK3CA* mutant tumors are not typical[45,46]. It is hard to conclude that BKM120 is a specific inhibitor for *PIK3CA*-driver mutation. A definitive conclusion awaits further detailed investigation of functionally well-characterized mutations in preclinical investigations and clinical trials with more patients. Additionally, the combination of CDK 4/6 inhibitors and PI3Ki, a novel strategy that may increase the efficacy of PI3Ki in tumors with genetic alterations, may be promising to overcome *PIK3CA* mutant-induced resistance[47].

The clinical significance of *PIK3CA* mutations in breast cancer remains complicated and controversial in literatures. Sobhani et al. suggested that mutated *PIK3CA* represents an independent negative prognostic factor in breast cancer[48]; while several studies indicated that mutations in *PIK3CA* were associated with favorable prognostic factor in unsorted breast cancer[49–52]. Notably, the detailed stratification of molecular subtypes in recent years has helped reduce controversy on the topic. (1) Luminal subtype: in

**Table 2 Clinicopathological variables and the domains with *PIK3CA* impactful mutation in TCGA database**

| Variables | Domains with *PIK3CA* impactful mutation | | | | | | | | | |
| --- | --- | --- | --- | --- | --- | --- | --- | --- | --- | --- |
| | Neg (%) | ABD (%) | RBD (%) | C2 (%) | Helica (%) | PI4K (%) | Linker region (%) | P1 | P2 | P3 |
| Total | 716 (73.1) | 0 (0) | 0 (0) | 21 (2.1) | 105 (10.7) | 138 (14.1) | 0 (0) | | | |
| Age (median 58, range 26–90) | | | | | | | | 0.535 | **0.025** | 0.255 |
| ≤50 years | 227 (23.2) | 0 (0) | 0 (0) | 8 (0.8) | 22 (2.2) | 37 (3.8) | 0 (0) | | | |
| >50 years | 489 (49.9) | 0 (0) | 0 (0) | 13 (1.3) | 83 (8.5) | 101 (10.3) | 0 (0) | | | |
| Menopausal status | | | | | | | | 0.535 | 0.800 | 0.644 |
| Premenopause | 227 (23.2) | 0 (0) | 0 (0) | 8 (0.8) | 32 (3.3) | 41 (4.2) | 0 (0) | | | |
| Postmenopause | 489 (49.9) | 0 (0) | 0 (0) | 13 (1.3) | 73 (7.4) | 97 (9.9) | 0 (0) | | | |
| Tumor size | | | | | | | | 0.051 | 0.769 | 0.388 |
| ≤2 cm | 181 (18.5) | 0 (0) | 0 (0) | 10 (1) | 30 (3.1) | 39 (4) | 0 (0) | | | |
| >2, ≤5 cm | 423 (43.2) | 0 (0) | 0 (0) | 10 (1) | 59 (6) | 73 (7.4) | 0 (0) | | | |
| >5 cm | 112 (11.4) | 0 (0) | 0 (0) | 1 (0.1) | 16 (1.6) | 26 (2.7) | 0 (0) | | | |
| LN status | | | | | | | | 0.639 | 0.641 | 0.374 |
| Negative | 344 (35.1) | 0 (0) | 0 (0) | 9 (0.9) | 53 (5.4) | 72 (7.3) | 0 (0) | | | |
| Positive | 372 (38) | 0 (0) | 0 (0) | 12 (1.2) | 52 (5.3) | 66 (6.7) | 0 (0) | | | |
| ER status | | | | | | | | **0.032** | **<0.001** | **<0.001** |
| Negative | 225 (23) | 0 (0) | 0 (0) | 2 (0.2) | 11 (1.1) | 20 (2) | 0 (0) | | | |
| Positive | 491 (50.1) | 0 (0) | 0 (0) | 19 (1.9) | 94 (9.6) | 118 (12) | 0 (0) | | | |
| PR status | | | | | | | | **0.030** | **<0.001** | **<0.001** |
| Negative | 306 (31.2) | 0 (0) | 0 (0) | 4 (0.4) | 21 (2.1) | 27 (2.8) | 0 (0) | | | |
| Positive | 410 (41.8) | 0 (0) | 0 (0) | 17 (1.7) | 84 (8.6) | 111 (11.3) | 0 (0) | | | |
| HER-2/neu status | | | | | | | | 0.576 | 0.800 | 0.197 |
| Negative | 579 (59.1) | 0 (0) | 0 (0) | 18 (1.8) | 86 (8.8) | 118 (12.0) | 0 (0) | | | |
| Positive | 137 (14) | 0 (0) | 0 (0) | 3 (0.3) | 19 (1.9) | 20 (2.1) | 0 (0) | | | |

*Note*: Based on Pearson $\chi^2$ test, for which *P* is based on Fisher's exact test; Bold values denote *P* value <0.05. *P1*, Neg vs. impactful mutation in C2 domain; *P2*, Neg vs. impactful mutation in Helica domain; *P3*, Neg vs. impactful mutation in PI4K kinase domain
*ER* estrogen receptor, *HER-2* human epidermal growth factor receptor 2, *PR* progesterone receptor, *Neg* no mutation or non-impactful mutation, *LN* lymph node

early breast cancers, *PIK3CA* mutations were more frequent in low-risk luminal BCs (lower grade, less lymph node involvement, and PR positivity)[53–55]. Sabine et al. suggested *PIK3CA* mutations were associated with favorable outcomes for luminal subtype in TEAM clinical trial[53]. In advanced ER-positive breast cancers with acquired resistance to endocrine therapy, *PIK3CA* mutations may behave as a mechanism of anti-estrogen resistance[43,56]. The importance of the simultaneous inhibition of the PI3K/mTOR and ER pathway is supported by the results from recent BELLE-2 and BOLERO-2 clinical trials[33,57]. It remains possible and even likely that mechanisms of de novo endocrine resistance in early cancer differ from those in progressing cancer that achieve endocrine resistance by tumor evolution. (2) HER-2 subtype: in advanced HER-2 subtype patients, *PIK3CA* mutations were associated with poorer clinical outcome, as inferred from the biomarker analyses in CLEOPATRA clinical trial and other study[58,59]. Additionally, Loibl et al. and Majewski et al. both suggest that *PIK3CA* mutations predicted poor pathologic complete response rate in HER2 subtype patients with neoadjuvant anti-HER2 therapies (NeoALTTO and other trials)[60,61]. These results suggest *PIK3CA* activating mutations might drive resistance to anti-HER2 therapies. (3) TNBC: Takeshita et al. reported that the presence of *PIK3CA* mutations was significantly correlated with phosphorylated androgen receptor which is an independent favorable prognostic factor of TNBC[62]. A continuous molecular profiling investigation on patients will help to elucidate the prognostic/predictive role of *PIK3CA* mutations in TNBC.

We reported strong correlations between the distributions of *PIK3CA* impactful mutations and hormone receptor positivity in breast tumors, whereas this relationship did not seem to hold true for non-impactful mutations. Unfortunately, impactful *PIK3CA* mutations did not seem to be related to the clinical outcome of breast cancer patients; and they do not have prognostic

significance in the subsequent stratification of molecular subtypes. One possibility may be the heterogeneity of patients' clinical characteristics and treatments in TCGA data set. Many studies reporting the prognostic significance of *PIK3CA* mutations used data from clinical trials, thus excluding the effect of stage and treatment heterogeneity. Another possibility is that *PIK3CA* mutations may play various roles in tumorigenesis and drug resistance. *PIK3CA* impactful mutations were proved to confer oncogenic activity in malignant transformation among normal mammary epithelial cells. The characterizations of PI3K mutations in luminal and HER-2 subtypes would be necessary to uncover their potentials in conferring endocrine or anti-HER2 resistances.

The recognition that cancer-associated mutations contribute to explore attractive targets for cancer therapeutics. However, it encounters the fact that our current ability to identify functional mutations is limited and labor-intensive. Here, we developed a ReMB library in which we barcoded clones bearing each individual mutation to permit fast and accurate functional discovery. The pooled library consists of different mutant, wild-type and control expression vectors with exact identical amount to eliminate the difference in expression level. Using viral transduction, the multiplicity of infection (MOI) can be kept low so that most cells receive a single virus that is stably integrated. This strategy is sought to be a reliable method that allow multiplex analysis of phenotypic outputs on a large-scale pooled screen[63]. Similar strategies are widely applied in pooled RNAi[64,65] or CRISPR[66,67] screens. This method provides a highly standardized and reproducible approach to exploring the biological effects of mutations in a high-throughput manner.

Together, we successfully created an onco-genotype spectrum of *PIK3CA* and *PIK3R1* impactful mutations, and explored their clinical associations in breast cancer. We hope that these

genotype-based mutation spectra will provide more precise annotations of tumor genomic alterations and contribute to the development of personalized treatments for breast cancer. Overall, this work provides rationales that can synergize with existing therapeutics to produce a path toward genetically informed therapies.

## Methods

**Patients and specimens**. In this study, we retrospectively obtained a total of 149 pathologically confirmed primary breast cancer samples from FUSCC to examine somatic mutation via targeted sequencing. Briefly, a total of 563 samples that were available for frozen fresh tissues were collected from 3252 patients who were diagnosed as breast cancer at the Department of Breast Surgery in FUSCC between June 2007 and December 2009. In this study, ER, PR, HER2, and Ki67 index were examined by IHC staining at the Department of Pathology in FUSCC. Additionally, the patients with HER2 expression status (IHC, score = 2) were subjected to florescence in situ hybridization (FISH) screening for HER2 gene amplification. The HER2 overexpression was defined as FISH positive or an IHC staining score = 3. The patients were classified into four subtypes: luminal A, luminal B, HER2-enriched, and triple-negative breast cancer (TNBC) based on their estrogen receptor (ER), progesterone receptor (PR), human epidermal growth factor receptor 2 (HER2) status, and Ki67 index according to the guidelines of the St. Gallen International Breast Cancer Conference (2011)[68]. Subsequently, we used the complete random sampling method to collect 107 (71.8%) luminal-like subtype cases, 17 (11.4%) HER2-enriched subtype cases, and 25 (16.8%) triple-negative subtype cases to perform targeted sequencing.

In this study, we revalidated the ER, PR, and HER2 status of all specimens which were subjected to amplicon sequencing, following the standard procedures and guidelines. Briefly, the HE staining of tumor tissue slides were retrieved and reviewed for determining breast carcinoma by two independent pathologists. The ER, PR, and HER2 IHC staining procedures were performed on a Ventana Benchmark automated immunostainer (Tucson, Arizona, USA) by the standard streptavidin-biotin staining method (Antibodies: ER, Roche Ventana, Clone SP1; PR, Roche Ventana, Clone IE2; HER2, Roche Ventana, Clone 4B5)[69]. Mammary tumors were considered positive for ER or PR if strong immunoreactivity was observed in more than 1% of tumor nuclei, according to the 2010 ASCO/CAP guidelines[70]. HER2 status was initially evaluated by IHC on a scale of 0–3, combining the intensity of membranous staining and the percentage of staining of invasive tumor cells, according to the 2013 ASCO/CAP guidelines[71]. Cases with scores of 3+ were identified as positive. Tumors with HER2 expression status (IHC, score equals to 2+) were further subjected to florescence in situ hybridization (FISH) test to determine the HER2 gene amplification with the FDA-approved PathVysion HER2/Neu DNA Probe Kit (Abbott Molecular, Abbott Park, IL, USA). The HER2 positivity subgroup was defined as FISH positive or IHC staining score equals to 3+. Representatives of IHC staining for ER, PR, and HER2 were illustrated in Supplementary Fig. 11.

The patients in this cohort underwent routine follow-up, and clinical outcomes were obtained for 146 of the cases. The most recent follow-up update was performed in October 2015. Ethical approval was obtained from the Institutional Review Board of FUSCC, and each participant gave written informed consent to participation in this research.

**Cell lines and cell culture**. The HEK293T (293T) cell line was obtained from the Shanghai Cell Bank Type Culture Collection Committee (CBTCCC, Shanghai, China) in 2014. The cells were cultured in DMEM (Gibco, Gaithersburg, MD, USA) supplemented with 10% fetal bovine serum (FBS) (Gibco), and 1% penicillin/streptomycin (Invitrogen, Carlsbad, CA, USA). The normal immortalized mammary epithelial cell line MCF-10A was obtained from CBTCCC in 2015 and cultured in growth medium consisting of DMEM/F-12 (Invitrogen) supplemented with 5% donor horse serum (HS) (Invitrogen), 20 ng ml$^{-1}$ EGF (Invitrogen), 10 µg ml$^{-1}$ insulin (Invitrogen), 0.5 µg ml$^{-1}$ hydrocortisone (Sigma-Aldrich, St. Louis, MO, USA), 1 ng ml$^{-1}$ cholera toxin (Sigma-Aldrich), and 1% penicillin/streptomycin (Invitrogen). The hTERT-immortalized normal human mammary epithelial cell line (HMEC) (Lonza Group Ltd, Walkersville, MD, USA) was maintained in complete growth medium same as that of MCF-10A. The identities of the cell lines were confirmed by Shanghai Cell Bank Type Culture Collection Committee (CBTCCC, Shanghai, China) using DNA profiling (short tandem repeat, STR). The cell lines were subjected to routine cell line quality examination (e.g., by morphology and mycoplasma testing) by HD Biosciences every 3 months.

**DNA preparation and Ion Torrent sequencing**. Genomic DNA was obtained from frozen breast cancer samples or mammary cell lines and extracted using QIAamp DNA Mini Kits (Qiagen, Hilden, Germany). Paired blood DNA was extracted from peripheral blood leukocytes using QIAamp Blood kits (Qiagen). To target gene exon sequences, 10 ng of DNA was amplified by the polymerase chain reaction (PCR) using the Ion AmpliSeq custom Panel Primer Pool (Life Technologies, Gaithersburg, MD). Then, the samples were barcoded using an Ion Xpress Barcode Adapters 1–16 Kit (Life Technologies) and sequenced using an Ion

PGM 200 Sequencing Kit on Ion 316 chips (Life Technologies). Data from the PGM runs were initially processed using the Ion Torrent platform-specific pipeline software Torrent Suite to generate sequence reads, trim adapter sequences, and filter and remove poor signal-profile reads. Initial variant callings from the Ion AmpliSeq sequencing data were generated using Torrent Suite Software v3.2. The Ion Torrent PGM sequencing was performed by Life Technologies (Shanghai, China).

**Sanger sequencing and pyrosequencing**. PIK3CA and PIK3R1 mutations were validated using Sanger sequencing for paired blood DNA to exclude germline mutations in all cases. For pyrosequencing, PCR products were processed and submitted to a pyrosequencing assay on a Pyro Mark Q24 platform (Qiagen, Hilden, Germany). Pyrosequencing was able to detect an allele frequency difference of less than 5% between pools, indicating that this method may be sensitive enough for use in mutation detection[72]. We used pyrosequencing to validate somatic mutations of PIK3CA and PIK3R1 in 41 malignant tissues. Twenty-three (23/28, 82%) somatic mutations in 41 cases could be validated successfully (Supplementary Fig. 12), suggesting that the amplicon sequencing using Ion Torrent PGM platform is a reliable method to detect low allele frequency mutations. In this study, we included the mutations with allele frequency >3% via PGM platform into the following analysis.

**Plasmid construction and mutagenesis**. The 30-bp annealed oligonucleotide pairs were ligated into a retroviral pDEST-HA-Flag backbone. We constructed a set of vectors that contained various barcodes inserted between the attB2 region and the mouse phosphoglycerate kinase 1 promoter (PGK). The human full-length PIK3CA or PIK3R1 coding sequence (CDS) was cloned into the pENTR vector using BP Clonase (Life Technologies). Site-directed mutagenesis was performed using a QuikChange II site-directed mutagenesis kit (Agilent Technologies, Palo Alto, CA, USA) to generate a set of PIK3CA and PIK3R1 mutants. The LR recombination reaction was performed using LR Clonase (Life Technologies) to generate sets of retrovirus-based pMSCV-barcoded-PIK3CA and -PIK3R1 mutation vectors. Each mutant was labeled with a unique barcode. HA-Flag-tagged PIK3CA or PIK3R1 expression is driven by the murine stem cell virus long-terminal-repeat promoter. The pooled PIK3CA-ReMB library contains 104 PIK3CA mutations, 4 PIK3CA wild-type constructs, and 3 controls with unique barcodes. Similarly, the PIK3R1-ReMB library consists of 27 PIK3R1 mutations, 3 PIK3R1 wild-type constructs, and 3 controls.

**Virus production and cell transduction using the ReMB library**. Retroviruses were produced by co-transfecting the pooled retroviral PIK3CA- or PIK3R1-ReMB libraries with MSCV-Gag-Pol envelopes and MSCV-VSVG packaging plasmids into 293T cells using Lipofectamine 2000 (Life Technologies). To achieve an MOI of 0.3–0.5, titrations were performed and the infection efficacy of each cell line was determined. For cell transduction of the ReMB library, MCF-10A and HMEC cells were infected with an optimal volume of virus in medium containing 8 µg ml$^{-1}$ polybrene (Sigma-Aldrich). Beginning at 36 h after infection, cells were selected by applying 2 µg ml$^{-1}$ puromycin (InvivoGen, San Diego, CA, USA) for 5 days.

**IC$_{50}$ assays and proliferation assays**. For the IC$_{50}$ assays, cells in the logarithmic growth stage were plated in 96-well plates. Doxorubicin and BKM120 were purchased from Selleck Chemicals. After the cells were allowed to adhere overnight, the medium was replaced with medium containing serially diluted concentrations of the reagent for 5 days. The IC$_{50}$ was calculated using GraphPad Prism (GraphPad Software, Inc). For the cell proliferation assays with drugs, cells were incubated in 96-well plates under the indicated drug conditions, and confluency was measured using an Incucyte Life Cell Imaging Device (Essen Bioscience).

**ReMB library screening for cell proliferation and drug responses**. For cell proliferation screening, $6 \times 10^6$ MCF-10A and HMEC cells were transduced with the PIK3CA mutation library. Puromycin (2 µg ml$^{-1}$) was added to the cells 36 h after transduction, and selection was continued for 5 days. A transduction efficiency of 30% was achieved. On Day 5, the cells were trypsinized and divided into four samples. Two of these samples contained a minimum of $4 \times 10^5$ cells per sample and were used for cell proliferation screening; biological replicates containing a minimum of $4 \times 10^5$ cells per sample were frozen for genomic DNA analysis. The medium was replaced with fresh medium every 2–3 days, and the cells were passaged when they reached full confluence. The cells were harvested at 7 and 14 days after proliferation screening.

For the compound response screens, $2 \times 10^7$ MCF-10A and HMEC cells were transduced with the PIK3CA mutation library at an MOI of ~0.3. Puromycin (2 µg ml$^{-1}$) was added to the cells 36 h after transduction, and selection was continued for 5 days. On Day 5, the cells were trypsinized and divided into six samples. Two "drug condition" samples, each of which contained a minimum of $1.2 \times 10^6$ cells, were subjected to treatment with various compounds to determine their IC50s for those compounds. Two controls with a minimum of $4 \times 10^5$ cells per sample were cultured in medium containing an equal volume of DMSO (Sigma-Aldrich). The other replicates, which contained a minimum of $4 \times 10^5$ cells per sample, were frozen for later baseline cell genomic DNA analysis. The cells were passaged or

given fresh medium containing either the drug or DMSO every 3–4 days. The cells were harvested at 7 and 14 days after the first addition of the test compound. Proliferation screening and compound resistance screening after transduction with the PIK3R1-ReMB library were performed similarly, with $1.2 \times 10^5$ cells in each replicate.

**Two-step PCR and MiSeq sequencing for barcode deconvolution**. The frozen cell pellets were thawed, and genomic DNA (gDNA) was extracted. Two-step PCR was performed using NEBNext High-Fidelity 2× PCR Master Mix (NEB, Herts, UK). Amplification was performed using 15 cycles for the first PCR and 15 cycles for the second PCR. For the first PCR, the amount of genomic DNA in each sample was calculated to achieve 2000× coverage of the mutation library; this corresponded to 1.5 μg of DNA per sample for the PIK3CA-ReMB library and to 450 ng of DNA for the PIK3R1-ReMB library (defined as 6.6 pg of genomic DNA per cell). For each sample, we performed five separate 50-μl reactions for the PIK3CA-ReMB library and one 50-μl reaction for the PIK3R1-ReMB library. We then mixed the resulting amplicons. The following primers were used to amplify the barcode region in the first round of PCR:

F1-PIK3CA: TCGTCGGCAGCGTCAGATGTGTATAAGAGACAGaaatggattggatcttccac (PIK3CA gene-specific primer);

F1-PIK3R1: TCGTCGGCAGCGTCAGATGTGTATAAGAGACAGagcctacccagtatatgca (PIK3R1 gene-specific primer);

R1 GTCTCGTGGGCTCGGAGATGTGTATAAGAGACAGtgggaaaagcgcctcccta (construct primer).

A second PCR was performed to add the Illumina adapters and to index the samples. The second PCR was performed in five separate 50-μl reactions, with each reaction containing 10 μl of the mixed product from the first PCR. The following primers, each of which included an 8-bp index for multiplexing different biological samples, were used for the second PCR:

F2: AATGATACGGCGACCACCGAGATCTACAC (8-bp index) TCGTCGGCAGCGTC;

R2: CAAGCAGAAGACGGCATACGAGAT (8-bp index) GTCTCGTGGGCTCGG.

The amplicons that resulted from the second PCR were purified using AMPure XP beads (Beckman Coulter, Nyon, Switzerland), quantified using Qubit2.0 (Life Technologies), mixed and sequenced using MiSeq (Illumina) at the Chinese National Human Genome Center in Shanghai (CHGCS).

**Sequencing data analysis**. The raw FASTQ files were demultiplexed using Geneious 7.0 (Biomatters Inc, Auckland, New Zealand) and processed so that they contained only the unique barcode sequence. To align the processed reads to the library, the designed barcode sequences from the library were assembled into a mapping reference sequence. The reads were then aligned to the reference sequence using the Map to Reference function in Geneious 7.0. After alignment, the number of uniquely aligned reads for each library sequence was calculated. The number of reads for each unique barcode for a given sample was normalized as follows: normalized read counts per unique barcode = reads per barcode/total reads for all barcodes in the sample $\times 10^5 + 1$. After normalization, we calculated an ES using $\log_2$ Day 14/Day 0 or Day 7/Day 0 ratios of the normalized read counts for each replicate. The ES value represents the change in relative abundance that was observed in a mutant relative to the difference in abundance in the initial and final samples for the control, with a positive ES indicating enrichment in the proliferation or compound response assays. Clustering of the two cell lines revealed the presence of consistent mutations in the proliferative and drug response screens. Hierarchical clustering analysis and heatmap generation were performed using MeV4.0 (Dana-Farber Cancer Institute, Boston, MA, USA). The Avg ES indicates the average of two biological replicates. Impactful mutations were indicated when the mean Avg ES were greater than 1 for PIK3CA and 0.8 for PIK3R1 in the MCF-10A and the HMEC cell lines.

**Morphogenesis assay**. Three-dimensional cultures of MCF-10A cells were grown on basement membranes. Briefly, $5 \times 10^3$ cells were resuspended in medium containing 2% growth factor reduced Matrigel (BD Biosciences, San Jose, CA, USA), 2% HS (Invitrogen), 5 ng ml$^{-1}$ EGF (Invitrogen), and the other additives mentioned above; the solution was then seeded on chamber slides coated with growth factor reduced Matrigel. The medium was changed every 2–3 days. Photomicrographs of representative fields were taken on Day 11 using a Leica DMI 6000B microscope.

**Western blot analysis**. Whole cell lysates were resolved using T-PER Tissue Extraction Reagent (Thermo Fisher Scientific Inc., MA, USA) with a complete ethylenediaminetetraacetic acid (EDTA)-free protease inhibitor and phosphatase inhibitor cocktail (Selleck Chemicals). Immunoblotting was performed using a standard method. The following primary antibodies and dilutions were used: anti-human p110α (#4249, 1:1000), anti-human p-AKT(Ser473) (#4060, 1:1000), anti-human AKT (#4691, 1:1000), anti-human p85 (#4257, 1:1000), anti-human ERK1/

2 (#4695, 1:1000), anti-human p-ERK1/2 (#4370, 1:1000), anti-glyceraldehyde 3-phosphate dehydrogenase (GAPDH) (#5174, 1:2000) antibodies from Cell Signaling Technology and anti-HA antibody (#H3663, 1:1000) from Sigma-Aldrich. The quality of the gel loading and the transfer processes was assessed by immunostaining the blots with the GAPDH antibody. The densitometry analysis was performed using ImageJ software (NIH, Bethesda, MD). Uncropped scans of important blots are provided as Supplementary Fig. 13 in the Supplementary Information.

**Statistical and bioinformatics analysis**. All experiments were performed with three independent experiments, and data are presented as the mean of biological replicates unless otherwise indicated. Error bars represent the s.d. from the mean, unless otherwise indicated. Correlations between clinical-pathological parameters and gene mutations were tested using the Chi-square test. Data points were compared using unpaired two-tailed Mann–Whitney tests, as indicated in the 3D culture assay. For TCGA data sets, the data were downloaded from the TCGA Data Coordination Center (DCC) or from the results of the TCGA Firehose pipeline at the Broad Institute (http://gdac.broadinstitute.org/). Cumulative survival time was calculated using the Kaplan–Meier method, which was analyzed using the log-rank test. Statistical significance was inferred at $P < 0.05$. All statistical analyses were performed using the GraphPad Prism software and SPSS Software version 17.0 (SPSS, Chicago, IL, USA).

**Data availability**. The ReMB screenings sequencing data have been deposited in the NCBI Sequence Read Archive (SRA) under Bioproject (SRA accession number: SRP132614). The authors declare that all relevant data of this study are available within the article or from the authors.

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

## Acknowledgements

This study was supported by a grant from the Ministry of Science and Technology of China (MOST2016YFC0900300, National Key R&D Program of China), a grant from the National Natural Science Foundation of China (81572583 and 81672601), a grant from the Shanghai Committee of Science and Technology Funds (15410724000 and 15411953300) and a grant from the Pudong's Science and Technology Development Fund (Pkj-z04). The funders had no role in the study design, collection and analysis of the data, decision to publish, or manuscript preparation. We thank all of the subjects who were included in this study for their participation. We thank Dr. Bin Wang for critically reading the manuscript.

## Author contributions

L.C. and X.H. designed the experiments. L.C., X.H., L.Yan. and L.Yao. developed the methodology and carried out most of the experiments. L.C., L.Yan., L.Yao., X.Y.K., S.L., and F.Q. assisted in acquisition of data, including animals and patients management. L. C., L.Yan., L.Yao., Y.R.L., Z.G.C., X.Y.Z., W.T.Y., S.L.Z., J.X.S., W.H., and X.H. assisted with analysis and interpretation of data. L.C., L.Yao., and X.H. helped to organize the data. L.C., L.Yan., W.J.Z., and X.H. wrote, reviewed, and revised the manuscript. X.H. and Z.M.S. supervised the research.

## Additional information

**Competing interests:** The authors declare no competing interests.

