## [Peer Review File · Nature Communications]

Reviewers' comments:

Reviewer #1 (Remarks to the Author):

The manuscript by Chen, et al examines PI3K pathway mutations identified in a Chinese breast cancer cohort. The frequency and the functional consequences of the range of mutations relative to proliferation and drug sensitivity are reported. The work has some interest particularly in the use of the recombination-based mutation barcoding (ReMB) libraries in the function screening. But I have specific concerns about the data, data analysis, and conclusions:

1. The authors wish to claim that Chinese patients have a different mutational profile than those published from caucasian cohorts. However, frequency comparisons across different platforms with different coverage and different analytical algorithms are difficult and subject to technical biases. In looking at the mutational frequency of another cancer gene, TP53, though comparable with other reports in general, was remarkably low in the triple negative breast cancers in this cohort. The Fudan group registers probably around 15-20% mutation rate in TP53 in TNBC, whereas almost all studies report a much higher rate (~60-80%). This suggests either that their diagnostics are off, that their sequencing/algorithms are not comparable with other studies, or that Chinese have much fewer TP53 mutations than any other population. This of course raise the question of comparability of frequency determinations. I would recommend that the authors do not stress this point.

2. The most interesting aspect of this study is the ReMB screen which is amongst the most complete for the PI3K pathway to date. But unfortunately, the authors chose not to spend much energy in the detailed computational and statistical analysis of this screen thus leaving many unanswered questions. For example, what are the domains that are mutated in PIK3CA showing specific phenotypes, and which are not effective? How has this information improved the functional prediction for this pathway? What are the mutations that render cells resistant to doxorubicin and the PI3K inhibitor? what is the overlap for these resistance mutations? If there are specific domains associated with resistance, how frequent are these domains mutated in larger sequencing datasets such as the TCGA? There is so much great structure-activity relationships that can be surmised from their data that it is a shame not more is analyzed. They are clearly capable of this analysis.

3. A technical clarification is needed. Their data suggests that mutations that propel growth of cells are also those associated with more resistance to doxorubicin and to the PI3K mutation (or is it? again, the manuscript is not clear in this issue). If so, then could this be an analytical artifact because if a mutation makes cells grow faster, then their frequency will be greater when the drug therapy is applied. This would result in the impression that there are more of these cells remaining when in fact that might not be the case.

Reviewer #2 (Remarks to the Author):

In the manuscript of Chen et al. somatic mutations of the PI3K pathway genes PIK3CA, PIK3R1, AKT1, AKT2, AKT3, PTEN and TP53 is determined in a Chinese cohort of 149 breast cancer patients using amplicon sequencing. For the most frequent mutated genes PIK3CA and PIK3R1 barcoded mutants were introduced in MCF10A and HMEC cell lines to screen for mutants that impacted either proliferation or drug response. Via this approach new impactful PIK3CA and PIK3R1 mutants were identified besides previously reported variants. Furthermore, impactful mutants were shown to stimulate proliferation and PI3K/AKT signaling.

Determination of the functional consequences of somatic mutations is an important step in the

vast expanding area of tumor sequencing and targeted therapies. The work of Chen et al. demonstrates that mutation-phenotype screening can easily identify functionally relevant mutations implicated in cancer development and drug response. The identification of new driver mutations in the PI3K pathway is interesting and may have potential clinical utility. However, some conclusions are not sufficiently supported by the data presented as detailed below.

Major Concerns:

It is doubtful whether the presented data provide solid support for the conclusion that “mutation pattern of Chinese breast cancer patients is distinct from other populations” (sentence 321-323) for the following reasons;

- Are 149 sequenced tumors in the Chinese cohort (versus thousands in TCGA and COSMIC) sufficient numbers to demonstrate a difference?
- Sequence coverage of Chinese cohort is much higher (1000 x in this study compared to TCGA average of 100x) maybe leading to higher mutation rates?
- The Chinese cohort seems to have an over-representation of Luminal-B breast cancers, which may contribute to higher overall mutation rates and higher mutation rates in specific genes (e.g. more mutations in PIK3R1 have been reported in LuminalB versus LuminalA tumors (Nature TCGA 2012), which is also shown in this study).

Authors should address these issues.

From Figure 7D it cannot be concluded that the two PIK3CA/H1047R and PIK3R1/K674R mutations together increase AKT signaling, since PIK3CA/H1047R together with PIK3R1/WT already shows maximal p473AKT levels (sentence 480-481). Similarly, from figure 7G,H it cannot be concluded that the PIK3CA/Q760L and PIK3R1/Q329L combination impairs proliferation and AKT signaling compared to PIK3R1/Q329L mutation and PIK3CA/WT (sentence 484-487) since no effect on p473AKT is visible, and the proliferation curves are overlapping. Actually, the PIK3CA/Q760L mutation seems to have pro-proliferation and AKT signaling effects on its own (the growth curves in 7G appear somewhat misleading as the WT lines proliferate much faster than in previous experiments 7C,E), thereby not supporting the qualification as a neutral mutation.

The genetic screen was setup to identify PIK3CA and PIK3R1 mutants that have an effect on proliferation, doxorubicin-response and BKM120-response. Screening data for all screens are shown, but the mutants were only validated in proliferation assays. To properly validate the functional screens performed, the mutants should also be re-tested for their effect on both doxorubicin and BKM120 treatments.

Minor points:

The authors do not include molecular subtyping in their discussions. For example, the PIK3CA/E39K and PIK3R1/Q329L mutations were identified in HER2 positive breast cancer patients. It would be of interest to test whether these impactful mutations confer resistance to HER2 targeting therapies in a HER2 positive cell line.

PIK3CA E453K mutant is not depicted in Figure2B

Figure 7C,E,G: it would be clearer for the readers if similar color coding (for mut combo, WT/MUT

etc) of the legends is used.

“Multiple mutations in the PI3K/AKT pathway in breast cancer have not been reported in other datasets” (sentence 321-322). According to Pereira et al Nature Communications 2016 PIK3CA or PIK3R1 and PTEN aberrations co-occur.

Reviewer #3 (Remarks to the Author):

This study uses a targeted sequencing approach to identify somatic mutations in PI3K pathway genes in tumours from a cohort of Chinese breast cancer patients. The authors then generate a retroviral library of the identified mutations and perform a high-throughput mutation-phenotype screen to identify the specific mutations that functionally contribute to cell proliferation and drug resistance.

This is an interesting study. While the use of individually bar-coded mutation libraries for high-throughput functional screens is not in itself new, the application of such an approach to screen for PIK3CA and PIK3R1 mutations that drive proliferation and drug response is, to my knowledge, novel. The results, although perhaps not altogether surprising, are interesting and should be of general interest to researchers in the field.

However, there are, in my mind, three main weaknesses in this study.

1. There is a general lack of information about the reproducibility of the data. Indeed, much of the data presented appears to be from single experiments (e.g. for the cell growth experiments the figure legends indicate that “Five wells were measured per condition” suggesting the results are the mean (and SD?) of 5 technical replicates from a single experiment). The data would be much more convincing if similar results were obtained in multiple independent experiments.
2. The functional experiments validating the “impactful” mutations derived from the screens (figs 5 and 6) only test mutations found to be enriched in the screens. It seems to me that in order to properly validate the screens, some of the non-enriched mutations should also be tested and shown not to drive proliferation and/or drug resistance.
3. In my mind, the most interesting question addressed in this study is the possibility that specific PIK3CA and/or PIK3R1 mutations may be responsible for resistance to treatment. Unfortunately, the data provided to support specific mutations driving drug resistance is not strong. While the qualitative western blot pAkt data is consistent with a decreased sensitivity to BKM-120 in cells expressing selected PIK3CA or PIK3R1 mutations, dose-response curves allowing quantitation of changes in IC50 would be more convincing. Also, although mutations enriched by exposure doxorubicin are identified, no data is shown at all for the effects of specific mutations on sensitivity to this drug.

Other minor specific comments

Line 289: “Most somatic mutations were validated using...”. This is ambiguous. Does it mean that validation was not attempted for all mutations identified or that some mutations did not validate? Were all 75 somatic mutations listed in table S2 validated?

Lines 332-333: “Based on the mutation profiles of the TCGA, COSMIC and FDUSCC cohorts, a set of PIK3CA and PIK3R1 mutants were generated”. What mutations were included in this “set” and how were they selected?

Lines 505-507: "the mutation spectrum in the Chinese cohort on which this study was based was distinct from the spectra reported in the TCGA and COSMIC datasets". Do the authors have any thoughts on why this might be the case?

Line 532: "Similar to the results of a previous study...". Which previous study? Reference?

Line 542: "PETN" should be "PTEN".

Line 548: "Consistent with the results of previous work...". Again, which previous work? Reference(s)?

Lines 597-598: "Thus, the sensitivity of breast cancer cells bearing PI3K mutations such as BKM120 and BYL719 to PI3Ki remains to be verified" presumably should be "Thus, the sensitivity of breast cancer cells bearing PI3K mutations to PI3Ki such as BKM120 and BYL719 remains to be verified".

Figures 5F-H: Given that potential differences in exposure time can influence the apparent intensity of bands on a western blot, the fact that many of the mutations are on different gels makes it difficult to accurately discern the extent to which some mutations (e.g. E39K, M1043V, and similar) activate pAkt. While the number of mutations investigated makes the use of multiple gels unavoidable, it is a pity that a control sample (e.g. WT) was not included on all gels to enable comparison across blots. In the absence of such a consistent control, care must be taken in interpreting the relative activities of the different mutations.

Figures 5C and 6C: It is not clear to me what value the images of the 3D structures add given the little information provided to help interpret them (For example, what do the different colours represent? Where are the various important sites within the structures. e.g. catalytic cleft, substrate binding site, p85 binding site, etc.). What are these images intended to illustrate?

Point-by-point response to the referees' comments

Reviewer #1 (Reviewer Comments to the Author):

The manuscript by Chen, et al examines PI3K pathway mutations identified in a Chinese breast cancer cohort. The frequency and the functional consequences of the range of mutations relative to proliferation and drug sensitivity are reported. The work has some interest particularly in the use of the recombination-based mutation barcoding (ReMB) libraries in the function screening. But I have specific concerns about the data, data analysis, and conclusions:

1. The authors wish to claim that Chinese patients have a different mutational profile than those published from Caucasian cohorts. However, frequency comparisons across different platforms with different coverage and different analytical algorithms are difficult and subject to technical biases. In looking at the mutational frequency of another cancer gene, TP53, though comparable with other reports in general, was remarkably low in the triple negative breast cancers in this cohort. The Fudan group registers probably around 15-20% mutation rate in TP53 in TNBC, whereas almost all studies report a much higher rate (~60-80%). This suggests either that their diagnostics are off, that their sequencing/algorithms are not comparable with other studies, or that Chinese have much fewer TP53 mutations than any other population. This of course raise the question of comparability of frequency determinations. I would recommend that the authors do not stress this point.

Response: We appreciate the reviewer's kind recommendations. We agreed that different platforms may bring technical biases when comparing mutational frequency between different cohorts. In recent years, next generation sequencing (NGS) technology has revolutionized genomic and genetic research. For example, targeted sequencing focusing on specific regions of a genome allows us to sequence target genes with a high level of coverage without generating significant quantities of off-target data, providing a highly efficient way to find biologically and pathologically relevant variants. Ion Torrent PGM and Illumina Hiseq/MiSeq are two major sequencing platforms currently in use. Quail et al sequenced the microbial genomes of *Bordetella pertussis* (67.7% GC), *Salmonella Pullorum* (52% GC), *Staphylococcus aureus* (33% GC) and *Plasmodium falciparum* (19.3% GC), and reported that the data using the Ion Torrent PGM platform had a higher raw error rate (~1.8%) compared to Illumina data (0.8%). However, if there is sufficient coverage, the representation and ability to identify SNPs was closely matched between PGM and Illumina, with more true positives being identified in PGM data, but with far fewer false positives in the Illumina data¹. Thus, we feel confident in pronouncing that the Ion Torrent PGM sequencing with coverage of 1000× conducted in our study is a reliable targeted sequencing platform, which provide scalable and lower-cost methods for DNA sequencing. To further verify *PIK3CA* and *PIK3R1* mutations, we also performed Pyrosequencing for breast cancer tissue DNA and Sanger sequencing for paired blood DNA (**Fig. R1**). Most somatic *PIK3CA* and *PIK3R1* mutations in our study are verified.

Figure R1

A *PIK3CA*

B *PIK3R1*

Figure R1. Mutations validation via Pyrosequencing for breast cancer tissue DNA and Sanger sequencing for paired blood DNA. (A) *PIK3CA*, (B) *PIK3R1*.

We think that the limited cohort size may be what mainly contribute to the bias. In this study, we retrospectively obtained a total of 149 pathologically confirmed primary breast cancer samples from the Department of Breast Surgery at Fudan University Shanghai Cancer Center (FDUSCC) to examine somatic mutation via targeted sequencing. These breast cancer samples were characterized as luminal-like, human epidermal growth factor receptor 2 (HER2)-enriched, and triple-negative subtypes according to their estrogen receptor (ER), progesterone receptor (PR), HER2 and Ki67 index. By this stratification, 25 breast cancer cases were characterized as triple-negative (ER-, PR-, and HER2-) subtype. We notice that the total mutation (44%, 11/25) and missense mutation (16%, 4/25) rates for *TP53* in TNBCs from FDUSCC were lower than those (73%, 108/147, for total; 36%, 53/147, for missense) in TNBCs from TCGA dataset, respectively. Considering that the number of TNBCs in FDUSCC is much smaller than that in TCGA dataset, we hope that the increment of cases will provide a more accurate mutation frequency for breast cancers in Chinese population in the future. We plan to sequence more mammary tumors to generate breast cancer genetic landscape in China which will be a good complement to TCGA and COSMIC datasets. In accordance to the reviewer's suggestion, we have removed the data from the manuscript.

2. The most interesting aspect of this study is the ReMB screen which is amongst the most complete for the PI3K pathway to date. But unfortunately, the authors chose not to spend much energy in the detailed computational and statistical analysis of this screen thus leaving many unanswered questions.

Response: We have carefully and comprehensively revised our manuscript in accordance to the reviewer's suggestion, and our responses to each individual questions are as follows:

(1) For example, what are the domains that are mutated in *PIK3CA* showing specific phenotypes, and which are not effective? How has this information improved the functional prediction for this pathway?

Response: *PIK3CA* is one of the most frequently mutated genes in breast cancer. The *PIK3CA*-encoded p110 α contains an N-terminal adaptor-binding domain (ABD), a Ras-binding domain (RBD), a C2 domain, a helical domain and a catalytic domain. The majority of *PIK3CA* mutations occur in the helical (exon 9) and kinase (exon 20) domains, for instance, E542K and E545K in exon 9 and H1047R in exon 20 are three hotspots (**Table R1**)². More evidences show that these hotspots result in the increased activity of *PIK3CA* kinase, activation of the PI3K/AKT pathway and *in vitro* and *in vivo*^{3,4}.

Table R1 Distribution of *PIK3CA* mutations in various functional domains

domains	TCGA ¹				FDUSCC ²					
	N	N/PCM	(%)	N/Total	(%)	N	N/PCM	(%)	N/Total	(%)
PI3K-ABD	5	5/318	2%	5/1105	0%	11	11/89	12%	11/149	7%
PI3K-RBD	0	0/318	0%	0/1105	0%	7	7/89	8%	7/149	5%
C2	35	35/318	11%	35/1105	3%	4	4/89	4%	4/149	3%
Helical	120	120/318	38%	120/1105	11%	13	13/89	15%	13/149	9%
PI4K	142	142/318	45%	142/1105	13%	39	39/89	44%	39/149	26%
Linker region	21	21/318	7%	21/1105	2%	15	15/89	17%	15/149	10%

Abbreviation: ABD, adaptor-binding domain; RBD, Ras-binding domain.

Note: Mutation denotes somatic missense mutation;

¹TCGA: *PIK3CA* mutation (PCM), 318 cases; Total, 1105 cases;

²FDUSCC: *PIK3CA* mutation (PCM), 89 cases; Total, 149 cases;

The location of the driver mutations in different domains of p110 α would suggest that they operate through different mechanisms. **(1) ABD domain:** Past researches supported that the ABD domain not only binds to the iSH2 domain of p85, but also interacts with the PI4K kinase domain in p110 α and a linker region between ABD and RBD⁵. Although fewer driver mutations were found in ABD, E39K might disrupt these interactions by causing a conformational change of the kinase domain. **(2) C2 domain:** The C2 domain has been postulated to link p110 α to the plasma membrane. Driver mutations (N345 and E453 substitutions) in the C2 domain increase the positive surface charge of the domain and are thought to facilitate its localization to the cell membrane, making lipid kinase activity independent of upstream signaling⁴. **(3) Helical domain:** The helical domain mediates the interaction with the nSH2 domain of p85, which is responsible for the p85-induced inhibition of p110 α . The mutations (E542K and E545K) in the helical domain could interfere with this p85-p110 α interaction and disrupt signal regulation^{3, 4, 6}. **(4) Kinase domain:** Interaction of Ras with p110 α is known to increase the kinase activity of p110 α , providing a conformational change for substrate-binding⁷. It has been proven that H1047* mutations in the kinase domain may induce a similar change to increase affinity with substrates⁸. Taken together, our study showed that the driver mutations are widely distributed over all domains of p110 α , with the exception of the RBD domain.

The *PIK3R1*-encoded p85 α regulatory subunit has an N-terminal SH3 domain, a domain homologous

to the Rho GTPase-activating protein domain (Rho-GAP), and three SH2 domains (nSH2, iSH2 and cSH2). As shown in **Table R2**, the somatic missense mutations in *PIK3R1* were mainly located in iSH2 and cSH2 domains in the TCGA dataset. Meanwhile, the frequency of Rho-GAP, iSH2, cSH2 and linker region mutations were found to be higher in the FDUSCC dataset.

Table R2 Distribution of *PIK3R1* mutations in various functional domains

domains	TCGA ¹					FDUSCC ²				
	N	N/PRM	(%)	N/Total	(%)	N	N/PRM	(%)	N/Total	(%)
SH3	0	0/7	0%	0/1105	0%	0	0/38	0%	0/149	0%
Rho-GAP	0	0/7	0%	0/1105	0%	5	5/38	13%	5/149	3%
nSH2	0	0/7	0%	0/1105	0%	0	0/38	0%	0/149	0%
iSH2	4	4/7	57%	4/1105	0%	15	15/38	39%	15/149	10%
cSH2	3	3/7	43%	3/1105	0%	7	7/38	18%	7/149	5%
Linker region	0	0/7	0%	0/1105	0%	11	11/38	29%	11/149	7%

Abbreviation: nSH2, N-terminal SH2 domain; iSH2, inter-SH2 domain; cSH2, C-terminal SH2 domain.

Note: Mutation denotes somatic missense mutation;

¹TCGA: *PIK3R1* mutation (PRM), 7 cases; Total, 1105 cases;

²FDUSCC: *PIK3R1* mutation (PRM), 38 cases; Total, 149 cases;

These *PIK3R1*-driver mutations were identified in various domains, suggesting they may also have different mechanisms. (1) **Rho-GAP domain:** Cheung et al. demonstrated that the *PIK3R1* E160* mutation disrupts the interaction between p85 α and PTEN, resulting in destabilization of PTEN, increased PI3K signaling and transformation⁹. We found that expression of p85 α E160D also promotes cell proliferation and AKT signaling, suggesting that novel E160D mutation might be a functional hotspot in the Rho-GAP domain of *PIK3R1*. (2) **SH2 domains:** The nSH2 and iSH2 domains of p85 α are required for interaction with the ABD, C2 and Helical domains of p110 α . Recent studies have indicated that other functional mutations (K379E, R503W, KS549delN, D560Y, KS549delN, N564D and QYL579delL) in the nSH2 and iSH2 domains of *PIK3R1* activate the canonical PI3K pathway by disrupting the inhibitory contact of the protein with the C2 domain of p110 α ¹⁰. The *PIK3R1* ReMB screen confirmed that the N564D mutation promotes and activates cell proliferation and AKT signaling in mammary cells. Recently, the *PIK3R1* R348* and L370fs forms of p85 α were shown to be truncated within the nSH2 domain and incapable of binding p110 α , suggesting that these two mutants activate the ERK and JNK pathways in a PI3K signaling-independent manner¹¹. (3) **Linker regions:** In this study, we also identified Q329L (at the nSH2 domain boundary), and K674R (at the cSH2 domain boundary) as mutations that affect cell proliferation and drug response. These findings suggest that more detailed classification of functional mutations in *PIK3R1* and additional research on the mechanisms involved will be needed.

(2) What are the mutations that render cells resistant to doxorubicin and the PI3K inhibitor? what is the overlap for these resistance mutations?

The *PIK3CA* ReMB screens show that the mutations resulting in resistance to the doxorubicin includes N345I/K and E453K in C2 domain; E542K in helical domain; and M1043V, H1047R/T/L and G1049R in kinase domain. Meanwhile, mutations that render cells resistant to the pan-PI3K inhibitor (BKM120) were as follows: E39K in ABD domain; N345I and E453K in C2 domain; E542K and E545K in helical domain; and H104R/L and G1049R in kinase domain (**Table R3**). Mutations N345I, E453K, E542K, H104R/L and G1049R were related to both doxorubicin and BKM120 resistance.

Table R3 The frequencies of high-ranking *PIK3CA* mutations in different datasets

Protein	Domain	Pro	Dox	BKM	FDUSCC	COSMIC	TCGA ¹	Comments ²
p.E39K	PI3K-ABD	x		x	1/149 (0.7%)	0/14923 (0.0%)	0/1105 (0.0%)	novel, rare
p.N345K	C2 PI3K-type		x	x	4/149 (2.7%)	80/14923 (0.5%)	16/1105 (1.4%)	hotspot
p.N345I	C2 PI3K-type	x	x		0/149 (0.0%)	2/14923 (0.0%)	0/1105 (0.0%)	rare
p.E453K	C2 PI3K-type	x	x	x	0/149 (0.0%)	16/14923 (0.1%)	7/1105 (0.6%)	rare
p.E542K	PIK helical	x	x	x	2/149 (1.3%)	453/14923 (3.0%)	41/1105 (3.7%)	hotspot
p.E545K	PIK helical	x		x	8/149 (5.4%)	779/14923 (5.2%)	63/1105 (5.7%)	hotspot
p.M1043V	PI3K/PI4K		x		0/149 (0.0%)	6/14923 (0.0%)	1/1105 (0.1%)	rare
p.H1047R	PI3K/PI4K	x	x	x	30/149 (20.1%)	2146/14923 (14.4%)	128/1105 (11.6%)	hotspot
p.H1047T	PI3K/PI4K	x	x	x	0/149 (0.0%)	1/14923 (0.0%)	0/1105 (0.0%)	rare
p.H1047L	PI3K/PI4K		x		3/149 (2.0%)	220/14923 (1.5%)	12/1105 (1.1%)	hotspot
p.G1049R	PI3K/PI4K	x	x	x	1/149 (0.7%)	31/14923 (0.2%)	2/1105 (0.2%)	rare

¹ Breast Invasive Carcinoma (TCGA, Provisional)² hotspot, mutation frequency $\geq 2\%$ in either datasets; rare, mutation frequency $< 2\%$; novel, new mutation identified in the FDUSCC cohort
Bold values denote mutation frequency $\geq 2\%$

The *PIK3R1* mutations rendering cells resistance to doxorubicin includes E160D in Rho-GAP domain; D560Y, N564D and R574T in iSH2 domain; and Q392L and K674R in linker domain (**Table R4**). These mutations also exhibited resistance to pan-PI3K inhibitor (BKM120), with the exception of K674R. And the frequency of drug-resistant mutations in *PIK3R1* was much lower than that in *PIK3CA*. Interestingly, these drug-resistant mutations mostly overlap with driver mutations identified in the proliferation assay.

Table R4 The frequencies of high-ranking *PIK3R1* mutations in different datasets

Protein	Domain	Pro	Dox	BKM	FDUSCC	COSMIC	TCGA ¹	Comments ²
p.E160D	Rho-GAP	x	x	x	4/149 (2.7%)	0/3554 (0.0%)	0/1105 (0.0%)	novel, hotspot
p.Q329L	NONE	x	x	x	1/149 (0.7%)	0/3554 (0.0%)	0/1105 (0.0%)	novel, rare
p.D560Y	iSH2		x	x	0/149 (0.0%)	1/3554 (0.0%)	0/1105 (0.0%)	rare
p.N564D	iSH2	x	x	x	1/149 (0.7%)	0/3554 (0.0%)	0/1105 (0.0%)	novel, rare
p.R574T	iSH2		x	x	0/149 (0.0%)	1/3554 (0.0%)	0/1105 (0.0%)	rare
p.K674R	NONE	x	x		2/149 (1.3%)	0/3554 (0.0%)	0/1105 (0.0%)	novel, rare

¹ Breast Invasive Carcinoma (TCGA, Provisional)² hotspot, mutation frequency $\geq 2\%$ in either datasets; rare, mutation frequency $< 2\%$; novel, new mutation identified in the FDUSCC cohort
Bold values denote mutation frequency $\geq 2\%$

(3) If there are specific domains associated with resistance, how frequent are these domains mutated in larger sequencing datasets such as the TCGA? There is so much great structure-activity relationships that can be surmised from their data that it is a shame not more is analyzed. They are clearly capable of this analysis.

Response: According to the reviewer's insightful suggestion, we further analyzed the frequency of mutations in specific domains in the TCGA and FDUSCC datasets. Analysis of *PIK3CA* ReMB screen confirmed the growth-promoting activity and drug resistance properties of E542K (in helical), E545K (in helical) and H1047R/L (in PI4K). These results were consistent with previous reports². Notably, other novel and rare mutations such as E39K (in ABD), N345K/I (in C2), E453K (in C2), M1043V and G1049R (in PI4K) were also enriched in these assays, and these were identified as novel PI3K-driver mutations (**Table R5**). **(1) ABD domain:** Impactful mutations within ABD domain occurred in 1 out of 149 patients in the FDUSCC dataset; meanwhile, no impactful mutations were found in TCGA dataset. The E39K in ABD domain could promote cell proliferation and BKM120 resistance in MCF-10A cell line. **(2) RBD domain:** In this analysis, no impactful mutations were found within the RBD domain. **(3) C2 domain:** C2 domain has been proposed to bind to cellular membranes and mutations within C2 domain, including N345K/I and E453K, were detected at a rate of 2% for TCGA and a rate of 3% for FDUSCC. **(4) Helical**

domain: The helical domain harbored two hotspot mutations, E542K and E545K, which promote proliferation and drug resistance. Both cohorts harbored the two mutations at a high frequency (9% for TCGA and 7% for FDUSCC). **(5) Kinase domain:** The driver mutations at PI4K domain occurred at a high frequency in breast cancer (13% for TCGA and 13% for FDUSCC), including M1043V, H1047R/T/L and G1049R.

Table R5 Distribution of impactful *PIK3CA* mutations in various functional domains

domains	Impactful PIK3CA mutation	TCGA ¹				FDUSCC ²					
		N	N/IPC	(%)	N/Total	(%)	N	N/IPC	(%)	N/Total	(%)
PI3K-ABD	E39K	0	0/269	0%	0/1105	0%	1	1/49	1%	1/149	1%
PI3K-RBD		0	0/269	0%	0/1105	0%	0	0/49	0%	0/149	0%
C2	N345K/I, E453K	23	23/269	9%	35/1105	2%	4	4/49	4%	4/149	3%
Helical	E542K, E545K	104	104/269	39%	120/1105	9%	10	10/49	11%	10/149	7%
PI4K	M1043V, H1047R/T/L, G1049R	142	142/269	53%	143/1105	13%	34	34/49	38%	34/149	23%
Linker region		0	0/269	0%	0/1105	0%	0	0/49	0%	0/149	0%

Abbreviation: ABD, adaptor-binding domain; RBD, Ras-binding domain.

Note: Mutation denotes somatic missense mutation;

¹TCGA: Impactful *PIK3CA* mutation (IPC), 269 cases; Total, 1105 cases;

²FDUSCC: Impactful *PIK3CA* mutation (IPC), 49 cases; Total, 149 cases;

To assess the clinical significance of these mutations in breast cancer, we analyzed the relationship between mutation status and clinicopathologic characteristics in the TCGA dataset. Among 1105 breast cancer cases, the clinical information of 977 cases were obtained. The clinicopathological characteristics of patients with mutations are as summarized in **Table R6**. 317 of 977 patients (32.4%) harbored *PIK3CA* missense mutation. Patients diagnosed after the age of 50 years did show a higher rate of *PIK3CA* mutation than younger patients ($p = 0.032$). Breast tumors with mutated *PIK3CA* were more likely to be ER positive ($p < 0.001$) and PR positive ($p < 0.001$) compared to cases without *PIK3CA* mutation. Furthermore, we classified all patients into three categories (no mutation, non-impactful mutation and impactful mutation) by their *PIK3CA* alteration. Among the 977 breast cancer patients, 55 patients (5.6%) had *PIK3CA* non-impactful mutation, and 262 patients (26.8%) had *PIK3CA* impactful mutations. *PIK3CA* impactful mutations were closely associated with the expression of hormone receptors (ER, $p < 0.001$; PR, $p < 0.001$).

Table R6 Clinicopathological variables and the mutation status of *PIK3CA* in TCGA database

Variables	Number of patients(%)	PIK3CA mutation		P value	PIK3CA impactful mutation			P1	P2
		Neg (%)	Pos (%)		No mut	Neg (%)	Pos (%)		
Total	977	660(67.6)	317(32.4)		660(67.6)	55(5.6)	262(26.8)		
Age (median 58, range 26-90)				0.032				0.186	0.060
≤50 years	294(30.1)	213(21.8)	81(8.3)		213(21.8)	13(1.3)	68(7.0)		
>50years	683(69.9)	447(45.8)	236(24.1)		447(45.8)	42(4.3)	194(19.8)		
Menopausal status				0.468				0.186	0.775
Premenopause	308(31.5)	213(21.8)	95(9.7)		213(21.8)	13(1.3)	82(8.4)		
Postmenopause	669(68.5)	447(45.8)	222(22.7)		447(45.8)	42(4.3)	180(18.4)		
Tumor size				0.622				0.246	0.191
≤2cm	258(26.4)	168(17.2)	90(9.2)		168(17.2)	13(1.3)	77(7.9)		
>2, 5≤cm	564(57.7)	386(39.5)	178(18.2)		386(39.5)	36(3.7)	142(14.5)		
>5cm	155(15.9)	106(10.8)	49(5.0)		106(10.8)	6(0.6)	43(4.4)		
LN status				0.563				0.897	0.479
Negative	477(48.8)	318(32.5)	159(16.3)		318(32.5)	26(2.7)	133(13.6)		
Positive	500(51.2)	342(35.0)	158(16.2)		342(35.0)	29(3.0)	129(13.2)		
ER status				<0.001				0.193	<0.001
Negative	258(26.4)	212(21.7)	46(4.7)		212(21.7)	13(1.3)	33(3.4)		
Positive	719(73.6)	448(45.9)	271(27.7)		448(45.9)	42(4.3)	229(23.4)		
PR status				<0.001				0.198	<0.001
Negative	357(36.5)	287(29.4)	70(7.1)		287(29.4)	19(1.9)	51(5.2)		
Positive	620(63.5)	373(38.2)	247(25.3)		373(38.2)	36(3.7)	211(21.6)		
HER-2/neu status				0.37				0.869	0.278
Negative	798(81.7)	534(54.7)	264(27.0)		534(54.7)	44(4.5)	220(22.5)		
Positive	179(18.3)	126(12.9)	53(5.4)		126(12.9)	11(1.1)	42(4.3)		

Abbreviations: ER, estrogen receptor; HER-2, human epidermal growth factor receptor 2; PR, progesterone receptor; SD, standard deviation; Neg, negative; Pos, positive; LN, lymph node.

Note: Based on Pearson χ^2 test, for which P is based on Fisher's exact test; Bold values denote P value < 0.05. P1, no mutation versus no impactful mutation; P2, no mutation versus impactful mutation.

Since mutations in different domains may be associated with different functions, we categorized patients carrying *PIK3CA* mutation into the subgroups according to the location of their mutation. Since

impactful mutations reside mainly in the helical, PI4K kinase and C2 domains, we further analyzed the relationship between the mutated domains and clinicopathologic characteristics (**Table R7**). When compared against patient who didn't harbor mutations, patients with C2 domain mutation were more likely to be premenopausal ($p = 0.034$), and more likely to be PR-positive ($p = 0.010$). Similarly, patients with *PIK3CA* mutation in Helical domain ($p = 0.002$) were more likely to be diagnosed after the age of 50 years. Finally, breast cancer with *PIK3CA* mutation in Helical domain (120 cases, 12.3%) or PI4K domain (154 cases, 15.8%) were more likely to be ER- and PR- positive ($p < 0.001$, respectively).

Table R7 Clinicopathological variables and the mutation status of domains in PIK3CA in TCGA database

Variables	Number of patients(%)	PIK3CA domain mutation							Linker region (%)	P1	P2	P3
		Neg (%)	ABD (%)	RBD (%)	C2 (%)	Helica (%)	PI4K (%)					
Total	977	660(67.6)	3(0.3)	0(0)	27(2.8)	120(12.3)	154(15.8)	13(1.3)				
Age (median 58, range 26-90)									0.085	0.002	0.128	
≤50 years	294(30.1)	213(21.8)	1(0.1)	0(0)	13(1.3)	22(2.3)	40(4.1)	5(0.5)				
>50years	683(69.9)	447(45.8)	2(0.2)	0(0)	14(1.4)	98(10.0)	114(11.7)	8(0.8)				
Menopausal status									0.034	0.224	0.374	
Premenopause	308(31.5)	213(21.8)	1(0.1)	0(0)	14(1.4)	32(3.3)	44(4.5)	4(0.4)				
Postmenopause	669(68.5)	447(45.8)	2(0.2)	0(0)	13(1.3)	88(9.0)	110(11.3)	9(0.9)				
Tumor size									0.106	0.529	0.236	
≤2cm	258(26.4)	168(17.2)	0(0)	0(0)	10(1.0)	33(3.4)	44(4.5)	3(0.3)				
>2, 5≤cm	564(57.7)	386(39.5)	3(0.3)	0(0)	15(1.5)	70(7.2)	82(8.4)	8(0.8)				
>5cm	155(15.9)	106(10.8)	0(0)	0(0)	2(0.2)	17(1.7)	28(2.9)	2(0.2)				
LN status									0.708	0.593	0.486	
Negative	477(48.8)	318(32.5)	0(0)	0(0)	14(1.4)	61(6.2)	79(8.1)	5(0.5)				
Positive	500(51.2)	342(35.0)	3(0.3)	0(0)	13(1.3)	59(6.0)	75(7.7)	8(0.8)				
ER status									0.058	<0.001	<0.001	
Negative	258(26.4)	212(21.7)	3(0.3)	0(0)	4(0.4)	14(1.4)	23(2.4)	2(0.2)				
Positive	719(73.6)	448(45.9)	0(0)	0(0)	23(2.4)	106(10.8)	131(13.4)	11(1.1)				
PR status									0.010	<0.001	<0.001	
Negative	357(36.5)	287(29.4)	3(0.3)	0(0)	5(0.5)	25(2.6)	32(3.3)	5(0.5)				
Positive	620(63.5)	373(38.2)	0(0)	0(0)	22(2.3)	95(9.7)	122(12.5)	8(0.8)				
HER-2/neu status									0.578	0.846	0.230	
Negative	798(81.7)	534(54.7)	3(0.3)	0(0)	23(2.4)	98(10.0)	131(13.4)	9(0.9)				
Positive	179(18.3)	126(12.9)	0(0)	0(0)	4(0.4)	22(2.3)	23(2.4)	4(0.4)				

Abbreviations: ER, estrogen receptor; HER-2, human epidermal growth factor receptor 2; PR, progesterone receptor; SD, standard deviation; Neg, negative; Pos, positive; LN, lymph node.

Note: Based on Pearson χ^2 test, for which P is based on Fisher's exact test; Bold values denote P value < 0.05. P1, no mutation versus impactful mutation in C2 domain; P2, no mutation versus impactful mutation in Helica domain; P3, no mutation versus impactful mutation in PI4K kinase domain

To assess the clinical significance of *PIK3CA* mutation in breast cancer, we also investigated the relationship between mutation status and survival in TCGA database (**Fig. R2**). There was no significant increase in risk of disease recurrence or death associated with *PIK3CA* mutation ($p > 0.05$). There were no significant differences in disease-free survival (DFS) and overall survival (OS) between the *PIK3CA* mutation carriers and non-mutation carriers (**Fig. R2A,B**), impactful mutations carriers and non-carriers (**Fig. R2C,D**), nor mutated domains carriers and non-carriers (**Fig. R2E,F**). Although the PI3K pathway is an attractive therapeutic target with a high frequency of activating mutations, *PIK3CA* mutations might not serve as an independent predictor of risk in breast cancer. The Bartlett's group reported the similar findings based on the mutational analysis of PI3K/AKT pathway in the TEAM clinical trial². Nonetheless, it remains possible and even likely that mechanisms of *de novo* drug resistance in early cancer differ from those in advanced cancer which achieve drug resistance by tumor evolution. The underlying mechanism remains to be elucidated in the future.

Figure R2

Figure R2. Kaplan-Meier estimates of disease-free survival (DFS) and overall survival (OS) by *PIK3CA* alteration. (A and B) DFS and OS by *PIK3CA* mutation status. (C and D) DFS and OS by *PIK3CA* impactful mutation status. (E and F) DFS and OS by mutated domains in *PIK3CA*.

The analysis of *PIK3R1* ReMB screen demonstrated that novel *PI3KR1* driver mutations E160D (in Rho-GAP), N564D (in iSH2), Q329L (in linker) and K674R (in linker) were enriched in proliferation and drug response assays (**Table R8**). We observed that most impactful *PIK3R1* mutations (8/149) only occurred in the patients from FDUSCC dataset but not from TCGA dataset. The *PIK3R1* impactful mutation rate was 3% (4/149) in Rho-GAP domain, 1% (1/149) in iSH2 domain and 2% (3/149) in linker regions. Due to the limited cohort size of the FDUSCC dataset, we did not analyze the correlation between mutations and clinicopathologic characteristics in our current study. In the future, we plan to perform a comprehensive analysis of somatic mutations in the PI3K/AKT/mTOR (PAM) pathway in the BOLERO-5 clinical trial. We hope that the future studies will provide more applicable information to uncover the clinical significance of *PIK3R1* mutations.

Table R8 Distribution of impactful *PIK3R1* mutations in various functional domains

domains	Impactful PIK3R1 mutation	TCGA ¹					FDUSCC ²				
		N	N/IPRM	(%)	N/Total	(%)	N	N/IPRM	(%)	N/Total	(%)
SH3		0	0/0	0%	0/1105	0%	0	0/8	0%	0/149	0%
Rho-GAP	E160D	0	0/0	0%	0/1105	0%	4	4/8	50%	4/149	3%
nSH2		0	0/0	0%	0/1105	0%	0	0/8	0%	0/149	0%
iSH2	N564D	0	0/0	0%	0/1105	0%	1	1/8	13%	1/149	1%
cSH2		0	0/0	0%	0/1105	0%	0	0/8	0%	0/149	0%
Linker region	Q329L, K674R	0	0/0	0%	0/1105	0%	3	3/8	38%	3/149	2%

Abbreviation: nSH2, N-terminal SH2 domain; iSH2, inter-SH2 domain; cSH2, C-terminal SH2 domain.

Note: Mutation denotes somatic missense mutation;

¹TCGA: Impactful *PIK3R1* mutation (IPRM), 0 cases; Total, 1105 cases;

²FDUSCC: Impactful *PIK3R1* mutation (IPRM), 8 cases; Total, 149 cases;

3. A technical clarification is needed. Their data suggests that mutations that propel growth of cells are also those associated with more resistance to doxorubicin and to the PI3K mutation (or is it? again, the manuscript is not clear in this issue). If so, then could this be an analytical artifact because if a mutation makes cells grow faster, then their frequency will be greater when the drug therapy is applied. This would result in the impression that there are more of these cells remaining when in fact that might not be the case.

Response: We thank the reviewer's reminder. We apologize for our insufficient descriptions regarding this point in our previous manuscript. As mentioned above, the driver mutations (such as E39K, N345I, E453K, E542K, E545K, H104R/L and G1049R in *PIK3CA* and E160D, Q392L, D560Y, N564D and K674R in *PIK3R1*) that propel growth of cells were also found to be related to resistance to doxorubicin and/or PI3K inhibitor BKM120, we speculate that these activating PI3K mutations may have some general effect upon both proliferation and chemotherapy responses.

Doxorubicin, an anthracycline antibiotic, remains the most effective agent for breast cancer patients in clinical practice^{12,13}. However, its clinical use is restricted by the emergence of multidrug resistance and its low specificity for cancer cell^{14,15}. Doxorubicin interacts with DNA by intercalation and stabilizes the topoisomerase II complex to prevent the DNA double helix from being resealed, thereby stopping the process of replication¹⁶. Doxorubicin exhibits its cytotoxic effects at a PI3K-pathway independent manner. Thus, the cells harboring *PIK3CA* clonal-advantage mutations are likely to be enriched both in the presence or absence of doxorubicin.

BKM120 is a pyrimidine-derived pan-PI3K inhibitor with specific activity against class I PI3Ks¹⁷. This small molecule can bind in the ATP-binding site of the lipid kinase domain in the PI3K p110 isoforms¹⁸. BKM120 decreases the cellular levels of p-Akt in mechanistic models and relevant tumor cell lines, and shows a strong anti-proliferative effect. Inhibition of PI3K is a potentially attractive strategy for breast cancer treatment, and BKM120 has already entered clinical trials^{19,20}. Previous studies suggested that breast cancers with *PIK3CA* mutations are also sensitive to BKM120 treatment¹⁸. This compound shows comparable potency against p110 α activating mutations (IC₅₀ = 58 nM, for H1047R; IC₅₀ = 99 nM, for E545K) compared to wild-type protein (IC₅₀ = 52 nM, for wild type)¹⁸. A panel screening 353 cell lines demonstrated that BKM120 was capable of inhibiting both tumor cells presenting oncogenic mutated *PIK3CA* and non-mutated wild-type form¹⁸. Although *PIK3CA* mutant tumors exhibit a general sensitivity to BKM120, the pattern and mechanism of PI3K driver mutations that are sensitive to BKM120 remains to be verified. In clinical trials, tumor stabilization and partial tumor responses have been observed in *PIK3CA* mutant breast cancers treated with BKM120; however, dramatic regressions of *PIK3CA* mutant tumors are not uncommon²¹. Thus, it is hard to conclude that BKM120 is a specific inhibitor for *PIK3CA* driver mutation. A definitive conclusion awaits more detailed investigation of functionally well-characterized mutations in preclinical investigations and clinical trials with larger patient cohorts.

In our study, the driver *PIK3CA* mutations are more likely to render cells resistant to doxorubicin and BKM120, consistent with their clonal advantages in proliferation assay. Western-blot analysis shows that these mutations may activate PI3K/AKT signaling in the presence of inhibitor at a low concentration. These findings suggested that the driver mutations in PI3K may maintain their clonal advantage under the stress of common cytotoxin or non-specific inhibitor treatments. Thus, exploring driver-mutations specific inhibitors would provide insights on novel therapeutics for tumors harboring PI3K mutants.

Reviewer #2 (Reviewer Comments to the Author):

In the manuscript of Chen et al. somatic mutations of the PI3K pathway genes *PIK3CA*, *PIK3RI*, *AKT1*, *AKT2*, *AKT3*, *PTEN* and *TP53* is determined in a Chinese cohort of 149 breast cancer patients using amplicon sequencing. For the most frequent mutated genes *PIK3CA* and *PIK3RI* barcoded mutants were introduced in MCF-10A and HMEC cell lines to screen for mutants that impacted either proliferation or drug response. Via this approach new impactful *PIK3CA* and *PIK3RI* mutants were identified besides previously reported variants. Furthermore, impactful mutants were shown to stimulate proliferation and PI3K/AKT signaling.

Determination of the functional consequences of somatic mutations is an important step in the vast expanding area of tumor sequencing and targeted therapies. The work of Chen et al. demonstrates that mutation-phenotype screening can easily identify functionally relevant mutations implicated in cancer development and drug response. The identification of new driver mutations in the PI3K pathway is interesting and may have potential clinical utility. However, some conclusions are not sufficiently supported by the data presented as detailed below.

Major Concerns:

1. It is doubtful whether the presented data provide solid support for the conclusion that “mutation pattern of Chinese breast cancer patients is distinct from other populations” (sentence 321-323) for the following reasons: Are 149 sequenced tumors in the Chinese cohort (versus thousands in TCGA and COSMIC) sufficient numbers to demonstrate a difference?

Response: We concede to the reviewer’s observation. We agreed that the limited cohort size in the current study which may result in bias. In this study, we retrospectively obtained a total of 149 pathologically confirmed primary breast cancer samples from Fudan University Shanghai Cancer Center (FDUSCC) to examine somatic mutation via targeted sequencing. A total of 563 samples that were available for frozen fresh tissues were collected from 3,252 patients who were diagnosed as breast cancer at the Department of Breast Surgery in FDUSCC between June 2007 and December 2009. These breast cancer cases were characterized as luminal-like, human epidermal growth factor receptor 2 (HER2)-enrichment, and triple-negative subtypes according to the expression statuses of estrogen receptor (ER), progesterone receptor (PR), and HER2. In this study, ER, PR, HER2 expression statuses and Ki67 index were examined by IHC staining at the Department of Pathology in FDUSCC. Subsequently, we used the complete random sampling method to collect 107 (71.8%) luminal-like subtype cases, 17 (11.4%) HER2-enriched subtype cases and 25 (16.8%) triple-negative subtype cases to perform targeting sequencing. Ethical approval was obtained from the Institutional Review Board of FDUSCC, and each participant gave written informed consent to participation in this research.

A limited number of cases from one center is indeed insufficient to represent the overall characteristics of mutation pattern in a Chinese population, and perhaps rather unsuitable to be compared against the mutation spectra reported in the TCGA and COSMIC datasets. In the revised manuscript, we have changed the sentence to: “Our study found a series of novel mutational changes among PI3K pathway in Chinese breast cancer patients, showing that there were some differences in the distribution of somatic mutations between different race/ethnicity. Recently, Pereira et al found that *PIK3CA* or *PIK3RI* aberrations

frequently co-occur with PTEN mutations in breast cancer²². Consistent with this phenomenon, we also demonstrated that a high proportion of patients from FDUSCC harbored multiple somatic mutations in the PI3K/AKT pathway." We hope that this revision will convey a more accurate message to the readers. In the future, we plan to perform an analysis of somatic mutations in the PI3K/AKT/mTOR (PAM) pathway in the multi-centered BOLERO-5 clinical trial. We hope that our future works will provide a comprehensive mutation profile of the PI3K pathway in a Chinese population.

2. Sequence coverage of Chinese cohort is much higher (1000 x in this study compared to TCGA average of 100x) maybe leading to higher mutation rates?

Response: We also agreed that sequencing with higher coverage may contribute to the higher mutation rates in the current study. The TCGA project analyzed the majority of samples by whole exome sequencing (WES) with the standard coverage (around 100×) to detect variants among all of the expressed genes genome-wide. Meanwhile, targeted sequencing, which focuses on specific regions of a genome to sequence the target genes with a high level of coverage, provides a highly effective way to find biologically and pathologically relevant variants in genes. While both methods generate reliable sequence data and can be applied with different purposes for genetic variants discovery, in this study we chose the latter. We performed targeted sequencing with coverage as high as 1000× using the Ion Torrent PGM platform, which provided scalable and lower-cost methods for mutation identification. Ion Torrent and Illumina systems are two major sequencing platforms currently in use. Quail et al sequenced the microbial genomes of *Bordetella pertussis* (67.7% GC), *Salmonella Pullorum* (52% GC), *Staphylococcus aureus* (33% GC) and *Plasmodium falciparum* (19.3% GC), and reported that the data using the Ion Torrent PGM platform had a higher raw error rate (~1.8%) compared to Illumina data (0.80%). However, if there is sufficient coverage, the representation and ability to identify SNPs was closely matched between PGM and Illumina, with more true positives being identified in PGM data, but with far fewer false positives in the Illumina data¹. Thus, we feel confident in pronouncing that the Ion Torrent PGM sequencing with coverage of 1000× conducted in our study is a reliable targeted sequencing platform, which provide scalable and lower-cost methods for DNA sequencing. To further verify *PIK3CA* and *PIK3R1* mutations, we also performed Pyrosequencing for breast cancer tissue DNA and Sanger sequencing for paired blood DNA (**Fig. R3**). Most somatic *PIK3CA* and *PIK3R1* mutations in our study are verified.

Figure R3

Figure R3. Mutations validation via Pyrosequencing for breast cancer tissue DNA and Sanger sequencing for paired blood DNA. (A) *PIK3CA*, (B) *PIK3R1*.

3. The Chinese cohort seems to have an over-representation of Luminal-B breast cancers, which may contribute to higher overall mutation rates and higher mutation rates in specific genes (e.g. more mutations in *PIK3R1* have been reported in Luminal B versus Luminal A tumors (Nature TCGA 2012), which is also shown in this study). Authors should address these issues.

Response: We appreciate the reviewer's insight regarding the proportion of Luminal-B subtype breast cancers in our study. The breast cancer samples were categorized into as luminal-like, human epidermal growth factor receptor 2 (HER2)-enriched, and triple-negative subtypes according to their estrogen receptor (ER), progesterone receptor (PR), HER2 statuses and Ki67 index. The luminal subtype can be roughly defined as hormone receptor (ER and/or PR) -positivity in breast cancer. However, we know from published studies that there are some differences in the distribution of hormone receptor status between different race/ethnicity. Chlebowski et al reported that African American women had lower incidences of ER-positive (57%) and PR-positive (41%) tumors than white women (ER, 77% ; PR, 63%, respectively) in the SEER dataset²³. Our previous study reported that the proportion of hormone-dependent breast cancer was relatively lower in Chinese females (50–60%) compared to females from western countries^{24, 25}. Interestingly, Zheng et al reported that the incidence of ER-positive breast cancer in China has been increasing in recent years²⁶. These changes may be due to the new definition of immunohistochemical ER-positive (lowered from 10% to 1%) for breast cancer according to the St Gallen International Expert Consensus 2011^{26, 27}. In the reported TCGA project, 75% (616/825) of the patients were hormone

receptor-positive. Our study retrospectively obtained 149 pathologically proven primary breast cancer samples which were pathologically confirmed and treated at Fudan University Shanghai Cancer Center (FDUSCC). The rate of hormone receptor-positivity in our cohort was 72% (107/149), which is comparable to the rate observed in the TCGA dataset.

In clinical practice, luminal breast cancers can be separated into two subgroups: luminal A and luminal B. Luminal A tumors are characterized as having low expression of proliferation-related genes, whereas luminal B cancers presents as HER2 positive or have higher expression of proliferation marker (Ki67 > 14%). Of the two subtypes, Luminal B breast cancer is has a more aggressive clinical behavior and higher relapse rate, whereas luminal A breast cancer is has a more favorable clinical outcome²⁸. Additionally, PI3K activation is implicated in the acquired endocrine resistance of luminal breast cancers²⁹. In the TCGA dataset, there was a high mutation frequency of *PIK3CA* in luminal breast cancers (luminal A, 45% and Luminal B, 29%)³⁰. However, the protein data demonstrated that pAKT, pS6 and p4EBP1, typical markers of PI3K pathway activation, were not frequently elevated in *PIK3CA*-mutated luminal A breast cancer. However, these markers were highly expressed in non-luminal A subtypes, hinting that other genetic alterations may also activate the PI3K pathway. Based on the available frozen fresh tissues at FDUSCC, we randomly enrolled 32 Luminal A (50.3%), 75 Luminal B (50.3%), 17 HER2 (11.4%) and 25 TNBC (16.8%) cases for targeted sequencing of genes relevant to the PI3K pathway, including *PIK3CA*, *PIK3R1*, *AKT1*, *AKT2*, *AKT3*, *PTEN*, *PDK1*. The mutation profile of Chinese breast cancer patients would be a good complement to the existing TCGA and Cosmic datasets. We agree that increased proportion of luminal B among total luminal cancers may affect the mutation rates of specific genes, and inadvertently introduce some selection bias. The luminal B specimens were shown to carry some functional genetic alteration among genes, with the exception of *PIK3CA*. And our following research demonstrated that the driver mutations in *PIK3R1* could also lead to oncogenesis and hyperactivity of the PI3K pathway, suggesting the importance of synergetic regulation in PI3K activation.

4. From Figure 7D it cannot be concluded that the two *PIK3CA*/H1047R and *PIK3R1*/K674R mutations together increase AKT signaling, since *PIK3CA*/H1047R together with *PIK3R1*/WT already shows maximal p473AKT levels (sentence 480-481). Similarly, from figure 7G, H it cannot be concluded that the *PIK3CA*/Q760L and *PIK3R1*/Q329L combination impairs proliferation and AKT signaling compared to *PIK3R1*/Q329L mutation and *PIK3CA*/WT (sentence 484-487) since no effect on p473AKT is visible, and the proliferation curves are overlapping. Actually, the *PIK3CA*/Q760L mutation seems to have pro-proliferation and AKT signaling effects on its own (the growth curves in 7G appear somewhat misleading as the WT lines proliferate much faster than in previous experiments 7C, E), thereby not supporting the qualification as a neutral mutation.

Response: We are obliged to the reviewer's intuitive comments. To confirm whether the combination of *PIK3CA* H1047R and *PIK3R1* K674R mutations cooperatively increased the activation of the AKT signaling pathway, we repeated the Western blot with a shorter exposure time. As shown in **Fig. R4B (original Fig. 7d)**, the H1047R mutation up-regulates the phosphorylation of AKT at Ser473, and additional expression of exogenous *PIK3R1* K674R propels the pS473AKT signaling.

We apologize for our mistaken conclusion that the *PIK3CA* Q760L and *PIK3R1* Q329L combination

impairs proliferation and AKT signaling compared to *PIK3R1* Q329L mutation and *PIK3CA* WT (**Fig. R4C,D, original Fig. 7**). We have revised the statement to “In another case, proliferation-driver mutation *PIK3R1* Q329L at the nSH2 domain boundary could up-regulate cell proliferation and the phosphorylation of AKT (Ser473); while the *PIK3CA* Q760L mutation located in linker region between helical and kinase domains exhibited pro-proliferation and AKT signaling effects. However, a combination of *PIK3R1* Q329L and *PIK3CA* Q760L did not increase proliferation and AKT signaling compared to *PIK3R1* Q329L mutation and *PIK3CA* WT, indicating that the mechanisms for combined mutations in clinical cases will be complicated and challenging.”

Figure R4. (A and C) Growth properties of MCF-10A cells expressing combined *PIK3CA* and *PIK3R1* mutations. (B and D) the levels of phospho-AKT at Ser-473 in the indicated cells analyzed by Western blotting. The error bars indicate \pm s.d. derived from three independent experiments.

5. The genetic screen was setup to identify *PIK3CA* and *PIK3R1* mutants that have an effect on proliferation, doxorubicin-response and BKM120-response. Screening data for all screens are shown, but the mutants were only validated in proliferation assays. To properly validate the functional screens performed, the mutants should also be re-tested for their effect on both doxorubicin and BKM120 treatments.

Response: We have accepted the reviewer’s suggestion, and validated the functional mutations in *PIK3CA* and *PIK3R1* for their response to doxorubicin and BKM120 treatments. MCF-10A cells stably expressing *PIK3CA* wild type and mutations (impactful or non-impactful mutations) were plated in 96-well plates (500 cells/well) with indicated concentration of doxorubicin (1nM) or BKM120 (1 μ M). The confluency of the cultures was measured using an Incucyte Life Cell Imaging Device. The impactful mutants of *PIK3CA* or *PIK3R1* implicated in doxorubicin and BKM120 response promote cell proliferation in growth medium containing doxorubicin or BKM120. In contrast, the growth of non-impactful mutants of *PIK3CA* (E710G, K51N) or *PIK3R1* (D440G) were repressed (**Fig. R5**).

Figure R5

Figure R5. (A and B) Growth curves of cells expressing *PIK3CA* wild type and mutations with Doxorubicin or BKM120; (C and D) Growth curves of cells expressing *PIK3R1* wild type and mutations with Doxorubicin or BKM120. Five wells were measured per condition. Error bars represent mean \pm s.d. derived from three independent experiments.

Minor points:

1. The authors do not include molecular subtyping in their discussions. For example, the *PIK3CA*/E39K and *PIK3R1*/Q329L mutations were identified in HER2 positive breast cancer patients. It would be of interest to test whether these impactful mutations confer resistance to HER2 targeting therapies in a HER2 positive cell line.

Response: We thank the reviewer’s suggestion. Although *PIK3CA* activating mutations are present in all breast cancer subtypes, they are more prevalent in luminal and HER2 subtypes and are less common in TNBCs³⁰. The PI3K pathway is a crucial part of HER2 downstream signaling, and has been implicated in mediating resistance to HER2 targeting therapies³¹. Several retrospective analyses in clinical trials supported that HER2 positive breast carcinomas with a *PIK3CA* mutation are more likely to develop resistance to trastuzumab treatment^{32, 33}. Considering the possibility that activating mutations in PI3K signaling might be a key regulator for the resistance of trastuzumab or lapatinib in breast cancer, we investigated the effect of HER2 targeting therapies upon *PIK3CA* and *PIK3R1* combined mutation transduced in SK-BR-3, a HER2 positive breast cancer cell line. As shown in **Fig. R6**, the impactful mutation *PIK3CA* E39K combined with *PIK3R1* wild type or impactful mutation Q329L can confer resistance to HER2 targeting agent lapatinib compared to the wild type combination.

Figure R6. Cell viability assay in SK-BR-3 cells expressing the combined *PIK3CA* and *PIK3R1* mutations treated with increasing doses of lapatinib. Error bars represent mean \pm s.d. derived from three independent experiments.

2. *PIK3CA* E453K mutant is not depicted in Figure 2B

Response: We thank the reviewer's kind reminder. We did not identify any *PIK3CA* E453K mutations in the FDUSCC cohort (149 cases), which is why it is not depicted in the figure. However, the mutant was reported in the TCGA and Cosmic datasets, and is depicted accordingly (**Original Figure 2b**, under the diagram for *PIK3CA* in purple).

3. Figure 7C,E,G: it would be clearer for the readers if similar color coding (for mut combo, WT/MUT etc) of the legends is used.

Response: We are grateful for the reviewer's helpful suggestions. We have changed the color coding in the revised version. We hope that the new version will be much clearer for the readers.

4. "Multiple mutations in the PI3K/AKT pathway in breast cancer have not been reported in other datasets" (sentence 321-322). According to Pereira et al Nature Communications 2016 *PIK3CA* or *PIK3R1* and PTEN aberrations co-occur.

Response: We are immensely sorry for having overlooked this article during our background research, and we have cited this paper in the revised version of our manuscript²². Thank you again for your timely reminder.

Reviewer #3 (Reviewer Comments to the Author):

This study uses a targeted sequencing approach to identify somatic mutations in PI3K pathway genes in tumours from a cohort of Chinese breast cancer patients. The authors then generate a retroviral library of the identified mutations and perform a high-throughput mutation-phenotype screen to identify the specific mutations that functionally contribute to cell proliferation and drug resistance. This is an interesting study. While the use of individually bar-coded mutation libraries for high-throughput functional screens is not in itself new, the application of such an approach to screen for *PIK3CA* and *PIK3R1* mutations that drive proliferation and drug response is, to my knowledge, novel. The results, although perhaps not altogether surprising, are interesting and should be of general interest to researchers in the field. However, there are, in my mind, three main weaknesses in this study.

1. There is a general lack of information about the reproducibility of the data. Indeed, much of the data presented appears to be from single experiments (e.g. for the cell growth experiments the figure legends indicate that “Five wells were measured per condition” suggesting the results are the mean (and SD?) of 5 technical replicates from a single experiment). The data would be much more convincing if similar results were obtained in multiple independent experiments.

Response: We appreciate the reviewer’s comment, and we would like to clarify that in this study, most experiments were repeated 3 times. For example, 3 replicates of the cell proliferation assays are as presented in **Fig. R7**. A representative image is shown for each assay in the manuscript. Error bars represent mean \pm s.d..

Figure R7

FigureR7. Growth curves of MCF-10A cells expressing *PIK3CA* and *PIK3R1* wild type, impactful mutations (250 cells/well) determined using Incucyte cell imaging system. (A, B and C): Growth curves of cells expressing *PIK3CA* wild type and impactful mutations in the absence of EGF; (D, E and F): Growth curves of cells expressing *PIK3R1* wild type and impactful mutations in the absence of EGF; (G, H and I): Growth curves of cells expressing *PIK3R1* wild type and impactful mutations with EGF; Five wells were measured per condition. Error bars represent mean \pm s.d..

2. The functional experiments validating the “impactful” mutations derived from the screens (figs 5 and 6) only test mutations found to be enriched in the screens. It seems to me that in order to properly validate the screens, some of the non-enriched mutations should also be tested and shown not to drive proliferation and/or drug resistance.

Response: According to reviewer’s well-founded suggestion, we further validated several non-enriched mutations in cell proliferation assays. As shown in **Fig. R8A-C**, the proliferation rates of the cells expressing non-enriched *PIK3CA* mutants (K51N, D538Y and E710G) were much lower those expressing driver mutation (E542K and H1047R) in the absence of EGF. Furthermore, the non-enriched *PIK3R1* mutants (I559N, D440G and G644C) could not promote proliferation in comparison to positive controls (K674R and N564D), whether in presence or absence of EGF (**Fig. R8D-I**). Five wells were measured per condition. Error bars represent mean \pm s.d..

Figure R8

Figure R8. Growth curves of MCF-10A cells expressing *PIK3CA* and *PIK3R1* wild type, impactful mutations and non-impactful mutations (250 cells/well) determined using Incucyte cell imaging system.
 (A, B and C): Growth curves of cells expressing *PIK3CA* wild type, impactful mutations and non-impactful mutations in the absence of EGF;
 (D, E and F): Growth curves of cells expressing *PIK3R1* wild type, impactful mutations and non-impactful mutations in the absence of EGF;
 (G, H and I): Growth curves of cells expressing *PIK3R1* wild type, impactful mutations and non-impactful mutations with EGF;
 Five wells were measured per condition. Error bars represent mean \pm s.d..

3. In my mind, the most interesting question addressed in this study is the possibility that specific *PIK3CA* and/or *PIK3R1* mutations may be responsible for resistance to treatment. Unfortunately, the data provided to support specific mutations driving drug resistance is not strong. While the qualitative western blot pAkt data is consistent with a decreased sensitivity to BKM120 in cells expressing selected *PIK3CA* or *PIK3R1* mutations, dose-response curves allowing quantitation of changes in IC₅₀ would be more convincing. Also, although mutations enriched by exposure doxorubicin are identified, no data is shown at all for the effects of specific mutations on sensitivity to this drug.

Response: We agreed with the reviewer’s suggestion, and have validated the response of functional mutations in *PIK3CA* and *PIK3R1* to doxorubicin and BKM120 treatments. MCF-10A cells stably

expressing *PIK3CA* wild type and mutations (impactful or non-impactful mutations) were plated in 96-well plates (500 cells/well) with indicated concentration of doxorubicin (1nM) or BKM120 (1μM). The confluency of the cultures was measured using an Incucyte Life Cell Imaging Device. The impactful mutants of *PIK3CA* or *PIK3R1* implicated in doxorubicin and BKM120 response promoted cell proliferation in growth medium containing doxorubicin or BKM120. In contrast, the growth of non-impactful mutants of *PIK3CA* (E710G, K51N) or *PIK3R1* (D440G) were repressed (**Fig. R9**).

Figure R9

Figure R9. (A and B) Growth curves of MCF-10A cells expressing *PIK3CA* wild type and mutations with Doxorubicin or BKM120; (C and D) Growth curves of MCF-10A cells expressing *PIK3R1* wild type and mutations with Doxorubicin or BKM120. Five wells were measured per condition. Error bars represent mean \pm s.d. derived from three independent experiments.

Other minor specific comments

1. Line 289: “Most somatic mutations were validated using...”. This is ambiguous. Does it mean that validation was not attempted for all mutations identified or that some mutations did not validate? Were all 75 somatic mutations listed in table S2 validated?

Response: We apologize for our negligence to provide supplemental information in the previous manuscript. To verify *PIK3CA* and *PIK3R1* mutations, we also performed Pyrosequencing for breast cancer tissue DNA to validate the mutations identified by Ion Torrent PGM platform (**Fig. R10**). And Sanger sequencings were performed for paired blood DNA to remove the interference from germline mutations.

Figure R10

Figure R10. Mutations validation via Pyrosequencing for breast cancer tissue DNA and Sanger sequencing for paired blood DNA, representively. (A) *PIK3CA*, (B) *PIK3R1*.

2. Lines 332-333: “Based on the mutation profiles of the TCGA, COSMIC and FDUSCC cohorts, a set of *PIK3CA* and *PIK3R1* mutants were generated”. What mutations were included in this “set” and how were they selected?

Response: To answer the reviewer’s question, the detailed information is displayed in the following pie charts. In order to assess the functional characteristics of *PIK3CA* mutations, we selected 94 representative mutations in COSMIC database whose breast cancer frequencies ranged from < 0.1% (present in one specimens) to 2%, as well as the hotspot mutations (E542K, E545K and H1047R/L) that ranged from 5.4% to 53.8 % of total *PIK3CA* mutations, respectively (Fig. R11A). We also included 10 novel mutations detected in our cohort. A total of 104 missense mutations in *PIK3CA* were chosen for functional ReMB screening within or proximal to the ABD, RBD, C2, Helical, and Kinase domains of p110 α . For the mutations in *PIK3R1* library, we selected 16 missense mutations from COSMIC database, as well as 11 novel mutations detected in our cohort (Fig. R11B).

Figure R11

Figure R11. *PIK3CA* and *PIK3R1* mutations profiles. (A) *PIK3CA* mutations profile and corresponding frequencies, including 94 representative mutations in COSMIC database whose breast cancer frequencies ranged 0.02% to 54% as well as the E545K and H1047R hotspot mutations. The 10 novel mutations (in red) detected in our cohort were added to the mutations profile. (B) *PIK3R1* mutations profile and corresponding frequencies, subsuming 16 missense mutations in COSMIC database, as well as 11 novel mutations (in red) detected in our cohort.

3. Lines 505-507: “the mutation spectrum in the Chinese cohort on which this study was based was distinct from the spectra reported in the TCGA and COSMIC datasets”. Do the authors have any thoughts on why this might be the case?

Response: We appreciated the reviewer’s kind comment. The mutation spectrum in the current Chinese cohort was different from that in the TCGA and COSMIC datasets in some genes. Several factors might contribute to this difference, as follow:

First, the sequencing platform differs between the current study and the others. Different platforms may bring technical biases to the mutational frequency during comparison between different cohorts. In recent years, next generation sequencing (NGS) technology has revolutionized genomic and genetic research. For example, targeted sequencing focusing on specific regions of a genome allows us to sequence

target genes with a high level of coverage without generating significant quantities of off-target data, providing a highly efficient way to find biologically and pathologically relevant variants. Ion Torrent PGM and Illumina HiSeq/MiSeq are two major sequencing platforms currently in use. Quail et al sequenced the microbial genomes of *Bordetella pertussis* (67.7% GC), *Salmonella Pullorum* (52% GC), *Staphylococcus aureus* (33% GC) and *Plasmodium falciparum* (19.3% GC), and reported that the data using the Ion Torrent PGM platform had a higher raw error rate (~1.8%) compared to Illumina data (0.80%). However, if there is sufficient coverage, the representation and ability to identify SNPs was closely matched between PGM and Illumina, with more true positives being identified in PGM data, but with far fewer false positives in the Illumina data¹. Thus, we feel confident in pronouncing that the Ion Torrent PGM sequencing with coverage of 1000× conducted in our study is a reliable targeted sequencing platform, that may identify more genetic alteration in specimens.

Second, we think that the limited cohort size may mainly contribute to bias. This study retrospectively obtained a total of 149 pathologically confirmed primary breast cancer samples from the Department of Breast Surgery at Fudan University Shanghai Cancer Center (FDUSCC) to examine somatic mutation via targeted sequencing. The relatively small number of tumors sequenced may cause the difference observed in comparison to the spectra reported in the TCGA and COSMIC datasets. In the future, we plan to perform an analysis of somatic mutations in the PI3K/AKT/mTOR (PAM) pathway in the multi-centered BOLERO-5 clinical trial. We hope that our future works will provide a comprehensive mutation profile of the PI3K pathway in a Chinese population.

Third, the difference in distribution of breast cancer subtypes in various ethnicities may contribute to the difference of mutation spectra between our study and TCGA and COSMIC datasets. The breast cancer cases were categorized into luminal-like, human epidermal growth factor receptor 2 (HER2)-enriched, and triple-negative subtypes according to their estrogen receptor (ER), progesterone receptor (PR), and HER2 statuses. The *PIK3CA* mutations are frequent events in luminal and HER2-positive subtypes, but are less common in TNBCs. However, there were some differences in the distribution of molecular subtypes by the stratification of race/ethnicity. Chlebowski et al reported that African American women had lower incidences of ER-positive (57%) and PR-positive (41%) tumors than white women (ER, 77% ; PR, 63%, respectively) in the SEER dataset²³. Our previous study reported that the proportion of hormone-dependent breast cancer was relatively lower in Chinese females (50–60%) compared to females from western countries^{24,25}. This difference may also contribute to the distinctive mutation profile observed in Chinese population.

4. Line 532: “Similar to the results of a previous study...”. Which previous study? Reference?

Response: We thank the reviewer for pointing out our error. We have added the references concerning *PIK3CA* E542K, E545K and H1047R/L mutations at the corresponding site in the revised version^{3, 4, 6, 34}.

5. Line 542: “PETN” should be “PTEN”

Response: We appreciate the reviewer’s correction of our mistake. We have changed “PETN” to “PTEN” in the revised version.

6. Line 548: “Consistent with the results of previous work...”. Again, which previous work? Reference(s)?

Response: Once again, we apologize for the omission, We have added the references concerning activating *PIK3R1* D560Y and N564D mutations at the corresponding site in the revised version¹⁰.

7. Lines 597-598: “Thus, the sensitivity of breast cancer cells bearing PI3K mutations such as BKM120 and BYL719 to PI3Ki remains to be verified” presumably should be “Thus, the sensitivity of breast cancer cells bearing PI3K mutations to PI3Ki such as BKM120 and BYL719 remains to be verified”.

Response: We thank the reviewer’s advice, and have amended the statement as suggested in the revised version.

8. Figures 5F-H: Given that potential differences in exposure time can influence the apparent intensity of bands on a western blot, the fact that many of the mutations are on different gels makes it difficult to accurately discern the extent to which some mutations (e.g. E39K, M1043V, and similar) activate pAkt. While the number of mutations investigated makes the use of multiple gels unavoidable, it is a pity that a control sample (e.g. WT) was not included on all gels to enable comparison across blots. In the absence of such a consistent control, care must be taken in interpreting the relative activities of the different mutations.

Response: We are grateful for the reviewer’s helpful comment. According to the reviewer’s advice, we have repeated the Western blot with a control sample (WT) on all gels for better comparison (**Fig. R12**). The results were consistent with the previous ones, and we updated the new figures into the revised manuscript.

Figure R12

Figure R12. Activation status of PI3K pathway components in the indicated cells. MCF10A cells expressing wild-type or mutations of *PIK3CA* were seeded in growth medium for 24 hours and then cultured in starvation medium (-) or in growth medium (G) containing EGF and insulin for an additional 24 hours. The cells were immunoblotted with the indicated antibodies.

9. Figures 5C and 6C: It is not clear to me what value the images of the 3D structures add given the little information provided to help interpret them (For example, what do the different colours represent? Where are they various important sites within the structures, e.g. catalytic cleft, substrate binding site, p85 binding site, etc.?). What are these images intended to illustrate?

Response: We apologize for our ambiguous illustration of the 3D structures. The colors in the two figures represent different driver mutations among *PIK3CA* and *PIK3R1* proteins. In order to make the images easier to comprehend, in the revised version of our manuscript, we have labeled the driver mutations in red.

References:

1. Quail MA, *et al.* A tale of three next generation sequencing platforms: comparison of Ion Torrent, Pacific Biosciences and Illumina MiSeq sequencers. *BMC genomics* **13**, 341 (2012).
2. Sabine VS, *et al.* Mutational analysis of PI3K/AKT signaling pathway in tamoxifen exemestane adjuvant multinational pathology study. *Journal of clinical oncology : official journal of the American Society of Clinical Oncology* **32**, 2951-2958 (2014).
3. Huang CH, *et al.* The structure of a human p110alpha/p85alpha complex elucidates the effects of oncogenic PI3Kalpha mutations. *Science (New York, NY)* **318**, 1744-1748 (2007).
4. Zhao L, Vogt PK. Helical domain and kinase domain mutations in p110alpha of phosphatidylinositol 3-kinase induce gain of function by different mechanisms. *Proceedings of the National Academy of Sciences of the United States of America* **105**, 2652-2657 (2008).
5. Zhao JJ, Liu Z, Wang L, Shin E, Loda MF, Roberts TM. The oncogenic properties of mutant p110alpha and p110beta phosphatidylinositol 3-kinases in human mammary epithelial cells. *Proceedings of the National Academy of Sciences of the United States of America* **102**, 18443-18448 (2005).
6. Kang S, Bader AG, Vogt PK. Phosphatidylinositol 3-kinase mutations identified in human cancer are oncogenic. *Proceedings of the National Academy of Sciences of the United States of America* **102**, 802-807 (2005).
7. Pacold ME, *et al.* Crystal structure and functional analysis of Ras binding to its effector phosphoinositide 3-kinase gamma. *Cell* **103**, 931-943 (2000).
8. Chan TO, *et al.* Small GTPases and tyrosine kinases coregulate a molecular switch in the phosphoinositide 3-kinase regulatory subunit. *Cancer cell* **1**, 181-191 (2002).
9. Cheung LW, *et al.* High frequency of PIK3R1 and PIK3R2 mutations in endometrial cancer elucidates a novel mechanism for regulation of PTEN protein stability. *Cancer Discov* **1**, 170-185 (2011).
10. Jaiswal BS, *et al.* Somatic mutations in p85alpha promote tumorigenesis through class IA PI3K activation. *Cancer cell* **16**, 463-474 (2009).
11. Cheung LW, *et al.* Naturally occurring neomorphic PIK3R1 mutations activate the MAPK pathway, dictating therapeutic response to MAPK pathway inhibitors. *Cancer cell* **26**, 479-494 (2014).
12. Jones RL, *et al.* A randomised pilot Phase II study of doxorubicin and cyclophosphamide (AC) or epirubicin and cyclophosphamide (EC) given 2 weekly with pegfilgrastim (accelerated) vs 3 weekly (standard) for women with early breast cancer. *British journal of cancer* **100**, 305-310 (2009).
13. Gandhi S, *et al.* Adjuvant chemotherapy for early female breast cancer: a systematic review of the evidence for the 2014 Cancer Care Ontario systemic therapy guideline. *Current oncology* **22**, S82-94 (2015).
14. O'Driscoll L, Clynes M. Biomarkers and multiple drug resistance in breast cancer. *Current cancer drug targets* **6**, 365-384 (2006).
15. Wijdeven RH, *et al.* Genome-wide identification and characterization of novel factors conferring resistance to topoisomerase II poisons in cancer. *Cancer research* **75**, 4176-4187 (2015).
16. Tacar O, Sriamornsak P, Dass CR. Doxorubicin: an update on anticancer molecular action, toxicity and novel drug delivery systems. *The Journal of pharmacy and pharmacology* **65**, 157-170 (2013).
17. Burger MT, *et al.* Identification of NVP-BKM120 as a Potent, Selective, Orally Bioavailable Class I PI3 Kinase Inhibitor for Treating Cancer. *ACS medicinal chemistry letters* **2**, 774-779 (2011).
18. Maira SM, *et al.* Identification and characterization of NVP-BKM120, an orally available pan-class I PI3-kinase inhibitor. *Molecular cancer therapeutics* **11**, 317-328 (2012).
19. Mayer IA, *et al.* Stand up to cancer phase Ib study of pan-phosphoinositide-3-kinase inhibitor buparlisib with letrozole in estrogen receptor-positive/human epidermal growth factor receptor 2-negative metastatic breast cancer. *Journal of clinical oncology : official journal of the American Society of Clinical Oncology* **32**, 1202-1209 (2014).

20. Bendell JC, *et al.* Phase I, dose-escalation study of BKM120, an oral pan-Class I PI3K inhibitor, in patients with advanced solid tumors. *Journal of clinical oncology : official journal of the American Society of Clinical Oncology* **30**, 282-290 (2012).
21. Juric D, *et al.* Convergent loss of PTEN leads to clinical resistance to a PI(3)K inhibitor. *Nature* **518**, 240-244 (2015).
22. Pereira B, *et al.* The somatic mutation profiles of 2,433 breast cancers refines their genomic and transcriptomic landscapes. *Nature communications* **7**, 11479 (2016).
23. Chlebowski RT, *et al.* Ethnicity and breast cancer: factors influencing differences in incidence and outcome. *Journal of the National Cancer Institute* **97**, 439-448 (2005).
24. Fan L, *et al.* Breast cancer in China. *The Lancet Oncology* **15**, e279-289 (2014).
25. Fan L, *et al.* Breast cancer in a transitional society over 18 years: trends and present status in Shanghai, China. *Breast cancer research and treatment* **117**, 409-416 (2009).
26. Zheng S, *et al.* The pathologic characteristics of breast cancer in China and its shift during 1999-2008: a national-wide multicenter cross-sectional image over 10 years. *International journal of cancer* **131**, 2622-2631 (2012).
27. Goldhirsch A, Wood WC, Coates AS, Gelber RD, Thurlimann B, Senn HJ. Strategies for subtypes--dealing with the diversity of breast cancer: highlights of the St. Gallen International Expert Consensus on the Primary Therapy of Early Breast Cancer 2011. *Annals of oncology : official journal of the European Society for Medical Oncology / ESMO* **22**, 1736-1747 (2011).
28. Ades F, *et al.* Luminal B breast cancer: molecular characterization, clinical management, and future perspectives. *Journal of clinical oncology : official journal of the American Society of Clinical Oncology* **32**, 2794-2803 (2014).
29. Miller TW, Balko JM, Arteaga CL. Phosphatidylinositol 3-kinase and antiestrogen resistance in breast cancer. *Journal of clinical oncology : official journal of the American Society of Clinical Oncology* **29**, 4452-4461 (2011).
30. Comprehensive molecular portraits of human breast tumours. *Nature* **490**, 61-70 (2012).
31. Berns K, *et al.* A functional genetic approach identifies the PI3K pathway as a major determinant of trastuzumab resistance in breast cancer. *Cancer cell* **12**, 395-402 (2007).
32. Loibl S, *et al.* PIK3CA mutations are associated with lower rates of pathologic complete response to anti-human epidermal growth factor receptor 2 (her2) therapy in primary HER2-overexpressing breast cancer. *Journal of clinical oncology : official journal of the American Society of Clinical Oncology* **32**, 3212-3220 (2014).
33. Majewski IJ, *et al.* PIK3CA mutations are associated with decreased benefit to neoadjuvant human epidermal growth factor receptor 2-targeted therapies in breast cancer. *Journal of clinical oncology : official journal of the American Society of Clinical Oncology* **33**, 1334-1339 (2015).
34. Isakoff SJ, *et al.* Breast cancer-associated PIK3CA mutations are oncogenic in mammary epithelial cells. *Cancer research* **65**, 10992-11000 (2005).

Reviewers' comments:

Reviewer #1 (Remarks to the Author):

This revision of the manuscript by Chen, et al. Is an improvement over the previous but still has some major publication issues for this reviewer that is fundamentally over two issues:

1. The removal of the TP53 mutation rate from the discussion does not erase the concern that the triple negative subset may be over diagnosed in the Chinese institution. the low TP53 mutation and the high PIK3CA seen in their data is significantly different from what is seen in other studies. This raises a serious question as to whether the clinical laboratory assessment of ER/PR/HER2 status is substandard. This concern raises suspicion that the IHC-based classifications from this group are incorrect.

2. The novelty of this work is not the frequency of PIK3CA mutations or their co-occurrence with PIK3R1, but the functional characterization of the mutations into impactful and non-impactful. The authors seem to downplay this in their writing wanting to focus on the frequency questions. For example, they never mention that only a portion of the PIK3CA mutations are "impactful". More importantly however, is that they avoided the more important analysis of what makes some mutations impactful and others not. For example, are there structural differences between the impactful and the non-impactful? Is the reason for the differences mainly because the expression from the artificial expression vectors is higher in the impactful vs. non-impactful? Moreover, are there consequences to the impactful mutations clinically or otherwise. They write eloquently about the clinical issues in the rebuttal (mostly negative data), but i do not see much mention of this in the revised manuscript. This also raises then the question of whether the assay is any good as a screen for any impactful phenotype. For example, if indeed PIK3CA mutations are associated with Doxorubicin as they allege, then is there any evidence that these individuals are indeed resistant. We are not given any information as to whether these individuals were given adjuvant chemotherapy (or what kind).

This clinical and functional validation is the key. The reason why this is important is that if published, the mutations and their functional screen output will be used by others, especially in clinical diagnosis, to ascertain the importance of such variants of unknown significance. Incorrect data in such databases have been the bane of the oncological pathology community.

Reviewer #2 (Remarks to the Author):

The authors have sufficiently addressed the comments in this revised version except for one point:

From the (repeated) Western blot in Figure 7D it cannot be concluded that "The combination of PIK3CA H1047R and PIK3R1 K674R mutations cooperatively promoted cell proliferation and significantly increased the activation of AKT signaling pathway". The PIK3CA/H1047R + PIK3R1/WT combination already results in maximal p473AKT levels which do not increase in the PIK3CA/H1047R + PIK3R1/K674R combination. As such, the repeated Western gives the same result as displayed in the original submission and does not support the conclusions.

Authors should include quantification of Western to convincingly demonstrate increased pAKT levels or adjust their statements.

Reviewer #3 (Remarks to the Author):

In my original review I indicated that I felt there were three main weaknesses in this study.

1. A general lack of information about the reproducibility of the data.

In their response, the authors indicated that they had indeed carried out 3 independent experiments for the cell growth experiments (and provided the data) but stated that they have only shown a representative experiment in the manuscript. That is fine but this should be clear in each of the figure legends (e.g. Shown are mean \pm SD of 5 technical replicates from an experiment that is representative of 3 performed). Similarly, for western blots (e.g. Shown are representative blots from "n" independent experiments).

2. The lack of functional experiments on the non-enriched mutations.

This concern has been addressed by the authors who now include this data as Supplementary Figure 7.

3. Data provided to support specific mutations driving drug resistance is not strong / dose-response curves would be more convincing.

The authors have addressed this issue by providing functional data confirming that the "impactful" mutations do indeed mediate resistance to doxorubicin and BKM120. This data, which is included as Supplementary Figure 8, adequately addresses my concern although, personally, I would have preferred for it to have been included in the main manuscript as I feel it is interesting and important data that validates their screening strategy.

Other minor specific comments from my original review:

Line 289: "Most somatic mutations were validated using...". This is ambiguous. Does it mean that validation was not attempted for all mutations identified or that some mutations did not validate? Were all 75 somatic mutations listed in table S2 validated?

I think the authors may have misunderstood my question. I was simply asking for clarification of the wording. When the authors state "Most somatic mutations were validated using..." (now line 127), do they mean that they only attempted to validate a selection (most) of the mutations or did they attempt to validate all mutations they detected but only some (most) could be validated?

Lines 505-507: Differences in the mutation spectrum in the Chinese cohort compared to the TCGA and COSMIC datasets - Do the authors have any thoughts on why this might be the case?

The authors addressed this in their response to the reviewers but do not appear to have incorporated any of these thoughts into the discussion section of their manuscript.

Figures 5C and 6C: What are these images intended to illustrate? Where are they various important sites within the structures, etc.

I do not feel this comment has been adequately addressed in the revised manuscript.

My remaining issues/comments have been adequately addressed.

Point-by-point response to the referees' comments

On NCOMMS-17-00264A: "High-throughput barcode screening elucidates the functional characteristics and mutual relevance of PIK3CA and PIK3R1 somatic mutations in Chinese patients with breast cancer".

Reviewer #1 (Remarks to the Author):

This revision of the manuscript by Chen, et al. is an improvement over the previous but still has some major publication issues for this reviewer that is fundamentally over two issues:

1. The removal of the TP53 mutation rate from the discussion does not erase the concern that the triple negative subset may be over diagnosed in the Chinese institution. the low TP53 mutation and the high PIK3CA seen in their data is significantly different from what is seen in other studies. This raises a serious question as to whether the clinical laboratory assessment of ER/PR/HER2 status is substandard. This concern raises suspicion that the IHC-based classifications from this group are incorrect.

Response: We really appreciated the reviewer's comment for this important point. In the project, we used the clinical-pathological records of ER/PR/HER2 immunohistochemical (IHC) status to determine surrogate molecular subtypes for the patients who were subjected to pathologically diagnosis in Fudan University Shanghai Cancer Center (FUSCC) during 2007 to 2009. To address the concern regarding IHC-related bias, under the support of the Department of Pathology at FUSCC, we revalidated the ER, PR and HER2 status of all specimens (149 cases) which were subjected to Amplicon sequencing in the study.

The H.E staining of tumor tissue slides were retrieved and reviewed for determining breast carcinoma by two independent pathologists. The corresponding formalin-fixed paraffin-embedded blocks were selected from the central storage bank to make consecutive tissue slides for each case. The ER, PR and HER2 IHC staining procedures were performed on a Ventana Benchmark automated immunostainer (Tucson, Arizona, USA) by the standard streptavidin-biotin staining method (Antibodies: ER, Roche Ventana, Clone SP1; PR, Roche Ventana, Clone IE2; HER2, Roche Ventana, Clone 4B5)¹. ER and PR positivity illustrated nuclear reactivity, while HER2 positivity showed membranous reactivity. Mammary tumors were considered positive for ER or PR if strong immunoreactivity was observed in more than 1% of tumor nuclei, according to the 2010 ASCO/CAP

guidelines². HER2 status was initially evaluated by IHC on a scale of 0–3, combining the intensity of membranous staining and the percentage of staining of invasive tumor cells, according to the 2013 ASCO/CAP guidelines³. Only those tumors with strong and complete membranous staining in >10% of invasive carcinoma cells (score of 3+) were considered HER2 positive. Cases with scores of 0 or 1+ were identified as negative. Tumors with HER2 expression status (IHC, score equals to 2+) were further subjected to fluorescence in situ hybridization (FISH) test to determine the HER2 gene amplification with the FDA-approved PathVysion HER2/Neu DNA Probe Kit (Abbott Molecular, Abbott Park, Illinois, USA), according to manufacturer's instructions. The HER2 positivity subgroup was defined as FISH positive or IHC staining score equals to 3+, according to the 2013 ASCO/CAP guidelines³.

Based on the IHC revalidation staining for ER, PR and HER2, we found that the majority of ER status (146/149) was identical with the original (**revised Supplementary Table 15**); meanwhile, there were 3 samples underwent the variation of ER status from positive to negative. Similarly, most of PR status (145/149) was unchanged, with the exception of 4 samples turning to negative from positive. The HER2 status was in accordance with original records in 147 out of 149 patients. The original IHC images were illustrated in **manuscript-related Figure 1**. We found that most IHC readouts in fresh-cut slides were consistent with the previous IHC records. Importantly, the TNBC patients defined in the original data were also classified into TNBC according to the revalidation results.

Previous studies have reported that long-term storage of paraffin blocks does not significantly impact IHC staining in general. When compared with freshly embedded controls, paraffin blocks archived at room temperature for as little as 2 years or as long as 25 years exhibited stable immunostaining for most antigens evaluated^{4,5,6,7}. Jacobs TW et.al reported that paraffin coating of the blocks did not significantly diminish antigen loss at room temperature. However, slide storage at room temperature resulted in loss of p53 reactivity, with some p53-positive becoming p53 negative after 12 weeks of storage, as well as loss of immunoreactivity for ER at 12 weeks^{8,9}. The IHC revalidation of ER, PR and HER2, which was based on fresh-cut slides from aging paraffin coating blocks, are proved to be reliable in general.

In this study the majority of ER, PR and HER2 status were in accordance with the original records with rare exceptions of negative transformations from positivity. We think there may be two reasons contributing to the loss of immunoreactivity of some cases. First, the internal heterogeneity may exist

within tumors. Second, it remains possible that the intensity of immunohistochemical staining may decrease in some cases after long-time storage. Collectively, our pathologists recommended us to utilize original pathological records to determine the subtype classification in this study. We added this information in the revised **Supplementary Information**.

We believed that the limited cohort size may bring the bias to the low TP53 mutation frequency in TNBCs. Here, we took advantage of five public datasets to analyze the frequencies of *TP53* mutation in TNBC (**manuscript-related Table 1**). We notice that the somatic mutation rates for *TP53* in TNBCs are versatile in the studies with small sample size (36.3% (4/11), Broad¹⁰; 64.3% (9/14), Sanger¹¹ and 52.5% (31/59), British Columbia¹²), whereas two studies containing more cases report a higher rate (82.1% (69/84), 2012 TCGA¹³ and 80.7% (67/83), 2015 TCGA¹⁴). To further address this concern, we analyzed the *TP53* mutation rate among 37 TNBC cases diagnosed at the Department of Breast Surgery in FUSCC between 2013 and 2016. Amplicon sequencing was performed on these TNBC specimens and matched normal blood samples using the Ion Torrent PGM platform. Among these TNBC cases, *TP53* exhibits frequent somatic mutations (81%, 30/37, for total; 73%, 27/37, for missense). It is noteworthy that the combined *TP53* mutation rates (58%, 36/62, for total; 50%, 31/62, for missense, **manuscript-related Table 2**) from two FUSCC cohorts is comparable to those from TCGA datasets. These suggest the increment of cases may provide a more accurate mutation frequency for the genetic landscape in tumors. We hope these works will provide valuable information to state the *TP53* mutation frequency in TNBCs.

2. The novelty of this work is not the frequency of *PIK3CA* mutations or their co-occurrence with *PIK3RI*, but the functional characterization of the mutations into impactful and non-impactful.

(1) The authors seem to downplay this in their writing wanting to focus on the frequency questions. For example, they never mention that only a portion of the *PIK3CA* mutations are "impactful". More importantly however, is that they avoided the more important analysis of what makes some mutations impactful and others not. For example, are there structural differences between the impactful and the non-impactful? Is the reason for the differences mainly because the expression from the artificial expression vectors is higher in the impactful vs. non-impactful?

Response: We really appreciate the reviewer's kind reminders. In the revised manuscript, we highlighted the importance of functional characterization of the mutations, and discuss the potential molecular mechanisms.

According to the reviewer's reminder, we looked back on the mutations of *PIK3CA*. Of all 197 mutations of *PIK3CA*, including mutations in TCGA, COSMIC datasets and FUSCC cohort, there were 11 mutations identified to be impactful, and the other 186 were non-impactful. Given the situation, it is important to consider the differences between impactful mutations and non-impactful mutations. We hypothesized that the positional and structural differences between impactful and non-impactful mutations had strong influence on their functions. As reported, most mutations of *PIK3CA* and *PIK3RI* occurred at residues lying at the interfaces between p110 α and p85 α , or between the functional domains within p110 α , thus disrupting the regulation of kinase activity by p85 α or the catalytic activity of p110 α ¹⁵. **(1) p110 α -ABD domain and p110 α -kinase domain interface:** Past researches supported that the ABD domain not only binds to the iSH2 domain of p85 α , but also interacts with the PI4K kinase domain in p110 α ¹⁶. As previously reported¹⁵, Arg38 and Arg88 were located at the interface between the ABD and the kinase domains, within hydrogen-bonding distance of Gln738, Asp743 and Asp746 of kinase domain. Mutations of Arg38 and Arg88 were likely to disrupt the interactions and result in conformational change of p110 α -kinase domain. The E39K mutation identified in the current study has not been reported elsewhere. We supposed that alterations at residue Glu39 of ABD domain of p110 α either directly affected interactions with the kinase domain or functioned by changing the construct of Arg38. **(2) p110 α -C2 domain and p85 α -iSH2 domain interface:** The C2 domain has been postulated to link p110 α to the plasma membrane. Asn345 and Glu542 substitutions in the C2 domain increase the positive surface charge of the domain and are thought to facilitate its localization to the cell membrane,

making lipid kinase activity independent of upstream signaling¹⁷. Moreover, Asn345 in the C2 domain of wild type p110 α is within hydrogen bonding distance of Asp560 and Asn564 of the p85 α -iSH2 domain. The Asn345 mutation may disrupt the interaction of the C2 domain with iSH2, thus altering the regulatory effect of p85 α on p110 α ¹⁵. Accordingly, mutations residing in Asp560 and Asn564 of p85 α -iSH2 domain may also interfere in interactions with C2 domain and the activation of p110 α ^{18,19}. We found that N564D could promote cell proliferation or induce drug resistance in mammary cells. Huang et.al proposed that loss of residues 572-600 in the truncation mutant might destabilize the iSH2 domain coiled-coil at the C2 domain contact sites¹⁵. Wu et.al reported that the truncated mutation at residue Q572 of iSH2 domain couldn't inhibit the activity of wild type p110 α ¹⁸. We assumed that the impactful R574T mutation in the current study mimicked the effect. **(3) p110 α -Helical domain and p85 α -nSH2 domain interface:** The helical domain of p110 α mediates interaction with the nSH2 domain of p85 α , which is responsible for p85 α -induced inhibition of p110 α . These impactful mutations in the helical domain could interfere with this p85 α -p110 α interaction and disrupt signal regulation^{15,17,20}. Glu542 and Glu545 interact with Lys379 and Arg340 of the p85 α -nSH2 domain, which in turn inhibits the activity of the catalytic subunit. E542K and E545K mutations cause a charge reversal and abrogate the inhibitory effect of nSH2^{15,21}. **(4) p110 α -Kinase domain:** Interaction of Ras with p110 α is known to increase the kinase activity of p110 α , providing a conformational change for substrate binding²². Ignacia Echeverria et.al reported hotspot mutations at residue His1047 (H1047R) in the kinase domain changed the orientation of His1047, which resulted in easier access to membrane-bound phosphoinositol-4,5-bisphosphate substrate of *PIK3CA*²³. Additionally, Paraskevi Gkeka et. al reported that the hydrogen bond between Met1043 and His1047, and also between Gly1049 and Asn1044 contributed in the stabilization of kinase domain of wild type *PIK3CA*, whereas the hydrogen bond was disrupted in H1047R mutant and consequently, residue His917 pointed towards the active site, which facilitated ATP hydrolysis²⁴. We speculate that mutations at residue Met1043 and Gly1049 could promote *PIK3CA* activity in a similar way. **(5) p85 α -Rho-GAP domain and PTEN:** SH3 and Rho-GAP domain is responsible for PTEN binding²⁵. Cheung et al. demonstrated that the *PIK3R1* E160* mutation disrupted the interaction between p85 α and PTEN, resulting in destabilization of PTEN, increased PI3K signaling and transformation²⁶. The ReMB screen showed that E160D had clonal advantages in cell proliferation and drug response assays, suggesting that novel E160D mutation might be a functional hotspot in the Rho-GAP domain of *PIK3R1*.

Taken together, we infer that the positional and structural differences mostly matter between impactful and non-impactful mutations. These impactful mutations were considered to be structural damaging alteration as disease-causing drivers. In contrast, non-impactful mutations were considered as neutral or benign variants without deleterious alteration. However, the mechanisms of some impactful mutations such as Q329L and K674R in the linker region in *PIK3RI* still remains unclear and further studies should be conducted. According to reviewer's kind suggestion, we have added this information in the revised **Results** and **Discussion**. (Line 237-256, 311-326 and 477-509)

In this study, we have developed a method called ReMB that enables a more rapid and efficient screening of mutations library. The pooled library consists of different mutant, wild-type and control expression vectors with exact identical amount to eliminate the difference in expression level. Using viral transduction, the multiplicity of infection (MOI) can be kept low so that that most cells receive a single virus that is stably integrated. We took advantage of the unique barcodes to de-convolute pooled mutations using polymerase chain reaction (PCR) and NGS methods. This strategy is sought to be a reliable method that allow multiplex analysis of phenotypic outputs on a large-scale pooled screen²⁷. This strategy is widely applied in pooled RNAi^{28, 29} or CRISPR^{30, 31} screens. We have added the information in the revised version. (Line 575-585)

(2) Moreover, are there consequences to the impactful mutations clinically or otherwise. They write eloquently about the clinical issues in the rebuttal (mostly negative data), but i do not see much mention of this in the revised manuscript. This also raises then the question of whether the assay is any good as a screen for any impactful phenotype. For example, if indeed PIK3CA mutations are associated with Doxorubicin as they allege, then is there any evidence that these individuals are indeed resistant. We are not given any information as to whether these individuals were given adjuvant chemotherapy (or what kind). This clinical and functional validation is the key. The reason why this is important is that if published, the mutations and their functional screen output will be used by others, especially in clinical diagnosis, to ascertain the importance of such variants of unknown significance. Incorrect data in such databases have been the bane of the oncological pathology community.

Response: We apologize for failing to mention the clinical implications of our findings in the last version. According to reviewer's kind suggestion, we added a new section to discuss the clinical significance of these impactful mutations in the revised manuscript.

We analyzed the relationship between mutation status and clinical-pathological characteristics in the TCGA dataset. Among 1105 breast cancer cases, the clinical information of 977 cases was obtained. 317 of 977 patients (32.4%) harbored *PIK3CA* missense mutation (**revised Table 1**). We found that patients diagnosed after the age of 50 years showed a higher rate of *PIK3CA* mutation than younger patients ($P = 0.032$). Breast tumors with mutated *PIK3CA* were more likely to be ER positive ($P < 0.001$) and PR positive ($P < 0.001$) compared to cases without *PIK3CA* mutation. Furthermore, we classified all patients into three categories (no mutation, non-impactful mutation and impactful mutation). Among the 977 breast cancer patients, 55 patients (5.6%) had *PIK3CA* non-impactful mutation, and 262 patients (26.8%) had *PIK3CA* impactful mutations. *PIK3CA* impactful mutations were closely associated with the expression of hormone receptors (ER, $P < 0.001$; PR, $P < 0.001$), meanwhile non-impactful mutations were not correlated with hormone receptors status.

We further investigated the relationship between the clinical-pathological characteristics and *PIK3CA* mutations in the FUSCC cohort. As there were insufficient cases in the FUSCC cohort, we observed that *PIK3CA* mutations displayed a trending correlation with ER status ($P = 0.060$, **revised Supplementary Table 13**). These findings suggest that the high frequency of *PIK3CA* impactful mutations may contribute to the hyper-activated PI3K signaling in the luminal subtype. Due to the

limited number of genetic alteration events in *PIK3RI*, the clinical significance analysis was not conducted in this study.

We further categorized patients carrying *PIK3CA* impactful mutation into the subgroups according to the locations of their mutations. Since impactful mutations reside mainly in the C2, helical and PI4K kinase domains, we analyzed the relationships between the mutated domains and clinical-pathological characteristics (**revised Table 2**). When compared against patients who harbored wild-type or non-impactful mutations, patients with helical domain mutation were more likely to be diagnosed after the age of 50 years ($P = 0.025$). Notably, breast cancers with impactful mutations in C2 domain (21 cases, 2.1%), helical domain (105 cases, 10.7%) or PI4K domain (138 cases, 14.1%) were more likely to be ER- and PR- positive ($P = 0.032$, $P < 0.001$, $P < 0.001$, for ER; $P = 0.030$, $P < 0.001$, $P < 0.001$, for PR; respectively). No other significant association was found with mutated *PIK3CA* domains. Unfortunately, the sample size in FUSCC BC cohort were too small to meet statistical significance (**revised Supplementary Table 14**).

To assess the clinical significance, we also investigated the relationship between mutation status and survivals in the TCGA dataset. Patients whose tumors contained *PIK3CA* mutation had similar clinical outcomes to patients without *PIK3CA* mutations ($P > 0.05$, **revised Supplementary Fig. 8a,b**). Additionally, there was no significant differences in disease-free survival (DFS) and overall survival (OS) between the *PIK3CA* wild type, impactful mutation and non-impactful mutation carriers (**revised Supplementary Fig. 8c,d**). Similarly, we observed no statistically significant differences between impactful mutation carriers and other subgroups in the FUSCC cohort (**revised Supplementary Fig. 8e-h**).

We further analyzed the correlations between the *PI3KCA* impactful mutations and doxorubicin benefit in the FUSCC cohort. All therapeutic regimen decisions were based on the Chinese Anti-Cancer Association guidelines for the treatment of breast cancer. According to the guidelines, doxorubicin or doxorubicin-based treatments were one of the recommended adjuvant chemotherapies for primary breast cancer during 2007-2009. Therefore, 137 patients underwent a mastectomy and axillary lymph node dissection or breast conservation surgery followed by doxorubicin-based adjuvant chemotherapy (**revised Supplementary Table 13**). When patients were assigned to subgroups on the basis of *PIK3CA* impactful mutation status, *PIK3CA* impactful mutations showed no significant effect upon the DFS or OS of patients undergoing doxorubicin treatment (**revised Supplementary Fig. 9a,b**). Though

these results are from a relatively small subset of patients, they suggest that *PIK3CA* impactful mutations were unlikely to be associated with doxorubicin efficacy. However, the effects of subsequent radiotherapy and hormone therapy could not be eliminated, and the association between impactful mutations and chemoresistance requires further investigation.

In conclusion, most *PIK3CA* impactful mutations could be proved as oncogenic alteration in functional validations, suggesting PI3K pathway is an attractive therapeutic target with a high frequency of impactful mutations. We observed strong correlations between the distributions *PIK3CA* impactful mutations and hormone receptor positivity in breast tumors, whereas this relationship did not seem to hold for non-impactful mutations. However, impactful *PIK3CA* mutations might not identify either a poor prognostic group or a group of doxorubicin-resistance in breast cancers. The equivocality remains over the association between *PIK3CA* mutations and their clinical significances in literatures. *PIK3CA* mutations in helical or kinase domains have been associated with poor prognosis^{32,33}, while Bartlett's group reported that *PIK3CA* mutations were associated with favorable outcomes in luminal subtype based on the TEAM clinical trial³⁴. Additionally, it remains possible and even likely that mechanisms of *de novo* drug resistance in early cancer differ from those in progressing cancer that achieve drug resistance by tumor evolution. Thus, a definitive conclusion awaits more comprehensive molecular profiling investigation in larger patient cohorts. We have added this information in the revised **Results and Discussion**. (Line 386-435 and 554-569)

Reviewer #2 (Remarks to the Author):

The authors have sufficiently addressed the comments in this revised version except for one point: From the (repeated) Western blot in Figure 7D it cannot be concluded that “The combination of PIK3CA H1047R and PIK3R1 K674R mutations cooperatively promoted cell proliferation and significantly increased the activation of AKT signaling pathway”. The PIK3CA/H1047R + PIK3R1/WT combination already results in maximal p473AKT levels which do not increase in the PIK3CA/H1047R + PIK3R1/K674R combination. As such, the repeated Western gives the same result as displayed in the original submission and does not support the conclusions. Authors should include quantification of Western to convincingly demonstrate increased pAKT levels or adjust their statements.

Response: We thank the reviewer for the kind reminders. According to the reviewer’s suggestions, we added the quantifications of western blots into the revised **Figures**. In the **revised Fig. 6d**, the intensity value of *PIK3CA* H1047R and *PIK3R1* WT did indicate a near maximal p473AKT level; the value of *PIK3CA* H1047R and *PIK3R1* K674R combination promoted p473AKT level slightly. As indicated, we rephrased our description in the revised manuscript, as follow: "The combination of *PIK3CA* H1047R and *PIK3R1* K674R mutations could promote cell proliferation significantly in comparison with the introduction of a single impactful mutation (**revised Fig. 6c**). Western blot analysis showed that p473AKT levels reached the maximum in the *PIK3CA* H1047R and *PIK3R1* K674R combination; meanwhile, the *PIK3CA* H1047R and *PIK3R1* WT combination resulted in a near maximal p473AKT level (**revised Fig. 6d**). "

Reviewer #3 (Remarks to the Author):

In my original review I indicated that I felt there were three main weaknesses in this study.

1. A general lack of information about the reproducibility of the data.

In their response, the authors indicated that they had indeed carried out 3 independent experiments for the cell growth experiments (and provided the data) but stated that they have only shown a representative experiment in the manuscript. That is fine but this should be clear in each of the figure legends (e.g. Shown are mean \pm SD of 5 technical replicates from an experiment that is representative of 3 performed). Similarly, for western blots (e.g. Shown are representative blots from “n” independent experiments).

Response: We are very grateful for the reviewer’s comments. Taking into account the reviewer’s suggestions, we rephrased the figure legends and stated the number of replicates for each experiment. Detail information was added to the corresponding legends of **revised Fig. 2b, 4d, 4f-j, 5d, 5f-i, 6c-h**.

2. The lack of functional experiments on the non-enriched mutations. This concern has been addressed by the authors who now include this data as Supplementary Figure 7.

Response: We appreciate the reviewer’s kind comment.

3. Data provided to support specific mutations driving drug resistance is not strong / dose-response curves would be more convincing.

The authors have addressed this issue by providing functional data confirming that the “impactful” mutations do indeed mediate resistance to doxorubicin and BKM120. This data, which is included as Supplementary Figure 8, adequately addresses my concern although, personally, I would have preferred for it to have been included in the main manuscript as I feel it is interesting and important data that validates their screening strategy.

Response: According to the reviewer’s well-founded suggestion, we have now included these results in the main manuscript. Detailed information was showed in **revised Fig. 4i,j and Fig. 5h,i**.

Other minor specific comments from my original review:

1. Line 289: “Most somatic mutations were validated using...”. This is ambiguous. Does it mean that validation was not attempted for all mutations identified or that some mutations did not validate? Were all 75 somatic mutations listed in table S2 validated? I think the authors may have misunderstood my question. I was simply asking for clarification of the wording. When the authors state “Most somatic mutations were validated using...” (now line 127), do they mean that they only attempted to validate a selection (most) of the mutations or did they attempt to validate all mutations they detected but only some (most) could be validated? did they attempt to validate all mutations they detected but only some (most) could be validated.

Response: We are so sorry for this ambiguity in our manuscript. Sanger sequencing of paired blood DNA was conducted in all cases to exclude germline mutations. For pyrosequencing, PCR products were processed according to the manufacturer’s instructions and submitted to a pyrosequencing assay on a Pyro Mark Q24 platform (Qiagen, Hilden, Germany). Pyrosequencing was able to detect an allele frequency difference of less than 5% between pools, indicating that this method may be sensitive enough for use in mutation detection³⁵. We used pyrosequencing to validate somatic mutations of *PIK3CA* and *PIK3RI* in 41 malignant tissues. 23 (23/28, 82%) somatic mutations in 41 cases could be validated successfully (**revised Supplementary Fig. 10**), suggesting that the amplicon sequencing using Ion Torrent PGM platform is a reliable method to detect low allele frequency mutations. In this study, we included the mutations with allele frequency > 3% via PGM platform into the following analysis. We have added this information in the **revised Methods**. (Line 635-644)

2. Lines 505-507: Differences in the mutation spectrum in the Chinese cohort compared to the TCGA and COSMIC datasets - Do the authors have any thoughts on why this might be the case?

The authors addressed this in their response to the reviewers but do not appear to have incorporated any of these thoughts into the discussion section of their manuscript.

Response: We appreciate the reviewer’s kind suggestion. In the revised manuscript, we have re-interpreted the differences in the mutation spectrum in the Chinese cohort compared to the TCGA and COSMIC datasets in the **revised Discussion**. (Line 458-473)

3. Figures 5C and 6C: What are these images intended to illustrate? Where are they various important

sites within the structures, etc. I do not feel this comment has been adequately addressed in the revised manuscript.

Response: We are very grateful for the reviewer's comments. To better interpret impactful mutations, we labeled these variants with different colors in according to their locations in 3D structures (**revised Fig. 4c and Fig. 5c**), and discussed the domains that are mutated in *PIK3CA* and *PIK3RI* showing specific phenotypes.

As shown in **revised Fig. 4c** and **revised Fig. 5c**, most impactful mutations occurred at the interface between p110 α and p85 α , or between kinase domain and other domains of p110 α , which was in accordance with previous reports. (1) **H1047R/L/T, G1049R and M1043V in p110 α -kinase domain:** Paraskevi Gkeka et. al reported that the hydrogen bond between His1047 and Met1043, between Gly1049 and Asn1044 kept the stabilization of kinase domain of wild type *PIK3CA*²⁴. However, the hydrogen bond was disrupted in H1047R mutant and consequently, residue His917 pointed towards the active site, which participated in ATP hydrolysis. We speculated that mutations at residue Met1043 and Gly1049 could mimic the effect and promote *PIK3CA* activity. (2) E542K, E545K in p110 α -helical domain and p85 α -nSH2 domain: Glu542 and Glu545 of p110 α interact with Lys379 and Arg340 of p85 α nSH2 domain, which inhibited the activity of p110 α . E542K and E545K mutations caused a charge reversal and abrogated the inhibitory effect of nSH2, thus activating p110 α ¹⁵. (3) **N345K/I, E453K in p110 α -C2 domain and N564D, D560Y in p85 α -iSH2 domain:** The C2 domain has been postulated to link p110 α to the plasma membrane. N345 and E453 substitutions in the C2 domain increase the positive surface charge of the domain and are thought to facilitate its localization to the cell membrane, making lipid kinase activity independent of upstream signaling¹⁷. In wild type p110 α , Asn345 in C2 domain is within hydrogen bonding distance of Asp560 and Asn564 of the iSH2 domain of p85 α . The Asn345 mutation may disrupt the interaction of the C2 domain with iSH2, thus altering the regulatory effect of p85 on p110 α ¹⁵. Similarly, mutations at residues Asp560 and Asn564 of the p85 α -iSH2 domain could induce activation of p110 α in a similar way¹⁹. We identified N345K/I and N564D, D560Y residing in interface between p110 α -C2 and p85 α -iSH2 domain was impactful mutations in the current study. (4) **E39K in p110 α -ABD domain and p110 α -kinase domain:** Arg38 and Arg88 at p110 α -ABD domain was reported to be interacted with p110 α -kinase domain. Mutations at Arg38 and Arg88 were likely to disrupt the interactions and resulted in conformational change of the kinase domain of p110 α ¹⁵. We speculated that the new identified E39K mutation in

p110 α -ABD domain either affected the interactions with the kinase domain directly or functioned by changing construct of Arg38. **(5) Q329L and K674R in linker regions of p85 α :** In this study, we also identified Q329L (at the nSH2 domain boundary), and K674R (at the cSH2 domain boundary) as mutations that affect cell proliferation and drug response. The potential mechanism of their impactful functions haven't been identified yet, and further investigation needs to be conducted. While, we supposed that functions of Q329L and K674R was caused by conformational changes of p85 α .

Together, we believed that the functions of p110 α and p85 α are closely correlated with their constructs and mutations that changed the conformation of p110 α - p85 α were likely to impactful. The revised figures and related discussion have been added in the **revised Results and Discussion**. (Line 237-256, 311-326 and 477-509) We hope that our revised manuscript will be much more complete and advanced than the original version.

4. My remaining issues/comments have been adequately addressed.

Response: We are very appreciative of the positive comments from the reviewer.

References:

1. Tang SX, *et al.* Characterisation of GATA3 expression in invasive breast cancer: differences in histological subtypes and immunohistochemically defined molecular subtypes. *Journal of clinical pathology*, (2017).
2. Hammond ME, *et al.* American Society of Clinical Oncology/College Of American Pathologists guideline recommendations for immunohistochemical testing of estrogen and progesterone receptors in breast cancer. *Journal of clinical oncology : official journal of the American Society of Clinical Oncology* **28**, 2784-2795 (2010).
3. Wolff AC, *et al.* Recommendations for human epidermal growth factor receptor 2 testing in breast cancer: American Society of Clinical Oncology/College of American Pathologists clinical practice guideline update. *Journal of clinical oncology : official journal of the American Society of Clinical Oncology* **31**, 3997-4013 (2013).
4. Camp RL, Charette LA, Rimm DL. Validation of tissue microarray technology in breast carcinoma. *Laboratory investigation; a journal of technical methods and pathology* **80**, 1943-1949 (2000).
5. Vis AN, Kranse R, Nigg AL, van der Kwast TH. Quantitative analysis of the decay of immunoreactivity in stored prostate needle biopsy sections. *American journal of clinical pathology* **113**, 369-373 (2000).
6. Balgley BM, Guo T, Zhao K, Fang X, Tavassoli FA, Lee CS. Evaluation of archival time on shotgun proteomics of formalin-fixed and paraffin-embedded tissues. *Journal of proteome research* **8**, 917-925 (2009).
7. Manne U, Myers RB, Srivastava S, Grizzle WE. Re: loss of tumor marker-immunostaining intensity on stored paraffin slides of breast cancer. *Journal of the National Cancer Institute* **89**, 585-586 (1997).
8. Jacobs TW, Prioleau JE, Stillman IE, Schnitt SJ. Loss of tumor marker-immunostaining intensity on stored paraffin slides of breast cancer. *Journal of the National Cancer Institute* **88**, 1054-1059 (1996).
9. Shin HJ, Kalapurakal SK, Lee JJ, Ro JY, Hong WK, Lee JS. Comparison of p53 immunoreactivity in fresh-cut versus stored slides with and without microwave heating. *Modern pathology : an official journal of the United States and Canadian Academy of Pathology, Inc* **10**, 224-230 (1997).
10. Banerji S, *et al.* Sequence analysis of mutations and translocations across breast cancer subtypes. *Nature* **486**, 405-409 (2012).
11. Stephens PJ, *et al.* The landscape of cancer genes and mutational processes in breast cancer.

Nature **486**, 400-404 (2012).

12. Shah SP, *et al.* The clonal and mutational evolution spectrum of primary triple-negative breast cancers. *Nature* **486**, 395-399 (2012).
13. Comprehensive molecular portraits of human breast tumours. *Nature* **490**, 61-70 (2012).
14. Ciriello G, *et al.* Comprehensive Molecular Portraits of Invasive Lobular Breast Cancer. *Cell* **163**, 506-519 (2015).
15. Huang CH, *et al.* The structure of a human p110alpha/p85alpha complex elucidates the effects of oncogenic PI3Kalpha mutations. *Science (New York, NY)* **318**, 1744-1748 (2007).
16. Zhao JJ, Liu Z, Wang L, Shin E, Loda MF, Roberts TM. The oncogenic properties of mutant p110alpha and p110beta phosphatidylinositol 3-kinases in human mammary epithelial cells. *Proceedings of the National Academy of Sciences of the United States of America* **102**, 18443-18448 (2005).
17. Zhao L, Vogt PK. Helical domain and kinase domain mutations in p110alpha of phosphatidylinositol 3-kinase induce gain of function by different mechanisms. *Proceedings of the National Academy of Sciences of the United States of America* **105**, 2652-2657 (2008).
18. Wu H, *et al.* Regulation of Class IA PI 3-kinases: C2 domain-iSH2 domain contacts inhibit p85/p110alpha and are disrupted in oncogenic p85 mutants. *Proceedings of the National Academy of Sciences of the United States of America* **106**, 20258-20263 (2009).
19. Jaiswal BS, *et al.* Somatic mutations in p85alpha promote tumorigenesis through class IA PI3K activation. *Cancer cell* **16**, 463-474 (2009).
20. Kang S, Bader AG, Vogt PK. Phosphatidylinositol 3-kinase mutations identified in human cancer are oncogenic. *Proceedings of the National Academy of Sciences of the United States of America* **102**, 802-807 (2005).
21. Backer JM. The regulation of class IA PI 3-kinases by inter-subunit interactions. *Current topics in microbiology and immunology* **346**, 87-114 (2010).
22. Pacold ME, *et al.* Crystal structure and functional analysis of Ras binding to its effector phosphoinositide 3-kinase gamma. *Cell* **103**, 931-943 (2000).
23. Echeverria I, Liu Y, Gabelli SB, Amzel LM. Oncogenic mutations weaken the interactions that stabilize the p110alpha-p85alpha heterodimer in phosphatidylinositol 3-kinase alpha. *The FEBS journal* **282**, 3528-3542 (2015).
24. Gkeka P, *et al.* Investigating the structure and dynamics of the PIK3CA wild-type and H1047R

oncogenic mutant. *PLoS computational biology* **10**, e1003895 (2014).

25. Chagpar RB, *et al.* Direct positive regulation of PTEN by the p85 subunit of phosphatidylinositol 3-kinase. *Proceedings of the National Academy of Sciences of the United States of America* **107**, 5471-5476 (2010).
26. Cheung LW, *et al.* High frequency of PIK3R1 and PIK3R2 mutations in endometrial cancer elucidates a novel mechanism for regulation of PTEN protein stability. *Cancer discovery* **1**, 170-185 (2011).
27. Shalem O, Sanjana NE, Zhang F. High-throughput functional genomics using CRISPR-Cas9. *Nature reviews Genetics* **16**, 299-311 (2015).
28. Bernards R, Brummelkamp TR, Beijersbergen RL. shRNA libraries and their use in cancer genetics. *Nature methods* **3**, 701-706 (2006).
29. Silva JM, *et al.* Profiling essential genes in human mammary cells by multiplex RNAi screening. *Science (New York, NY)* **319**, 617-620 (2008).
30. Shalem O, *et al.* Genome-scale CRISPR-Cas9 knockout screening in human cells. *Science (New York, NY)* **343**, 84-87 (2014).
31. Wang T, Wei JJ, Sabatini DM, Lander ES. Genetic screens in human cells using the CRISPR-Cas9 system. *Science (New York, NY)* **343**, 80-84 (2014).
32. Lai YL, Mau BL, Cheng WH, Chen HM, Chiu HH, Tzen CY. PIK3CA exon 20 mutation is independently associated with a poor prognosis in breast cancer patients. *Annals of surgical oncology* **15**, 1064-1069 (2008).
33. Barbareschi M, *et al.* Different prognostic roles of mutations in the helical and kinase domains of the PIK3CA gene in breast carcinomas. *Clinical cancer research : an official journal of the American Association for Cancer Research* **13**, 6064-6069 (2007).
34. Sabine VS, *et al.* Mutational analysis of PI3K/AKT signaling pathway in tamoxifen exemestane adjuvant multinational pathology study. *Journal of clinical oncology : official journal of the American Society of Clinical Oncology* **32**, 2951-2958 (2014).
35. Lavebratt C, Sengul S. Single nucleotide polymorphism (SNP) allele frequency estimation in DNA pools using Pyrosequencing. *Nature protocols* **1**, 2573-2582 (2006).

REVIEWERS' COMMENTS:

Reviewer #2 (Remarks to the Author):

The authors have sufficiently addressed previous comments on the Western analysis in this revised version by including quantifications and adjusting their statements concerning Figure 6D.

Shortening the description of potential functional mechanisms of the mutations in the Discussion section and moving that to Results section is recommended.

Author's discussion on assessment clinical significance of the impactful PIK3CA mutations is rather weak; the TCGA dataset lacks the novel mutations in PIK3CA and PIK3R1, prevalence of PIK3CA mutation in luminal subtype is not a novel finding, in survival analysis molecular intrinsic subtypes were not analyzed separately, recent studies on PIK3CA mutation status as a prognostic/predictive biomarker are poorly cited.

Reviewer #3 (Remarks to the Author):

The authors have now adequately addressed all my comments.

Reviewer #4

Editorial note:

Reviewer 4 commented for the editors only and supported publication. He/she suggested to include some IHC images in the supplementary file to allow verification in response to the concern raised by reviewer 1 regarding the assessment of ER/PR/HER2 status (comment 1).

Furthermore Reviewer 4 suggests caution in drawing strong conclusions on the frequency of mutations, given the small numbers overall.

Point-by-point response to the referees' comments

Reviewer #2 (Remarks to the Author):

1. The authors have sufficiently addressed previous comments on the Western analysis in this revised version by including quantifications and adjusting their statements concerning Figure 6D.

Response: We are very appreciative of the positive comment from the reviewer.

2. Shortening the description of potential functional mechanisms of the mutations in the Discussion section and moving that to Results section is recommended.

Response: We are very grateful for the reviewer's comment. Taking into account the reviewer's suggestion, we have rephrased the description of potential functional mechanisms of the mutations and have moved it to the **Results** section. (Line 260-292)

3. Author's discussion on assessment clinical significance of the impactful PIK3CA mutations is rather weak; the TCGA dataset lacks the novel mutations in PIK3CA and PIK3R1, prevalence of PIK3CA mutation in luminal subtype is not a novel finding, in survival analysis molecular intrinsic subtypes were not analyzed separately, recent studies on PIK3CA mutation status as a prognostic/predictive biomarker are poorly cited.

Response: We thank the reviewer for the kind reminder and apologize for the insufficient discussion regarding this point in our previous manuscript. We have carefully reviewed current literatures, reanalyzed the survivals in molecular intrinsic subtypes and revised our assessment regarding the clinical significance of the impactful *PIK3CA* mutations in the **Discussion** section, as follow:

"The clinical significance of *PIK3CA* mutations in breast cancer remains complicated and controversial in literatures. Sobhani et al suggested that mutated *PIK3CA* represent an independent negative prognostic factor in breast cancer¹; while several studies indicated that mutations in *PIK3CA* were associated with favorable prognostic factor and lower risk of relapse in unsorted breast cancer^{2,3,4}⁵; and furthermore, some studies suggested that *PIK3CA* mutations have no significant prognostic effect^{6,7}. However, the detailed stratification of molecular subtypes in recent years has helped reduce controversy on the topic. (1) Luminal subtype: In early breast cancers, *PIK3CA* mutations were more frequent in low-risk luminal BCs (lower grade, less lymph node involvement, and progesterone

receptor positivity)^{8, 9, 10}. Sabine et al suggested *PIK3CA* mutations were associated with favorable outcomes for luminal subtype in TEAM clinical trial⁸. In late ER-positive breast cancers with acquired resistance to endocrine therapy, *PIK3CA* mutations may behave as a mechanism of anti-estrogen resistance^{11, 12}. In these patients, the importance of the simultaneous inhibition of the PI3K/mTOR and ER pathway is supported by the results from recent BELLE-2 and BOLERO-2 clinical trials^{13, 14}. It remains possible and even likely that mechanisms of *de novo* endocrine resistance in early cancer differ from those in progressing cancer that achieve endocrine resistance by tumor evolution. (2) HER-2 subtype: In advanced HER-2 subtype patients, *PIK3CA* mutations were associated with poorer clinical outcome, as inferred from the biomarker analyses in CLEOPATRA clinical trial and other study^{15, 16}. Additionally, Loibl et al and Majewski et al both suggest that *PIK3CA* activating mutations predicted poor pCR in patients with HER2-subtype breast cancer treated with neoadjuvant anti-HER2 therapies (NeoALTTO and other trials)^{17, 18}. These results suggest *PIK3CA* activating mutations might drive resistance to anti-HER2 therapies. (3) TNBC: Takeshita et al reported that the presence of *PIK3CA* mutations was significantly correlated with positive phosphorylated form of androgen receptor which is an independent favorable prognostic factor of TNBC¹⁹. A continuous molecular profiling investigation on patients will help to elucidate the prognostic/predictive role of *PIK3CA* mutations in TNBC.

In this study, we reported strong correlations between the distributions of *PIK3CA* impactful mutations and hormone receptor positivity in breast tumors, whereas this relationship did not seem to hold true for non-impactful mutations. Unfortunately, impactful *PIK3CA* mutations did not seem to be related to the clinical outcome of breast cancer patients in the analyses of TCGA dataset; and they do not have prognostic significance in the subsequent stratification of molecular subtypes. One possibility may be the heterogeneity of patients' clinical characteristics and treatments in TCGA dataset. Many studies reporting the prognostic significance of *PIK3CA* mutations in breast cancer used data from clinical trials, thus excluding the effect of stage and treatment heterogeneity. Another possibility is that *PIK3CA* mutations may play various roles in tumorigenesis and drug resistance. *PIK3CA* impactful mutations identified in this study were proved to confer oncogenic activity; these aspects of their potentials were reflected in malignant transformation among normal mammary epithelial cells (MCF-10A and HMEC). Further characterizations of PI3K mutations in luminal and HER-2 subtype breast cancer would be necessary to uncover their potentials in conferring endocrine or anti-HER2 resistance in the future."

According to reviewer's kind suggestion, we have added the survival analysis in molecular subtypes in the **Results** section (Line 474-476) and updated the **Discussion** section (Line 650-712). We hope that our revised manuscript will be met with satisfaction.

Reviewer #3 (Remarks to the Author):

The authors have now adequately addressed all my comments.

Response: We appreciate the reviewer's kind comment.

Reviewer #4 Editorial note:

1. Reviewer 4 commented for the editors only and supported publication. He/she suggested to include some IHC images in the supplementary file to allow verification in response to the concern raised by reviewer 1 regarding the assessment of ER/PR/HER2 status (comment 1).

Response: We appreciate the reviewer's kind suggestion to further improve our manuscript. According to reviewer's comments, we have newly included some representative images of ER/PR/HER2 status in the supplementary file in our revised version.

2. Furthermore Reviewer 4 suggests caution in drawing strong conclusions on the frequency of mutations, given the small numbers overall.

Response: We thank the reviewer's comment for this point. We also recognized that a limited number of cases from one center may be insufficient to represent the overall characteristics of mutation pattern in a Chinese population. According to your kind suggestion, we have addressed this limitation in our revised manuscript and proceeded to draw a milder conclusion on the frequency of mutations (Line 181, 532-535). We hope that this revision will convey a more accurate message to the readers. In the future, we plan to perform an analysis of somatic mutations in the PI3K pathway through multi-centered clinical trials. We hope that our future work will provide a comprehensive mutation profile of the PI3K pathway in a Chinese population.

References:

1. Sobhani N, *et al.* The prognostic value of PI3K mutational status in breast cancer: a meta-analysis. *Journal of cellular biochemistry*, (2018).
2. Bernichon E, *et al.* Genomic alterations and radioresistance in breast cancer: an analysis of the ProfILER protocol. *Annals of oncology : official journal of the European Society for Medical Oncology* **28**, 2773-2779 (2017).
3. Abramson VG, *et al.* Characterization of breast cancers with PI3K mutations in an academic practice setting using SNaPshot profiling. *Breast cancer research and treatment* **145**, 389-399 (2014).
4. Loi S, *et al.* Somatic mutation profiling and associations with prognosis and trastuzumab benefit in early breast cancer. *Journal of the National Cancer Institute* **105**, 960-967 (2013).
5. Kalinsky K, *et al.* PIK3CA mutation associates with improved outcome in breast cancer. *Clinical cancer research : an official journal of the American Association for Cancer Research* **15**, 5049-5059 (2009).
6. Engels CC, *et al.* The clinical value of HER-2 overexpression and PIK3CA mutations in the older breast cancer population: a FOCUS study analysis. *Breast cancer research and treatment* **156**, 361-370 (2016).
7. Papaxoinis G, *et al.* Significance of PIK3CA Mutations in Patients with Early Breast Cancer Treated with Adjuvant Chemotherapy: A Hellenic Cooperative Oncology Group (HeCOG) Study. *PLoS one* **10**, e0140293 (2015).
8. Sabine VS, *et al.* Mutational analysis of PI3K/AKT signaling pathway in tamoxifen exemestane adjuvant multinational pathology study. *Journal of clinical oncology : official journal of the American Society of Clinical Oncology* **32**, 2951-2958 (2014).
9. Beelen K, *et al.* PIK3CA mutations, phosphatase and tensin homolog, human epidermal growth factor receptor 2, and insulin-like growth factor 1 receptor and adjuvant tamoxifen resistance in postmenopausal breast cancer patients. *Breast cancer research : BCR* **16**, R13 (2014).
10. Wilson TR, *et al.* The molecular landscape of high-risk early breast cancer: comprehensive biomarker analysis of a phase III adjuvant population. *NPJ breast cancer* **2**, 16022 (2016).
11. Mayer IA, Arteaga CL. PIK3CA activating mutations: a discordant role in early versus advanced hormone-dependent estrogen receptor-positive breast cancer? *Journal of clinical oncology : official journal of the American Society of Clinical Oncology* **32**, 2932-2934 (2014).
12. Dupont Jensen J, *et al.* PIK3CA mutations may be discordant between primary and

corresponding metastatic disease in breast cancer. *Clinical cancer research : an official journal of the American Association for Cancer Research* **17**, 667-677 (2011).

13. Baselga J, *et al.* Buparlisib plus fulvestrant versus placebo plus fulvestrant in postmenopausal, hormone receptor-positive, HER2-negative, advanced breast cancer (BELLE-2): a randomised, double-blind, placebo-controlled, phase 3 trial. *The Lancet Oncology* **18**, 904-916 (2017).
14. Hortobagyi GN, *et al.* Correlative Analysis of Genetic Alterations and Everolimus Benefit in Hormone Receptor-Positive, Human Epidermal Growth Factor Receptor 2-Negative Advanced Breast Cancer: Results From BOLERO-2. *Journal of clinical oncology : official journal of the American Society of Clinical Oncology* **34**, 419-426 (2016).
15. Baselga J, *et al.* Biomarker analyses in CLEOPATRA: a phase III, placebo-controlled study of pertuzumab in human epidermal growth factor receptor 2-positive, first-line metastatic breast cancer. *Journal of clinical oncology : official journal of the American Society of Clinical Oncology* **32**, 3753-3761 (2014).
16. Xu B, *et al.* Association of phosphatase and tensin homolog low and phosphatidylinositol 3-kinase catalytic subunit alpha gene mutations on outcome in human epidermal growth factor receptor 2-positive metastatic breast cancer patients treated with first-line lapatinib plus paclitaxel or paclitaxel alone. *Breast cancer research : BCR* **16**, 405 (2014).
17. Majewski IJ, *et al.* PIK3CA mutations are associated with decreased benefit to neoadjuvant human epidermal growth factor receptor 2-targeted therapies in breast cancer. *Journal of clinical oncology : official journal of the American Society of Clinical Oncology* **33**, 1334-1339 (2015).
18. Loibl S, *et al.* PIK3CA mutations are associated with lower rates of pathologic complete response to anti-human epidermal growth factor receptor 2 (her2) therapy in primary HER2-overexpressing breast cancer. *Journal of clinical oncology : official journal of the American Society of Clinical Oncology* **32**, 3212-3220 (2014).
19. Takeshita T, *et al.* Prognostic role of PIK3CA mutations of cell-free DNA in early-stage triple negative breast cancer. *Cancer science* **106**, 1582-1589 (2015).